# Structural basis for pH-responsive amino acid transport via SLC7A4.

Dimitrios Kolokouris [1,2,3], Anuja Bothra [1,2], Takafumi Kato[1,2], Yi C. Zeng [1,2], Simon Lichtinger [1,3], Joanne L. Parker [1,2], Philip C. Biggin [1,3] & Simon Newstead [1,2] ✉

The transport of amino acids across cell membranes is essential for metabolism, neuronal signalling, and immune system function. The amino acid polyamine organocation (APC) superfamily controls amino acid transport via mechanisms including amino acid exchange, facilitative diffusion, and sodium- or proton-coupled transport. Although many mammalian APC members functioning as exchangers and sodium-coupled systems have been identified, the mechanisms underlying pH-regulated amino acid transport in mammalian cells remain unclear. Here, we show that the plasma membrane amino acid transporter SLC7A4 is regulated by low extracellular pH and functions as a leucine transporter in human cells. Using Cryo-EM structures of the plant homologue, CAT4, from Arabidopsis thaliana in outward-open apo and L-ornithine-bound states, as well as transport assays and molecular dynamics simulations based on homology models of the human transporter, we identify residues responsible for amino acid selectivity that supports an allosteric mechanism linking ligand recognition to pH regulation. This mechanism is consistent with an evolutionary link to proton-coupled prokaryotic homologues. Overall, our findings provide a structural and functional basis for pH-gated leucine transport by the human SLC7A4 transporter and provides a framework for understanding amino acid selectivity within the wider SLC7 family.

Amino acids are essential for protein synthesis and serve as precursors for nucleotides, lipids, and neurotransmitters, while also functioning as intermediates in energy production via the TCA cycle[1,2]. Branched-chain and cationic amino acids, such as leucine, arginine, and lysine, play a crucial role in regulating cellular metabolism by activating the mTOR complex[3,4], supporting the immune system through T-cell activation, and facilitating cancer cell proliferation[5]. Understanding the molecular mechanisms underpinning amino acid transport within cells is therefore essential for developing new therapies for a range of human diseases, spanning immunology, oncology, metabolic, and neurological disorders[6,7].

Currently, around 40 different amino acid transporters have been identified, which belong to seven distinct solute carrier (SLC) families[8].

Among these, the SLC7 family plays an essential role in regulating the uptake and distribution of all 20 canonical amino acids in the body[9]. Many SLC7 members are the focus of clinical studies evaluating inhibitors to treat various types of cancer[10–14], neurological diseases[15] and to enable targeted drug delivery across the blood-brain barrier[16].

The SLC7 family comprises 13 members, and can be broadly divided into two subgroups: the cationic amino acid transporters (SLC7A1-A4; CATs) and the heteromeric amino acid transporters (SLC7A5-A13; HATs)[9,17]. The HATs comprise a catalytic transporter, or light chain (LAT; e.g., SLC7A5/LAT1, SLC7A8/LAT2), associated with a type-II glycoprotein heavy chain (SLC3A1 or SLC3A2) for trafficking. Both the CAT and HAT subfamilies are members of the larger APC superfamily[18], which shares a common 12-transmembrane (TM)

[1]Department of Biochemistry, University of Oxford, Oxford, UK. [2]Kavli Institute for Nanoscience Discovery, University of Oxford, Oxford, UK. [3]Structural Bioinformatics and Computational Biochemistry, Department of Biochemistry, University of Oxford, Oxford, UK. ✉e-mail: simon.newstead@bioch.ox.ac.uk

domain fold, commonly referred to as the LeuT fold[19]. Within this fold, the first five TM helices are related to the second five helices via a pseudo-two-fold symmetry axis running parallel to the plane of the membrane and characterised by TMs 1 and 6 adopting discontinuous regions in the middle of the transporter[20,21]. Structures of many eukaryotic HAT transporters have now been reported[22–30]. Interestingly, while these studies have revealed a nearly identical binding site for amino acids within the discontinuous regions of TMs 1 and 6, each specific member of the SLC7 family appears to have adopted a unique mechanism to select for specific pools of amino acids[9], making detailed studies necessary to understand amino acid selectivity within this transporter family[29].

The CATs and HATs share a close structural homology with amino acid transporters in the SLC36 and SLC38 families, which are often associated with pH-driven amino acid transport across organelle membranes, such as the endosome and lysosome[3,31,32]. However, to date, the role of pH in regulating transport within the mammalian SLC7 family remains understudied, as does the molecular mechanisms underpinning pH-regulated amino acid transport more generally within eukaryotic cells. Recently, we reported the mechanism of proton-coupled amino acid transport in a distantly related amino acid transporter from the hyperthermophilic bacterium *Geobacillus kaustophilus*[33], GkApcT, and noted the close homology with CAT homologues in *A. thaliana* and the human transporter SLC7A4, which is currently considered an orphan transporter[34].

Our previous study identified Glu115 on TM3, which, together with Asp237 on TM6, couples alanine transport in GkApcT to the proton electrochemical gradient[33,35]. Homologues of mammalian transporters in plants can provide valuable models for understanding human transporters[36]. Indeed, whereas mammalian CAT1, 2, and 3 operate as facilitators[37,38], in plants they are pH-dependent[39], similar to mammalian SLC36 and SLC38 organellar amino acid transporters[40]. *A. thaliana* has a cluster of nine CATs with close sequence similarity to mammalian CAT genes[39,41–43], of which CAT1, CAT5 and CAT6 have been reported to mediate pH-dependent amino acid uptake[44,45]. Intriguingly, all of the CAT homologues in *A. thaliana* conserve the acidic residue on TM3, whereas in the human SLC7s, only SLC7A4 conserves an acidic side chain in this position in the transporter (Supplementary Fig. 1a, b), leading us to speculate whether SLC7A4 might also function as a pH-regulated amino acid transporter in plants and humans.

The human SLC7A4 gene was first identified in 1998 and mapped to chromosome 22q11.2, a region often deleted in patients with velo-cardiofacial syndrome (VCFS, also known as Shprintzen syndrome)[46]. Although the gene shares high sequence identity to SLC7A1 (CAT1) and SLC7A2 (CAT2)[34], studies utilising *Xenopus* oocytes demonstrated that human SLC7A4 does not transport the cationic amino acids arginine or lysine[34]. Thus, many questions remained open concerning the function of SLC7A4, its mechanism of action and its links to human physiology and disease.

Here, we identify SLC7A4 as a pH-responsive plasma membrane leucine transporter in humans and demonstrate that its plant homologue can bind both cationic amino acids and leucine. Combining structural, biochemical and functional data, we further demonstrate the close evolutionary conservation between human and plant transporters and identify key side chains required for cationic amino acid recognition within the SLC7 family.

## Results
### Structure determination of SLC7A4/CAT4 from *Arabidopsis thaliana*
To identify suitable proteins for biochemical analysis, we screened SLC7A4 homologues from five different species, encompassing mammals, fish, and plants (Supplementary Fig. 1c, d). Based on recombinant expression levels, we chose to pursue the purification and structural characterisation of SLC7A4/CAT4 from *Arabidopsis*

*thaliana*, hereafter referred to as AtCAT4. AtCAT4 shares 35% identity and 57% similarity with human SLC7A4, increasing to 48% identity and 78% similarity for residues within 6 Å of the substrate-binding site. The plant CAT4 homologue was of particular interest, as it provided a closer evolutionary link to the human transporter than the bacterial counterpart, GkApcT (Supplementary Fig. 1b). Crucially, AtCAT4 retains conservation of a protonatable side chain at the equivalent position to Glu115 on TM3 in GkApcT (Supplementary Fig. 1a). Using differential scanning fluorimetry (nanoDSF) on detergent-purified AtCAT4, we screened the ability of all proteinogenic amino acids (excluding L-tryptophan, due to spectral overlap in the fluorescence signal), alongside non-proteinogenic amino acids L-ornithine and L-citrulline, to elicit thermal stabilisation (Fig. 1a). We observed notable stabilisation in the presence of cationic amino acids (CAAs) L-Arginine (2.0 °C), L-Lysine (3.0 °C) and L-Ornithine (3.2 °C), in addition to the branched chain amino acid (BCAA) L-Leucine (1.9 °C), demonstrating that AtCAT4 could recognise these AAs as ligands.

To gain further insight into how these AAs bind to AtCAT4, we next determined the cryo-EM structures in the apo state and in complex with L-Ornithine, which showed the largest thermal stabilisation, to final global resolutions of 3.3 Å for each structure (Fig. 1b, Supplementary Fig. 2a-b and Supplementary Table 1). The structures were determined in complex with a synthetic nanobody (sybody), which we generated to aid particle alignment during data processing (Supplementary Fig. 3). The structure of AtCAT4 adopts an outward open state, with the extracellular halves of TM1 and TM6 (named TM1b and TM6a) splayed apart from TMs 3 and 10 (Fig. 1b). The intracellular side of the binding site is tightly sealed through the packing of the intracellular halves of TM1 and TM6 (named TM1a and TM6b) against TMs 3 and 8, aided by the formation of salt bridges between TMs 1 and 5 and IL1 and TM8 (Supplementary Fig. 4a-b).

Although the apo AtCAT4 structure adopts the same outward-facing states observed in LAT1/SLC7A5 (PDB: 8KDP)[22] and y + LAT1/SLC7A7 (PDB: 9KJU)[24], the root mean square deviation (r.m.s.d.) values indicate only moderate global similarity (4.5 Å over 280 Cα and 5.6 Å over 240 Cα, respectively). However, we chose to use these LAT structures for our comparison, as they are the most evolutionarily close reference structures available. In contrast, recent CAT-family structures of SLC7A1 from *Mus musculus* (hereafter referred to as MmCAT1) and GkApcT capture different conformational states (inward-facing apo and occluded)[33,47,48].

Unlike the LAT1 and y + LAT1 transporters, AtCAT4 contains two additional helices, TMs 11 and 12 (Fig. 1b), which pack against TM13 and project outward from the side of the transporter. These two helices are located on the opposite side of the transporter from the single TM helices of SLC3A1 (rBAT) and SLC3A2 (CD98hc or 4F2hc), which form part of the heteromeric amino acid transporter (HAT) complexes and pack against TM4[28]. Instead, the additional helices adopt a conformation similar to that observed in MmCAT1, suggesting that the other members of the CAT subfamily likely share this overall architecture.

Although the function of these additional helices is currently unclear, the placement of the CAT-specific TM helices (TM11-12) creates an extended hydrophobic surface compatible with protein–protein contacts and raises the possibility that CAT-family members may form higher-order assemblies. In related LeuT/APC-fold transporters that lack these extra helices (e.g., AdiC) and in SLC12 cotransporters (e.g., NKCC1 and KCC2), homodimers are formed via the terminal helical pair of the core fold (TM11–TM12), which corresponds to TM13–TM14 in eukaryotic CATs (Fig. 1b); in this architecture, the CAT-specific TMs 11–12 would flank and potentially support the TM13–TM14 dimerisation interface[49,50]. Given that SLC7 HATs already function as obligate heterodimers with an SLC3 heavy chain, it will be interesting to determine whether the CAT subfamily also employs oligomerisation as a regulatory mechanism.

Two additional structural differences are also observed in the apo structure of AtCAT4. The most notable being that TM1 adopts a continuous helical configuration and does not form the canonical break observed in all SLC7 and APC structures to date[51] (Fig. 1b, c). The second is that extracellular loop 4 (EL4), which typically forms a short helical hairpin that packs against the transporter's TM1 and points towards the substrate entry tunnel, has flipped out, with the loop sitting above the transporter (Supplementary Fig. 4c, d). The space vacated by the flipping away of EL4 has been occupied by a Lauryl-maltose neopentyl glycol (LMNG) detergent molecule (Supplementary Fig. 4e, f). The acyl chains of the detergent extend into two hydrophobic pockets between TMs 7 and 8, with the maltose head groups positioned near the extracellular part of TM1, where a conserved arginine, Arg55, makes a hydrogen bond to the C3 hydroxyl from the sugar headgroup and the bound sybody through its amino terminus and CDR1 region.

The extension of TM7 to accommodate the new position of EL4 is recognised by the CDR3 loop of SybB5, which also packs against the TM5–TM6 loop of AtCAT4. The sybody further stabilises a bound cholesterol near TM6a via π-stacking with a conserved tryptophan in the scaffold region. To probe whether EL4 mobility or LMNG pushes the EL4 loop into the flipped-out state, we performed unbiased MD simulations starting from the outward-open cryo-EM structure of AtCAT4 (PDB:9HJK) and the inward-facing AlphaFold2 model, which places the EL4 loop in the canonical flipped-in position. Principal-component analysis of the pooled MD simulations reveals that the

dominant component (>80% of the variance) corresponds to a flip-out-to-flip-in conformational change in EL4, supporting our hypothesis that EL4 is highly dynamic in AtCAT4 (Supplementary Fig. 5a).

Interestingly, the EL4 loop remained stable, either flipped in or flipped out, in the simulations. While we did not observe an EL4 transition event during the simulations to flipped-out-to-flipped-in (or vice versa), we noted higher backbone RMSF values for the loop region in the outward-open state (where the EL4 loop is solvent-exposed) compared to the flipped-in conformation (Supplementary Fig. 5a).

Given the nature of the sybody interaction near EL4, we considered the possibility that sybody binding may have induced the structural changes discussed above in EL4 and TM1. We therefore obtained a structure of AtCAT4 without the sybody, albeit at a lower global resolution of 4.2 Å (Supplementary Fig. 6 and Supplementary Table 1). The maps were of sufficient quality to trace the 14 TM helices but lacked sufficient information to build several loops reliably, including EL4. However, the map clearly shows the presence of the symmetry-related loop region IL1, which forms the canonical hairpin with TM3. We thus conclude that whilst we cannot rule out the impact of the sybody on the position of EL4 in the higher-resolution structure, our data suggest that EL4 is dynamic and may flip away from the transporter in the outward-facing state. Further supporting this hypothesis, we note that focused 3D classification on sybody-free AtCAT4 yielded a minor class (34,329/120,868 particles, 28%) with low-resolution continuous density where we could model EL4 in flipped-in position packed against TM1b (Supplementary Fig. 5b). This minor

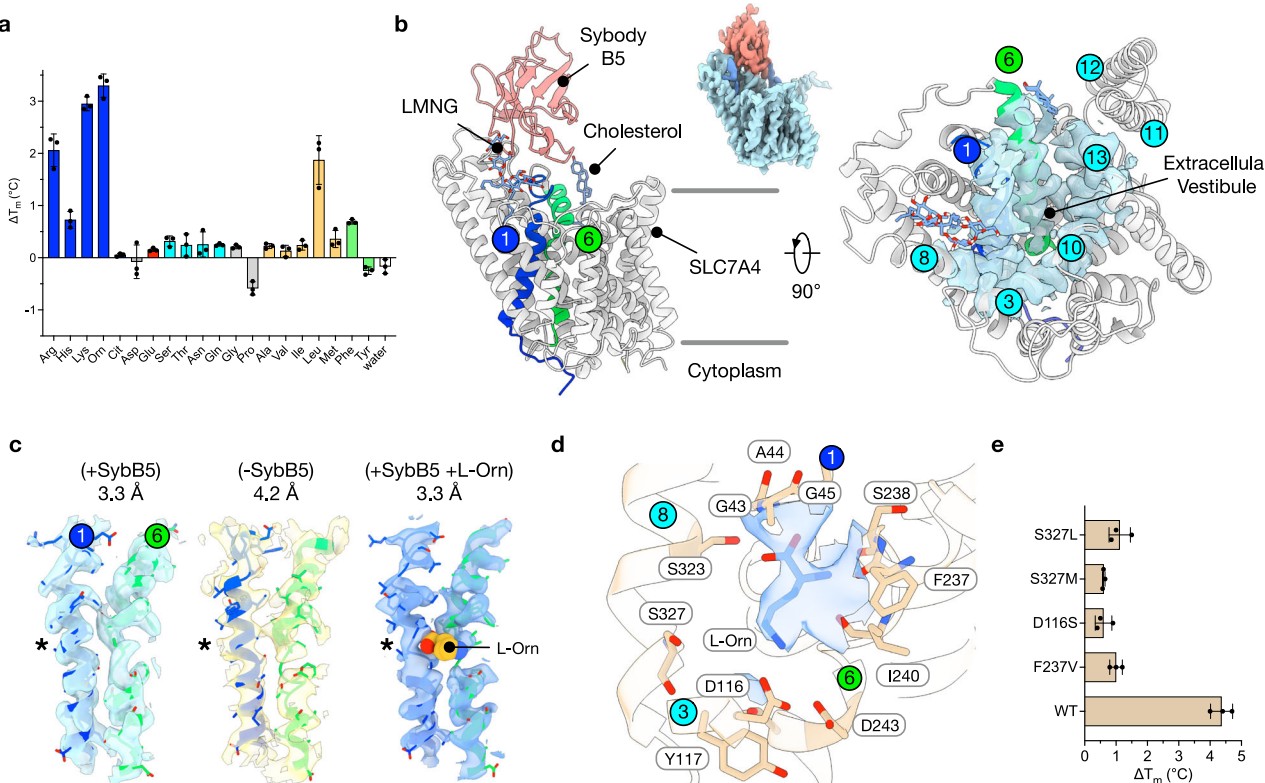

**Fig. 1 | Cryo-EM structure of AtCAT4. a** Thermal stability screen using nanoDSF, identifying interactions between AtCAT4 and amino acids. The measure of centre for the error bars is the mean. $n$ = three independent experiments; errors shown are standard deviations (s.d.). **b** Cryo-EM structure of apo AtCAT4 with TMs 1 and 6 highlighted and an LMNG and cholesterol molecules bound to the extracellular side of the outward-open transporter shown as blue sticks. Cryo-EM density (blue) is shown for the binding site, contoured at 0.45. Inset: cryo-EM density for the AtCAT4-Sybody complex, contoured at 0.45. **c** Cryo-EM density for TMs 1 and 6 for AtCAT4 bound to Sybody B5 (+SybB5; PDB 9HJK; contour 0.5), without the sybody

(-SybB5; PDB 9SQH; contour 0.2) and complexed to sybody B5 in the presence of 10 mM L-Ornithine (+SybB5, +L-Orn; PDB 9SP8; contour 0.1). The asterisk marks the region of TM1 that breaks following L-Orn binding. **d** The binding site showing the interaction with L-Orn. Key residues interacting with the substrate are shown as sticks. Cryo-EM density for the L-Orn is shown (blue), contoured at 0.09 ($\sigma$ = 6.5). **e** Effect of binding site mutations on the thermal stability of AtCAT4 in the presence of 10 mM L-Orn. n = three independent experiments; errors shown are s.d. Source data are provided as a Source Data file.

structural class further indicates that EL4 is mobile in the WT structure in LMNG and, together with the unbiased MD results discussed above, indicates that EL4 can sample multiple conformations. Accordingly, the absence of interpretable density for EL4 in the sybody-free reconstruction is best explained by the conformational heterogeneity of EL4 rather than limited map quality, since other flexible elements and the full 14-TM architecture remain traceable at the same resolution.

We also used the sybody-free map to validate our observation regarding the helical nature of TM1 (Supplementary Fig. 6). When compared internally, TM1 clearly adopts a helical structure. In contrast, the map reveals a break in the volume around Gly241 in TM6, which separates TM6a from TM6b (Fig. 1c). Taken together with the information from the higher resolution apo structure, our data further indicate that in the outward-facing state, TM1 forms a complete helix, rather than adopting a discontinuous configuration. As discussed below, this feature may play a crucial role in the amino acid recognition mechanism.

In the cryoEM map of the L-Orn-bound structure, we observed several densities consistent with lipid molecules (Supplementary Fig. 7a-b). On the extracellular leaflet, adjacent to the sybody, we observed density characteristic of sterol in a pocket formed between TM6 and TM13 (sterol site 1). This cholesterol was also observed in the apo structure. Additional cholesterol was added to the L-Orn sample, along with soy lipid extract; therefore, we have modelled this sterol as cholesterol in the cryo-EM maps. The sterol hydroxyl is located adjacent to the bulk solvent, near the sybody, and the aliphatic tail is inserted into a hydrophobic pocket formed by Leu229 and Ala233 from TM6 and Tyr546 from TM13. The steroid nucleus, in contrast, interacts with Trp57 on the sybody. A similar sterol pocket is observed in the L-Arg-bound structures of SLC7A1[48]. However, the cholesterol molecule is modelled further down into the bilayer contacting region in AtCAT4 (Supplementary Fig. 7c). It is possible that the interaction with the sybody pulled the cholesterol up through the hydrophobic channel between TM6 and TM13. Cholesterol plays a role in the function and structure of SLC7 transporters, most notably SLC7A5[52,53], where it binds to specific sites, modulating their structural dynamics[54] and transport activity[52]. The role of cholesterol in regulating the CAT subfamily, however, is underexplored. Our data indicate that this pocket is broadly conserved within the CAT subfamily and warrants further study to understand any regulatory role cholesterol might play in CAT transporter function.

Density for a second sterol was also observed at the opposite end of the transporter, interacting with a long extracellular loop between TM3 and TM4 (sterol site 2) (Supplementary Fig. 7a-b). This loop is also present in MmCAT1, where it forms an ordered alpha helix positioned above the membrane[47,48]. In contrast, in AtCAT4, this loop is disordered. This loop is a unique structural feature of the CAT subfamily, as the equivalent region in the HATs is much shorter, due to the presence of the heavy chain glycoprotein[28]. Several sterol molecules are also located in a similar region of SLC7A5, which appear to stabilise the interaction between the heavy chain and light chain of the transporter[55] (Supplementary Fig. 7d). It thus appears as though sterol molecules play an important and potentially conserved role in stabilising the structures of both the HATs and the CATs in the membrane.

## Structural basis for L-Ornithine recognition

The structure of AtCAT4 bound to L-Ornithine (L-Orn) adopts an outward-open conformation that closely matches the apo structure (Supplementary Fig. 2b), with an r.m.s.d. of 1.16 Å over 512 $C_a$ atoms. The structure provides valuable insight into the mechanisms of ligand capture in the CAT family when adopting an outward-open conformation. The cryo-EM map shows continuous, ligand-shaped density in the canonical substrate pocket. Its location and overall pose are consistent with the conserved α-amino-acid binding geometry

described across SLC7 transporters in outward-facing substrate-bound states[24,51,56,57].

The most striking difference compared with the apo structure is that TM1 now adopts the discontinuous configuration observed in previous APC family transporters[51]. The regular alpha helix of TM1 observed in the Apo structure is broken at Ala44, which sits within a conserved di-glycine ([43]GAG[45]) motif on TM1 (Fig. 1c and Supplementary Fig. 1a). The break in TM1 results from interactions with the carboxyl group of L-Orn via the backbone amide groups (Supplementary Fig. 2). Specifically, the backbone amide groups of Ala44 and Gly45 move ~ 2.8 Å towards the extracellular space, relative to the Apo structure, forming a short π-helix that accommodates the amide group of the L-Orn ligand. L-Ornithine makes further interactions with the backbone carbonyl groups of Phe237, Ser238 and Ile240 on TM6a. However, the ε-amino group on the L-Orn side chain forms a salt bridge with the side chain of Asp116 on TM3 (heavy-atom distance 3.2 Å) and is positioned within 5.8 Å of Asp243 on TM6b (Fig. 1d and Supplementary Fig. 8a). Despite water-like densities present near Asp243 and Ser327 on TM8, at 3.3 Å global resolution, we do not attempt to assign individual water molecules.

Interestingly, the binding pose of L-Orn is different from that observed for the closely related CAT homologue, MmCAT1 (SLC7A1; PDB: 9FQW, RMSD = 6.35 Å) (Supplementary Fig. 8b). In MmCAT1, the amino group on the side chain orientates towards TM1, interacting with a backbone carbonyl group on this helix and making a further hydrogen bond to a conserved serine on TM3[48]. However, the binding pose observed in our structure is very similar to the reported pose for L-Arg bound to SLC7A1 (PDB:9FQU), wherein the guanidino group hydrogen bonds with the hydroxyl and carbonyl groups of Ser120 on TM3. Ser120 in SLC7A1 corresponds to Asp116 in AtCAT4, which is consistent with our observations on the importance of this side chain in L-Orn recognition. The binding pose we observe for L-Orn in AtCAT4 is also consistent with the pose we reported previously for the more distantly related prokaryotic homologue, GkApcT (PDB: 6F34), which was captured bound to L-Arginine. In the GkApcT L-Arg bound structure, we also observed an interaction of the side chain guanidino group with Glu115 on TM3[33] (Supplementary Fig. 8c). Our structural analysis, therefore, suggests that the presence of Asp116 on TM3 (Glu125 in human SLC7A4) results in a mechanism of L-Orn recognition that is more similar to GkApcT than SLC7A1. The hydrophobic gating residue, Phe237 in AtCAT4 (Tyr246 in human SLC7A4), is in an open conformation, i.e., not capping the substrate, similar to the L-arginine-bound y+LAT1 structure (PDB: 8YLP). Our structure reveals an equivalent conformation for the hydrophobic lid, a hallmark of the outward-open state in the APC fold transporter family[24,58,59].

Unfortunately, we were unable to observe CAA transport for AtCAT4 in either Xenopus oocytes or HeLa cells overexpressing the protein, in part due to low expression levels. No activity was observed following liposome reconstitution despite the protein being homogeneously purifiable, well-folded, and dose-responsive to increasing concentrations of L-Orn (Supplementary Fig. 3e-g). Therefore, to validate the L-Orn binding pose, we employed the previously used nano-DSF binding assay. To verify the observed stabilisation, we mutated the hydrophobic gating residue on TM6a, Phe237, to valine. In all currently reported APC family transporters, Phe237 occupies a strictly conserved aromatic position in TM6a known as the thin gate, and mutations of this position are known to generate transport-deficient mutants of SLC7 proteins[22,29,33,60]. We observed no detectable stabilisation in the nanoDSF assay with the Phe237Val mutant (Fig. 1e). Within the binding site, we also mutated Asp116 to serine, as this would remove the acidic side chain while retaining the hydrogen bonding potential observed in SLC7A1, and Ser327 which, as discussed above, was observed to interact with L-Orn in SLC7A1, to either leucine as observed in the human SLC7A4 structure or methionine which was shown to modulate AA specificity in GkApcT[33]. All three mutations

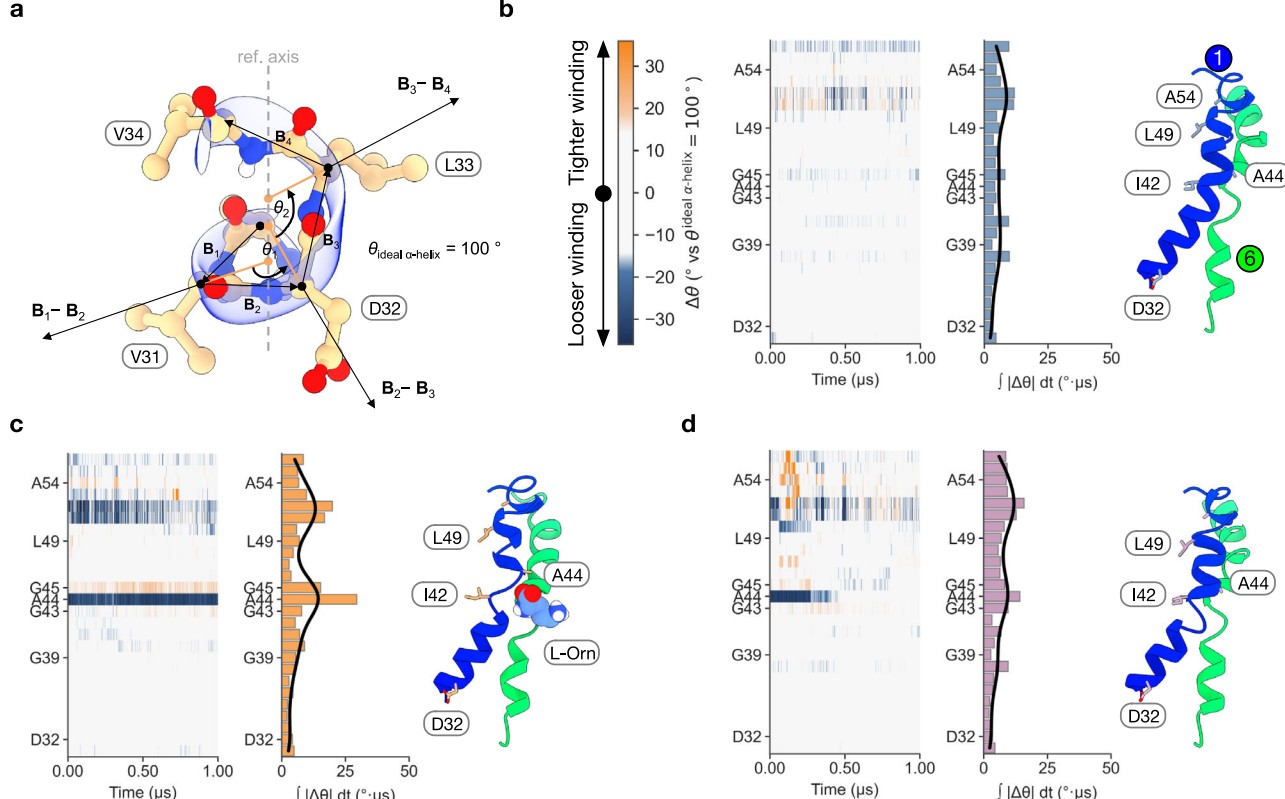

**Fig. 2 | Analysis of TM1 dynamics in response to L-Orn unbinding. a** Structure of TM1 showing the per-residue helical twist angle (θ) used to calculate the time-integrated deviation from the ideal alpha helix ( $\int |\Delta\theta|$ dt (°·µs)). **b** Analysis of TM1 dynamics in the Apo state from one of three independent simulations. Left−2D plots showing the local deviation of helical unwinding observed in TM1 over the 1 µs simulation. Middle−bar chart showing the time-integrated deviation of each residue in TM1. Right−The structure of TMs 1 and 6 at the end of the simulation, 1 µs. **c** Analysis of TM1 dynamics in the presence of L-Orn in repeat one from three independent simulations. **d** Analysis of TM1 dynamics following the unbinding of L-Orn in simulation repeat three.

disrupted the stabilisation observed in the presence of L-Orn in the WT protein (Fig. 1e), supporting our conclusion that AtCAT4 specifically recognises CAAs and the BCAA L-leucine.

**Amino acid-induced structural changes in TM helix 1**

The unwinding observed in TM1 following L-Orn binding prompted us to explore the possibility of an induced fit interaction between AtCAT4 and the ligand. We therefore employed molecular dynamics (MD) to study the impact of L-Orn binding on the geometry and dynamics of TM1. To ascertain the changes in TM1 geometry in the apo (PDB: 9HJK) and L-Orn bound structures (PDB: 9SP8), we employed the HELANAL algorithm[61], which measures the geometry of helices based solely on their Cα atoms (Fig. 2a). The benefit of this approach is that it allows us to map the deviation of the local twist of each Cα atom from an ideal α-helix ($\Delta\theta = \theta$ - 100°; where θ is computed from a sliding 4-Cα window) along the length of TM1 over the course of the simulation and summarise the magnitude of these changes along the helix as the time-integrated deviation ( $\int |\Delta\theta|$ dt (°·µs)) from ideal α-helix geometry. The deviation of each Cα-atom can then be compared to the values obtained for TM1 in the apo state (Fig. 2b), offering a localised analysis of any structural changes that occur during the MD simulation.

Triplicate 1 µs MD simulations starting from the L-Orn bound cryo-EM structure were run, where AtCAT4 remained stable in the POPC:cholesterol lipid membrane (Supplementary Fig. 9a-c and Table 2), and the helicity of TM1 was analysed. Across 3 × 1 µs unbiased replicas, the carboxylate of L-Orn unbound TM1 in 2 out of 3 simulations (rep2-3; Supplementary Fig. 9c). In the 1 µs repeat where L-Orn remained bound to TM1 (rep1; indicated by % H-bond occupancy and L-

Orn−TM1 minimum distance shown in Supplementary Fig. 9c), we observed an increase in the unwinding of TM1 centred at Ala44, which is sandwiched between the di-glycine GAG[43] motif in TM1 (Fig. 2c). Presumably, the di-glycine motif provides the flexibility to unwind in this region of the helix. The deviation from helical geometry in this region is consistent with the cryo-EM structure of AtCAT4 bound to L-Orn and with previous APC family transporters that show a discontinuity in TM1 at this location[51]. Helix TM1 unwinding at the GAG motif facilitates better accommodation of the L-Orn carboxylate at the binding site. Supporting this, in the MD simulation we observe a water molecule entering the binding site and coordinating the interaction between the carboxylate group of L-Orn and Gly45 from TM1, as well as Tyr274 from TM7 (Supplementary Fig. 9d). A similar water molecule was previously observed in our GkApcT crystal structure, coordinating the L-Ala ligand[33].

In the second and third repeats of the simulation, we observed L-Orn unbinding from TM1. As for replicate 1, we quantified the visually observed L-Orn unbinding by tracking its engagement with TM1 (as % H-bond occupancy) and minimum L-Orn distance from TM1 (Supplementary Fig. 9c). Interestingly, the TM1 unbinding events in replicates 2 and 3 were different. In simulation two, L-Orn unbound and diffused out of the binding site, whereas in simulation three, the L-Orn carboxylate group disengaged the GAG motif, along with the α-amino group disengaging TM6, with the ligand remaining in the binding site. The common observation in both of these simulation repeats, where L-Orn disengages TM1, is that TM1 reverts to the all helical configuration (Fig. 2d) of the apo state (Fig. 2b). Therefore, the helical break

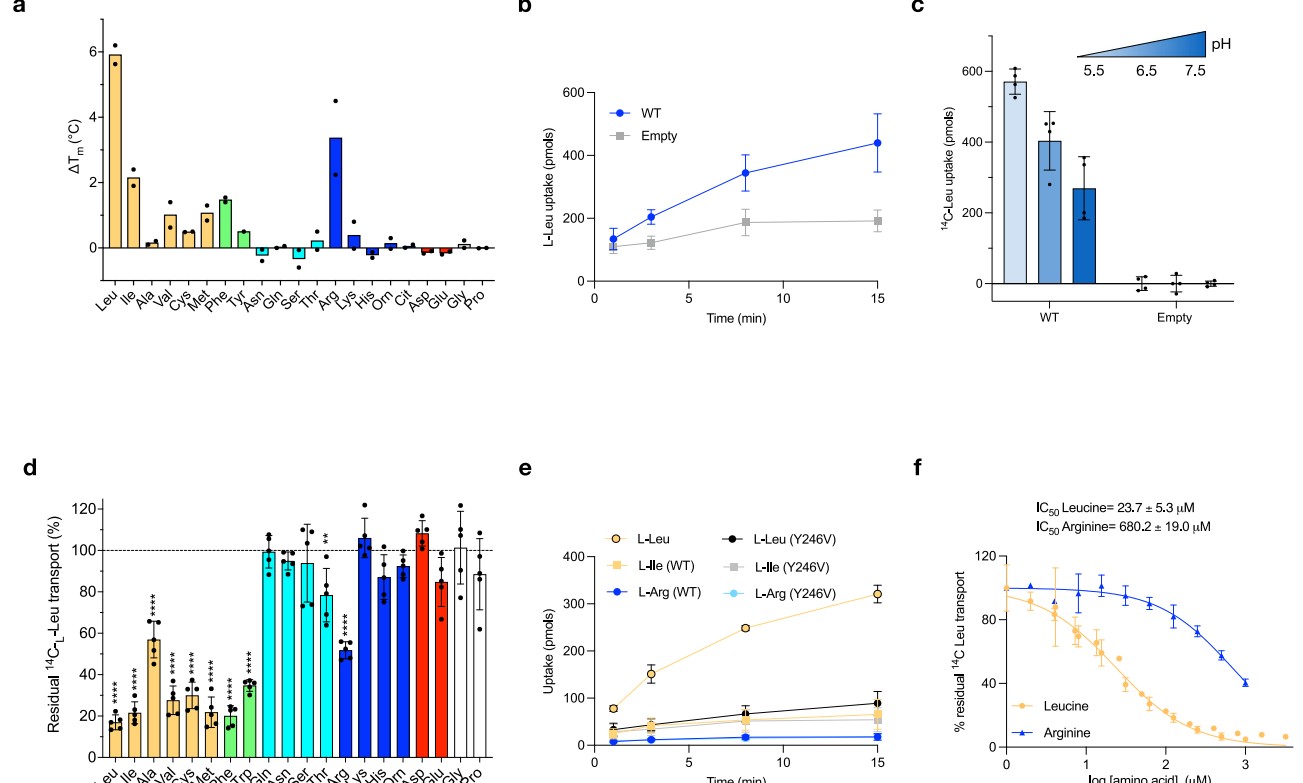

**Fig. 3 | Functional characterisation of human SLC7A4. a** Thermal stability screen identifying interactions between SLC7A4 and amino acids. n= two independent experiments. **b** Uptake of [14]C L-Leu in HCT116 SLC7A5 KO cells overexpressing SLC7A4. Empty refers to the empty plasmid control. $n$ = four independent experiments; errors shown are s.d. **c** Effect of external pH on [14]C L-Leu uptake at $t$ = 15 min in HCT116 SLC7A5 KO cells. Empty refers to the empty plasmid control. $n$ = four independent experiments; errors shown are s.d. **d** Assay data showing the ability of 500 µM external amino acids to compete for [14]C L-Leu uptake (reference set to 100%) in HCT116 SLC7A5 KO cells. $n$ = five independent experiments; errors

shown are s.d. Statistical significance tests were performed using one-way ANOVA compared to the hot-only [14]C-Leu control, with Dunnett's multiple comparisons (two-sided, family-wise α = 0.05). ns $P > 0.05$; * $P \le 0.05$; ** $P \le 0.01$; *** $P \le 0.001$; **** $P \le 0.0001$. **e** Cell-based uptake assays with [14]C L-Leu, [3]H L-Ile, and [3]H L-Arg compared to a transport-deficient mutant, Tyr246Val (Y246V). $n$ = four independent experiments; errors shown are s.d. **f** Cell-based IC50 measurements for L-Arg and L-Leu. n = three independent experiments; errors shown are s.d. Data are mean ± s.d. Source data are provided as a Source Data file.

induced by L-Orn binding in TM1 appears reversible upon ligand-TM1 disengagement.

To investigate the coordination and stable binding of L-Orn in replicate 1, we clustered the sampled binding site configurations (defined as residues within 10 Å of L-Orn) from the 1 µs unbiased MD simulation and further seeded five independent 200 ns unbiased simulations (see "Methods" and Supplementary Table 2). These seeded simulations yielded stable L-Orn binding in 5/5 runs, allowing us to collect atomistic interaction fingerprint statistics (Supplementary Fig. 9d, e). Compared to the cryo-EM model, Ala44 is shifted by an additional -1.7 Å over the span of the MD repeats, further drawing the ligand backbone carboxyl into the fully accessible GAG backbone pocket and the Gly45/Tyr274-mediated water bridge. Taken together, our results indicate that AtCAT4 undergoes an induced fit mechanism to recognise the L-Orn amino acid backbone, with TM1 undergoing an ordered-to-disordered transition to accommodate the carboxylate of the ligand. The positively charged ε-amino side chain was stabilised through favourable interactions with polar binding site residues, including Asp116 and Ser327 and pi-cation with Tyr117.

Across the five seeded simulation repeats, we consistently observe a water molecule occupying the gap between the L-Orn ε-amino group and Asp116, present for ~98% of the gross simulation time (Supplementary Fig. 9d, e). The presence of water in this position within the binding site is mechanistically reasonable, especially given that the interacting AtCAT4 side chain is an aspartate, which is shorter than glutamate, and the ligand is ornithine, which is shorter than either

lysine or arginine. Thus, a water molecule can stabilise the bound L-Orn molecule, as observed with the shorter aspartate side chain in AtCAT4. Collectively, the MD data suggest a coordinated water network is likely to exist within the AtCAT4 binding site. However, due to the resolution of the cryo-EM dataset we obtained, we could not reliably model the water molecules into the structure. However, our observations are consistent with previous SLC7/HAT transporter structures, which are reported to function via an induced-fit mechanism[29,60]. It is therefore reasonable to conjecture that ligand-induced structural changes may constitute an essential part of ligand selection within the wider SLC7 family.

## Human SLC7A4 is a pH-regulated leucine transporter

Given that nanoDSF could be used to verify that AtCAT4 binds cationic amino acids, we used the same methodology to analyse the substrate specificity for the human homologue (Uniprot: O43246). Of the 20 canonical amino acids tested, we observed the greatest stabilisation in the presence of L-Leu (-6 °C), suggesting that SLC7A4 in humans may transport the BCAA leucine (Fig. 3a). We sought to verify this using cell-based transport studies. Many commonly used cell types exhibit high background uptake of L-Leu due to upregulation of SLC7A5 (LAT1). To reduce background transport, we used an SLC7A5 knock-out HCT-116 cell line to study the transport characteristics of SLC7A4[62]. Having established that SLC7A4 is expressed at the plasma membrane (Supplementary Fig. 10a), we observed uptake of [14]C L-Leu in cells transfected with SLC7A4 at pH 7.5 compared to cells transfected with empty

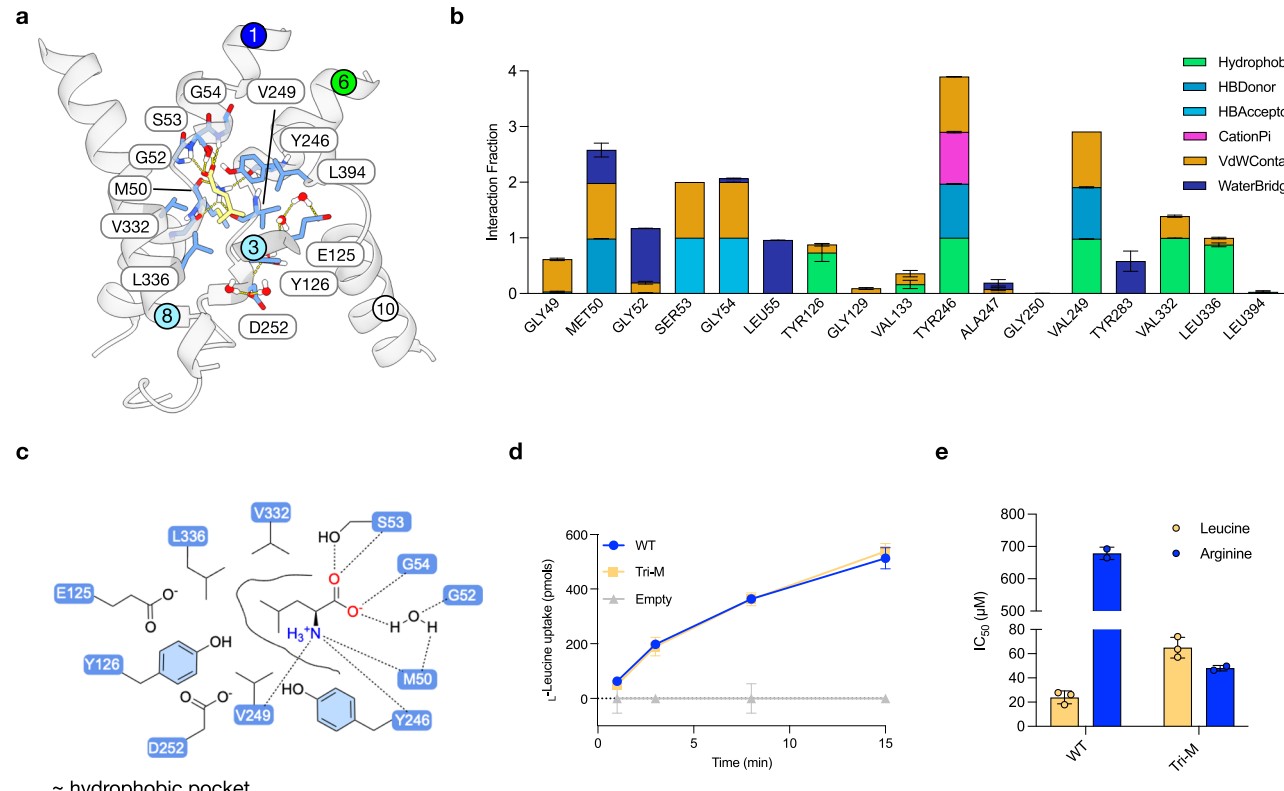

**Fig. 4 | Structural basis for L-leucine selectivity in human SLC7A4. a** L-Leucine (yellow) bound to the MD-refined homology model of Hs SLC7A4 after 1.2 μs total simulation. Key residues interacting with the substrate are shown as sticks, and hydrogen bonds are represented as dashed lines. **b** Bar chart showing the per-residue interaction profile, averaged across simulation replicates, with standard deviation reported per interaction type. n = three independent 200 ns replicates seeded from the cluster median and from a single 1000 ns simulation; errors shown are s.d. **c** Schematic L-Leu binding interactions. **d** Uptake of [14]C L-Leu in HCT116 SLC7A5 KO cells overexpressing WT SLC7A4 (WT) and the V332S/L336S/L394M triple mutant (Tri-M). Empty refers to the empty plasmid control. n = three independent experiments; errors shown are s.d. **e** IC50 measurements for L-Arg and L-Leu. n = three independent experiments; errors shown are s.d. Data are mean ± s.d. Source data are provided as a Source Data file.

plasmid (Fig. 3b). We initially focused on SLC7A4 as a potential pH regulated amino acid transporter due to the conservation of Glu125 on TM3 with GkApcT (Supplementary Fig. 1a, b). We therefore tested the effect of lowering external pH on leucine uptake in cells and in reconstituted proteoliposomes. Consistent with our hypothesis, we observed a substantial increase in L-Leu uptake at pH 6.5 and 5.5 compared to pH 7.5 in HCT-116 cells, indicating that SLC7A4 is activated at acidic pH (Fig. 3c). We also observed uptake of L-Leu into proteoliposomes under an acetate-induced pH gradient (Supplementary Fig. 10c-d).

After confirming that SLC7A4 facilitates L-Leu uptake across a physiological pH range and that transport efficiency increases under acidic conditions, we proceeded to define the transporter's broader substrate profile. As nanoDSF indicated the ability to recognise L-Arg (Fig. 3a), we subsequently screened whether the 20 canonical amino acids could compete for [14]C Leucine cell-based uptake (Fig. 3d). We observed that several non-polar amino acids could compete with leucine transport, suggesting that SLC7A4 may function as a pH-sensitive general non-polar amino acid transporter. However, when we tested the uptake of H3 L-Isoleucine (L-Ile), which is chemically very similar to leucine, or H3 L-Arg, which inhibited leucine uptake by 50%, we did not observe transport above a transport-deficient mutant (Tyr246Val) (Fig. 3e). Finally, we determined the IC50 for L-Leu as 24 ± 5 μM. The much lower IC50 value for L-Leu compared with the IC50 of 680 ± 19 μM for L-Arg (Fig. 3f) demonstrates a clear preference for L-Leu binding to SLC7A4, consistent with our transport data.

## The structural basis for leucine selectivity in SLC7A4

Compared to the HAT subfamily, much less is known concerning the molecular basis of amino acid selectivity within the CAT subfamily[9]. We therefore examined the structural basis for AA selectivity within SLC7A4. Despite collecting a large cryo-EM dataset of SLC7A4 in complex with L-Leu (ca. 30,000 movies), we were unable to solve the structure of the human transporter in complex with its ligand. We therefore used the cryo-EM structure of L-Orn-bound AtCAT4 to generate a homology model of the human transporter in the outward-facing state. L-Orn was mutated in silico to L-Leu, and the binding site was energy-minimised using the AMBER force field (see "methods"). The system was then equilibrated for 1 μs of unrestrained MD. We then monitored the stability of the complex using all-atom MD from triplicate 200 ns simulations seeded from the 1 μs MD (Fig. 4a-c, Supplementary Fig. 11a and Table 2). The MD data show that the carboxylate and amino group of L-Leu interact similarly with those observed with L-Orn, discussed above for AtCAT4, and the ligand pose was stable over three repeats of 200 ns of simulation (Supplementary Fig. 11b-c). Specifically, the carboxylate interacts with the backbone amides in Gly52, Ser53 and Gly54, which constitute the di-glycine motif in TM1 in the human transporter (Fig. 4a-c). The amino group of L-Leu interacts with the backbone carbonyl groups of Met50, Tyr246 and Val249. The L-Leu side chain is accommodated by hydrophobic residues Val249, Val332, Leu336, and Tyr246, the latter of which forms the thin extra-cellular gate.

As a CAT-subfamily member, we reasoned that SLC7A4 might retain a latent evolutionary propensity for cationic amino-acid recognition, in line with the nanoDSF binding and competition assays (Fig. 3a, d). To investigate whether it was possible to switch SLC7A4 from binding L-Leu to L-Arg, we compared SLC7A4 to the binding site of MmCAT1 (PDB:9FQW), a specialised CAA transporter[48]. Within the binding site of SLC7A4, we identified three side chains - Val332Ser, Leu336Ser, and Leu394Met - that are unique to SLC7A1 but differ in SLC7A4/CAT4 (Supplementary Fig. 1a). Encouragingly, Val332 and Leu336 were also observed in our MD analysis of the homology model, while Leu394 on TM10 is within ~ 5 Å of the bound L-Leu and lines the ligand entrance path.

To experimentally test our hypothesis that these side chains influence amino acid selectivity in SLC7A4, we created a triple mutant (Val332Ser, Leu336Ser, and Leu394Met), hereafter referred to as Tri-M. In our cell-based uptake assays using the HCT-116 SLC7A5 KO cell line, we observed robust uptake of $^{14}$C-Leu in the Tri-M comparable to the WT protein (Fig. 4d). After confirming that the Tri-M protein could recognise L-Leu, we then tested L-Arg recognition. Previously, we used a competition assay to determine the $IC_{50}$ for L-Leu and L-Arg in the WT protein (Fig. 3e). The $IC_{50}$ for L-Leu in the Tri-M increased slightly (65 µM ± 8.5 vs. 23.7 ± 5.3 µM for the WT), likely due to mutation of the hydrophobic pocket (Fig. 4e). However, there was a dramatic increase in affinity for L-Arg, with the $IC_{50}$ decreasing from 680.2 ± 19.0 µM in the WT to 48.6 ± 2.3 µM in the Tri-M. The hydrophobic pocket, Val332/Leu336/Leu394, is tuned for a branched side chain of leucine, providing a simple explanation for why closely related BCAAs can compete for access yet fail to show measurable uptake (e.g., isoleucine does not show detectable transport, (Fig. 3d-e), consistent with a balanced requirement for side-chain fit that couples binding to the conformational activation step.

Our data thus demonstrate that, as with the bacterial GkApcT transporter, amino acid selectivity in SLC7A4 can be modulated by tuning recognition of the ligand side chain. Our data, therefore, place SLC7A4 within the CAT subfamily in terms of structural homology with AtCAT4 but demonstrate that SLC7A4 diverged from the specialised human CAA transporters, CAT1, CAT2 and CAT3, to select for L-Leu. Strikingly, swapping just three residues (Val332Ser/Leu336Ser/Leu394Met) was sufficient to re-establish a preference for L-Arg recognition (Fig. 4d, e), demonstrating that amino acid specificity within the CAT subfamily is determined through only a small set of sidechains in the binding site.

Taken together, our data also explain why the Arabidopsis CAT4 and human SLC7A4 homologues have diverged towards preferential recognition of L-ornithine versus L-leucine, and why this divergence is physiologically coherent. In AtCAT4, the binding pocket remains highly polar and is organised around the conserved TM3 acidic residue (Asp116) and the surrounding hydrogen-bonding network. This network stabilises protonated, flexible side chains such as L-ornithine (and related CAAs), while still tolerating L-Leu binding as indicated by our nanoDSF screen. Prior work further reports that AtCAT4 localises predominantly to the tonoplast and is broadly expressed[63]. In this context, an Orn/CAA-biased pocket is most consistent with a role in vacuolar amino-acid buffering and nitrogen remobilisation[64,65]. It would help control cationic amino acid pools that feed plant nitrogen metabolism and polyamine/arginine-ornithine metabolism during growth and reproductive transitions.

In contrast, HsSLC7A4 has remodelled the side-chain environment around the canonical TM1-TM6 motif into a predominantly hydrophobic pocket, centred on Val332, Leu336 and Leu394. This pocket tightly accommodates the branched side chain of L-Leu but disfavours the more extended and charged side chains of cationic amino acids. This structural feature is reflected functionally by ~30-fold lower $IC_{50}$ values for L-Leu compared with L-Arg and by the absence of detectable L-Arg transport, despite measurable L-Arg binding. The Tri-

M mutant, in which we introduce three CAT1-like residues into this pocket, re-establishes high-affinity L-Arg recognition and transport while modestly weakening L-Leu affinity. This result demonstrates that a small set of side chains can switch the SLC7A4/CAT4 scaffold between an Orn/CAA-biased state suited to plant vacuolar homeostasis and a Leu-biased state suited to mammalian leucine handling.

## Glu125 functions as a pH sensor

To date, the pH dependence of amino acid transport via the SLC7 family remains unclear. However, proton-coupled homologues have been identified in the SLC36 family[31] and in prokaryotic APC superfamily homologues[33,66]. Previously, we identified Glu115 in TM3 as important for proton coupling in GkApcT, with protonation favouring the occluded state and deprotonation driving the transition to the inward-open state[33,35]. SLC7A4 also has a glutamate, Glu125, at the same position, whereas SLC7A1-3 in humans, which are not pH-regulated[9], contain serines at the equivalent site (Supplementary Fig. 1a), while retaining Asp252 in TM6. Mutating Asp252 to Ala resulted in a transport-deficient mutant, similar to GkApcT[33], suggesting that this residue is integral in the transport mechanism (Fig. 5a). We therefore aimed to understand the role of Glu125 in the transport mechanism of SLC7A4. We first mutated Glu125 to serine to mimic CAT1/2/3 members. In line with our hypothesis, the replacement of Glu125 with serine abolished the pH-induced activation at acidic pH. However, mutation of Glu125 to aspartate, which still can accept protons, remains pH-sensitive, showing increased activity at acidic pH. We also tested glutamine as it resembles a protonated glutamate, and since glutamine side chains can nucleate transient water wires, stabilising hydrogen-bonded networks that traverse hydrophobic pores[67]. Replacement of Glu125 with glutamine also abolished the observed pH response, suggesting that the acidic side chain of Glu125 serves as a titratable site. Overall, our data support a mechanism in which Glu125 is protonated under acidic conditions, thereby enhancing leucine transport in SLC7A4. Interestingly, all mutations involving a shorter side chain than glutamate result in higher transport, even after taking expression levels into account (Supplementary Fig. 10b). An explanation for this observation is discussed below in the context of the structural comparison between the cryo-EM structure AtCAT4 and our homology model of SLC7A4.

We next turned to our L-Leu-bound MD model of SLC7A4 (Fig. 4a) to rationalise these results. Interestingly, the stable pose adopted by Glu125 in the MD-refined model differs from that observed for the equivalent side chain, Asp116, in the L-Orn bound Cryo-EM structure of AtCAT4 (Fig. 5b). In SLC7A4, TM3 has rotated ~ 45° to bring Glu125 into proximity with TM10, where the side chain interacts with the discontinuous region, we observed in the outward-open AtCAT4 structure. A similar discontinuous region in TM10 is observed in recent outward-facing structures, such as SLC7A5 (LAT1; PDB: 8KDD)[22], SLC7A7 (y+LAT1; PDB: 8XXI)[24], SLC7A6 (y+LAT2; PDB: 9LDR)[25], the γ-aminobutyric acid transporter (GAT1; PDB: 8GNK)[68], and the bacterial arginine-agmatine antiporter, AdiC (PDB: 3HQK), where it was shown to play a role in ligand selection and transport[69]. In SLC7A4, the carboxylate group of Glu125 interacts with the backbone amide groups of Ser395, Leu396, Gly397, and Thr398, as well as the hydroxyl group of Thr398. In this position, the central ligand binding site is still open to the extracellular side of the membrane, although it is less open than our AtCAT4 structure due to the movement of Tyr246 on TM6a over the ligand. Nevertheless, this conformation provides important clues to the function of Glu125. Supporting the protonation mechanism, PROPKA3[70] calculates the pKa of Glu125 as slightly elevated (5.0).

Most notable is that Glu125 forms part of a solvent accessible tunnel that connects the amino acid binding site to the exterior of the cell (Fig. 5c). In this position, Glu125 could function as a responsive conformational sensor. Under neutral conditions, Glu125 would be deprotonated, stabilising the interaction between TM3 and the

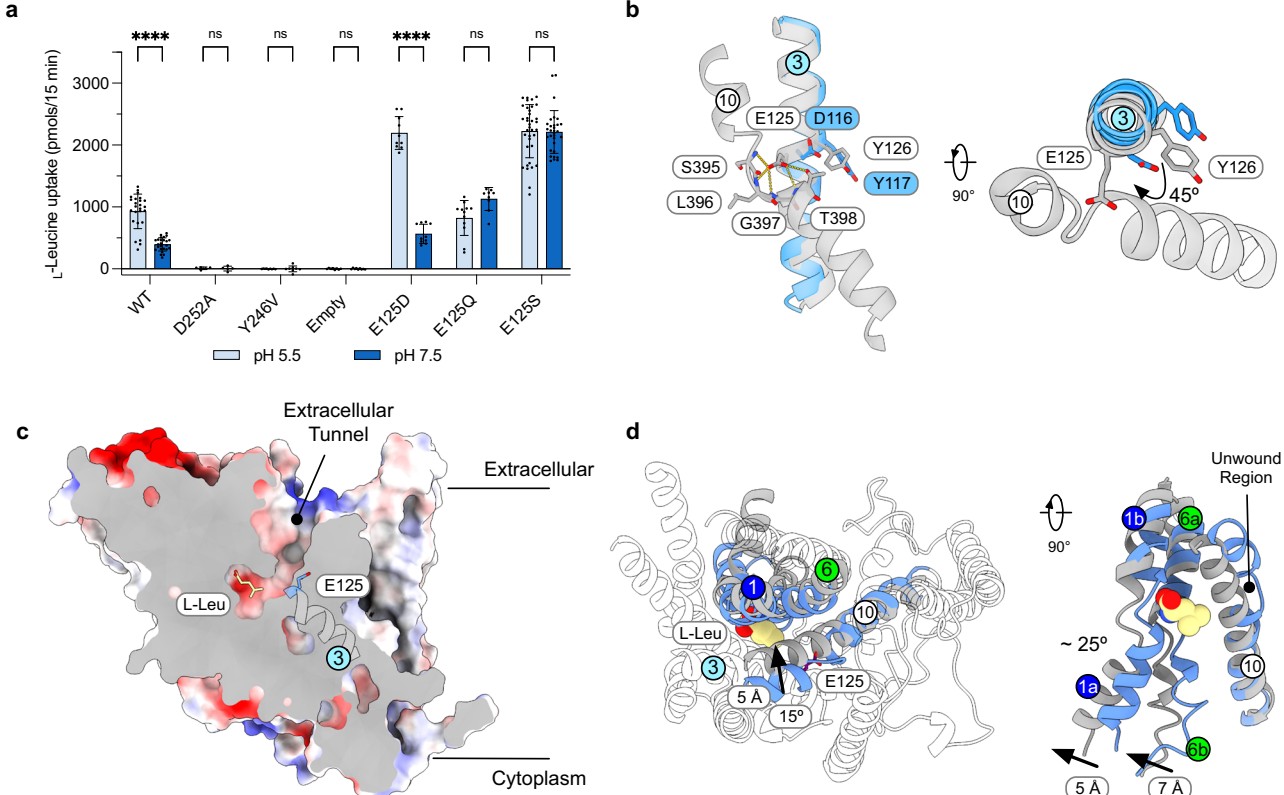

**Fig. 5 | Proposed mechanism of pH sensing in human SLC7A4. a** Cell-based uptake assays testing the effect of external pH on SLC7A4. Empty refers to the empty plasmid control. The measure of centre for the error bars is the mean. For pH 5.5, $n = 24$ for WT; five for D252A; nine for Y246V; ten for E125D; nine for E125Q; thirty six for E125S; seven for empty. For pH 7.5, $n = 25$ for WT; five for D252A; eight for Y246V; ten for E125D; twelve for E125Q; thirty for E125S; nine for empty. The errors shown are s.d. Statistical significance comparisons were calculated between pH 5.5 and 7.5 within each mutant via two-way ANOVA with a two-sided $\alpha = 0.05$.

**b** Structural superimposition of the cryo-EM structure of AtCAT4 (blue) and the homology model of SLC7A4 (grey). **c** Electrostatic surface representation of the homology model of SLC7A4 bound to L-Leu in the outward-facing state. The positions of E125 and TM3 are indicated. **d** Structural superimposition of the homology model of SLC7A4 (blue) in the outward open state with the cryo-EM structure of apo SLC7A1 from *Mus musculus* (MmCAT1; PDB: 9FQT) in the inward open state. Key helix movements are indicated. L-Leu is shown in space fill. Source data are provided as a Source Data file.

discontinuous region in TM10. However, under external acidic conditions, Glu125 would take up a proton, disrupting the interaction observed with the discontinuous region in TM10. It is plausible to consider that under these conditions, and in the presence of bound L-Leu, TM3 could then rotate to the position observed in the AtCAT4 cryo-EM structure and trigger the conformational changes that close the extracellular gate and transition the transporter into the inward open conformation observed for MmCAT1 (PDB: 9FQT)[48] via an intermediate occluded state.

A key structural change within the SLC7 family is the movement of TM10 towards TMs 1a and 6b to close the extracellular side of the transporter during the transport mechanism[22,24,71]. Indeed, if we compare our model of SLC7A4 in the outward open state with SLC7A1, in the inward open state (PDB: 9FQT), we observe that TM10 rotates towards TMs 1 and 6 by ~15° and moves inwards by ~5 Å (Fig. 5d). This is accompanied by a similar movement in TMs 13 and 14. At the opposite end of the binding pocket, TM1a rotates by ~25° and TM6b by ~23°, opening the central binding site to the cytoplasm by ~5–7 Å. In the inward open conformation in SLC7A1, TM10 adopts a straight conformation, with no unwound region. Our data thus provide further support for a mechanism, observed originally in SLC7A11 by us and subsequently in SLC7A5, where the transmembrane helices within the SLC7 family can undergo drastic structural transitions following ligand binding and transport[22,29,56]. Our data support a general role for TM1 and TM10 in ligand-induced conformational changes within the binding of SLC7A4, which may have important implications for developing targeted inhibitors for these transporters[22]. Our model also explains

why substituting Glu125 with shorter side chains would increase transport (Fig. 5a); essentially, the shorter side chains stabilise the conformation of TM10 to a lesser extent (Fig. 5b), potentially enabling SLC7A4 to undergo faster transport cycles. Supporting this hypothesis, we note that Glu125Gln, which is similar in length to glutamate, displayed transport activity comparable to WT but lacked pH dependence. A similar mechanism was proposed to explain why certain drugs are transported faster by SLC7A5 (LAT1); smaller drugs create a weaker steric hindrance to the movement of TM10[22].

## Discussion

Our data allow us to propose a preliminary model for L-Leu transport via SLC7A4 and to suggest how low extracellular pH could regulate its function (Fig. 6).

In the outward-facing state, the canonical APC fold binding site is exposed to the outside of the cell (state i). In this apo state, TM1 adopts a fully helical conformation and Glu125 makes extensive interactions with the backbone amides of the unwound region in TM10. A continuous solvent path connects Glu125 to the extracellular space through the tunnel. In a neutral environment, Glu125 would stay deprotonated and the interaction between TM3 and TM10 would be stabilised. Additionally, we expect EL4 to adopt the flipped-out state we observed in the AtCAT4 cryo-EM structure, maximising the solvent-exposed surface area of the transporter's entrance pathway. At neutral pH, L-Leu could access and bind the transporter to initiate transport, albeit with less efficiency.

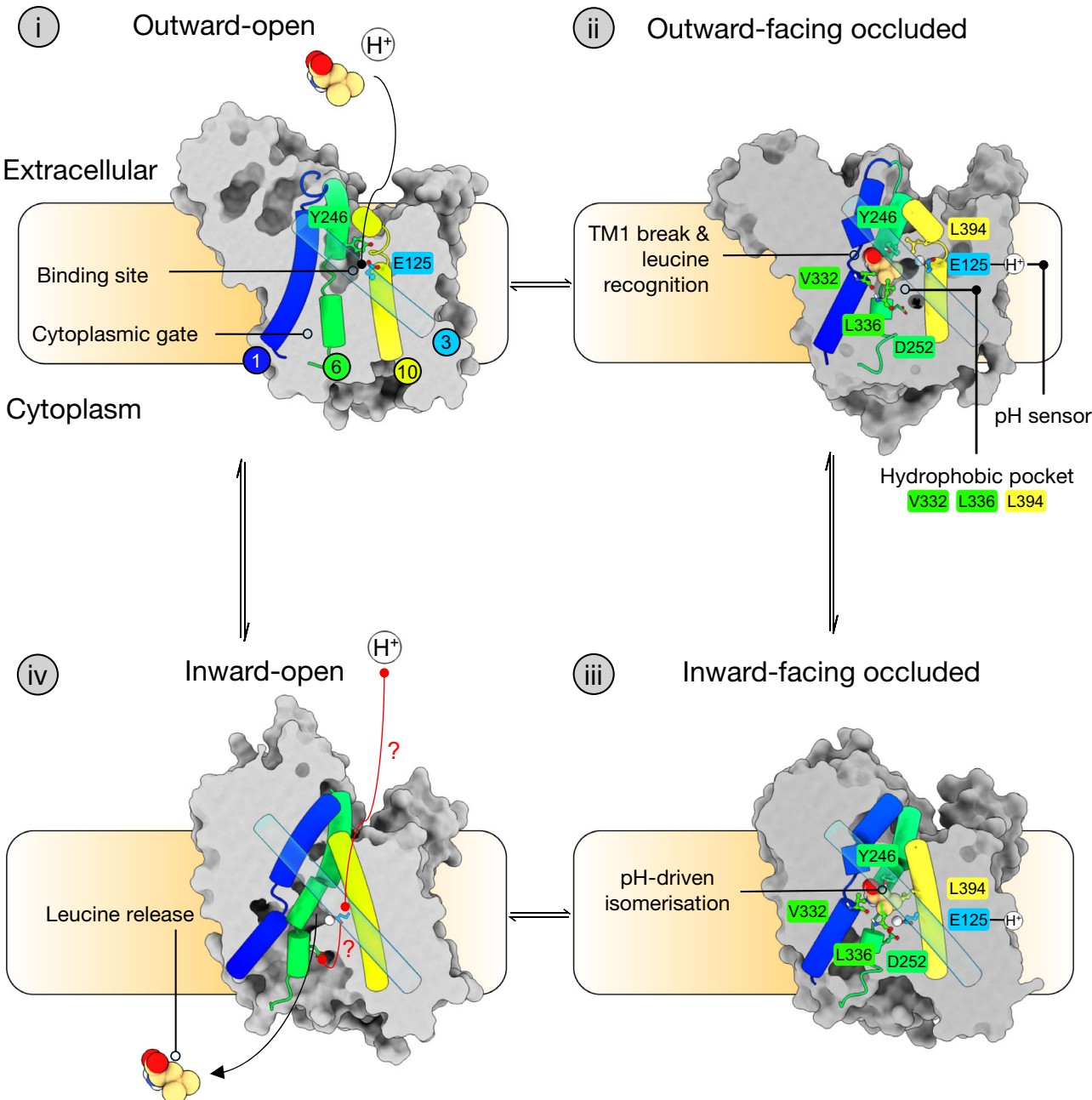

**Fig. 6 | Mechanism of Amino Acid Transport via SLC7A4.** Model for alternating access transport via SLC7A4 and the role of extracellular pH in regulating transport. (i) TM1 adopts a fully helical conformation to allow L-Leu to bind, and a continuous solvent path connects Glu125 to the extracellular space through the tunnel. (ii) TM6a rotates, placing Tyr246 over the L-Leu in the binding site, while Phe245 closes the solvent-accessible tunnel connecting Glu125 to the extracellular space. (iii) TM10 transitions into a continuous alpha helix, pushing the protonated Glu125 to face Asp252, stabilising TM6b. (iv) The cytoplasmic ends of TMs 1 and 6 would be free to move away from the hash domain, allowing L-Leu to exit the transporter. The proton is proposed to remain bound to Glu125 in this mechanism and exchange with the extracellular solution.

Our data support that SLC7A4 is pH-activated and displays higher transport efficiency at acidic pH. When the extracellular pH drops, Glu125 takes up a proton from the extracellular side of the membrane, weakening the interaction we observe with TM10. Concurrently, L-Leu binding triggers TM1 to adopt the canonical discontinuous state, which we also observed in the Orn-bound AtCAT4 cryo-EM structure. This transition may then be accompanied by the flipping-in of EL4, packed into the canonical position against TM1, as observed in other members of the APC superfamily. The unwinding of TM1 exposes the backbone amide group and side chain of Ser53 and the backbone amide group of Gly54 in the GSG[52] motif, which coordinate the carboxylate of L-Leu. The L-Leu amino group is coordinated through interactions with the backbone carbonyl groups of Gly49 and Met50. Additional interactions between the amino group of L-Leu are made to the backbone carbonyl group of Tyr246, which functions as the extracellular gate in SLC7A4 and Val249, which forms part of the reciprocal unwound region in TM6. The isobutyl side chain of the ligand fits into the hydrophobic pocket formed by Val332, Leu336, and Leu394 on TM8 and TM10. These conformational changes collectively transition SLC7A4 to the outward-facing occluded state (state ii).

For SLC7A4 to isomerise from the outward-facing occluded state to the inward-facing occluded state, TM6a rotates, placing Tyr246 over the L-Leu in the binding site, while also positioning the preceding side chain, Phe245, to close the solvent-accessible tunnel connecting Glu125 to the extracellular space. Recent structures of MmCAT1 (PDB: 9FQW/9FQT, templates for SLC7A4 homology models presented in Fig. 6iii-iv, respectively) also suggest that TM10 must transition into a continuous alpha helix, pushing the protonated Glu125 to face Asp252 (state iii), as we observed in GkApcT[35]. After reaching the inward occluded state, the cytoplasmic ends of TMs 1 and 6 would be free to move away from the hash domain (state iv), and L-Leu could then exit the transporter.

Currently, there is no evidence to suggest that human SLC7A4 is proton-coupled or that protons leave the transporter with the ligand, only that Glu125 is the pH sensor. Further kinetic studies in liposomes or oocytes are needed to fully clarify this mechanism. Nonetheless, the most probable explanation is that, under acidic conditions, Glu125 remains protonated, rendering the transporter hyperactive relative to its deprotonated state because Glu125 cannot reengage TM10. An important point to consider in our model is that SLC7A4 can still function at neutral pH, even though we predict that Glu125 will be deprotonated. The order of proton and amino acid binding is therefore another detail that will need to be studied, and whether, indeed, protonation of Glu125 impacts the affinity of SLC7A4 for L-Leu rather than facilitating the conformational cycle of the transport mechanism.

More broadly, our study uncovers a plasma membrane leucine transporter within the SLC7 family, with the distinctive ability to respond to changes in extracellular pH. Gene expression data indicate SLC7A4 enrichment in the choroid plexus (blood–brain barrier)[72] and testis[73], with secondary expression reported in placental trophoblasts[74]. These tissues share a common physiological property: they function as highly regulated nutrient-exchange interfaces, where acid–base handling plays an integral role[75,76]. In this physiological setting, a pH-gated leucine transport system could be advantageous as it would harness local acidity changes as an activation mechanism that promotes cellular leucine uptake.

## Methods

### Cloning, expression, and purification of *Arabidopsis thaliana* SLC7A4

The gene encoding *Arabidopsis thaliana* CAT4 (Uniprot code Q8W4K3) was codon optimised, synthesised (Geneart) and cloned into pDDGFP-Leu2D[77] (AddGene ID: 58352), carrying a C-terminal TEV protease cleavage sequence (ENLYFQG/S), yeast-enhanced green fluorescent protein (yeGFP) and a $His_{12}$ tag for purification. Variant forms of AtCAT4 were created by site-directed mutagenesis using the overlap extension PCR method. *S. cerevisiae* BJ5460 cells (MATa ura3-52 trp1 lys2-801 leu2Δ1 his3Δ200 pep4::HIS3 prb1Δ1.6 R can1 GAL) were transformed with sequence-verified AtCAT4-pDDGFP2-Leu2D plasmid using the lithium acetate method and plated onto auxotrophic media (Uracil and Leucine drop-out to increase plasmid copy number) sequentially until 70-80% saturation of plate surface with transformants. The yeast were seeded in $2 \times 100$ ml overnight cultures in synthetic-complete Leucine drop-out media (SC-LEU; 1.92 g/l; prepared in-house with L-α-amino acids) supplemented with 2% glucose and yeast nitrogen base (YNB; 6.9 g/l) for 20–22 h or until $OD_{600nm} = 3.0$, after which cultures were upscaled to $2 \times 700$ ml in the same media, grown for another 24 h to an $OD_{600nm}$ of 5.0. Yeast was grown at 30 °C at 240 rpm in flasks. 1.2 L of overnight culture was inoculated into 9.6 L SC-LEU + YNB media with 2% lactate (aliquoted over 12 baffled Erlenmeyer flasks). The cultures were grown in lactate media for 17 h until $OD_{600nm}$ 2.0–2.5 and were induced with galactose (1.5% final) to express the protein. Cultures were harvested 18–20 h after induction, at an $OD_{600nm}$ routinely ranging between 4.0 and 4.5, depending on the preparation. The expression level was

quantified by whole-cell GFP fluorescence (λexcitation = 488 nm, λemission = 512 nm) from 8 mL of culture resuspended in 150 µL of PBS, measured using a SpectraMax M3 Microplate Reader (Molecular Devices). Cells were harvested at $5000 \times g$ for 25 min, and the pellet was resuspended in PBS pH 7.4 (5 ml·g⁻¹ total volume) and stored at -80 °C until further use.

To isolate the membrane fraction, cells were lysed using a TS Series Continuous Cell Disruptor (Constant Systems) operated at 38 kpsi, with a NESLAB ThermoFlex 1400 Chiller (Thermo Fisher Scientific) maintaining a temperature of 4 °C. The lysate was clarified by centrifugation at $30,000 \times g$ for 30 min, and the supernatant was then ultracentrifuged at 235,872 g for 1.5 h to pellet the membranes. The pellet was washed in a high-salt buffer consisting of 20 mM HEPES-NaOH, pH 7.5, and 1.0 M potassium acetate, and Dounce-homogenised. The suspension was ultracentrifuged at $235,872 \times g$ for 1 h, and the washed membrane pellet was resuspended in PBS (5 ml/g total volume).

AtCAT4 membrane fractions were solubilised in 1% (w/v) detergent (LMNG; Anatrace), buffered with PBS pH 7.4, and supplemented with an additional 150 mM NaCl and 10% (v/v) glycerol for 1.5 h. The insoluble material was separated by ultracentrifugation at $235,872 \times g$ for one hour. The solubilised fraction was supplemented with 20 mM imidazole. It was then incubated with HisPur Ni-NTA resin (Thermo Fisher Scientific) until the GFP-tagged protein bound to the resin plateaued, as monitored by GFP fluorescence, for up to 3.5 h. Purification proceeded via standard IMAC steps. Specifically, the resin was washed in a gravity column with PBS supplemented with 150 mM NaCl, 0.01% (w/v) LMNG, 10% (v/v) glycerol (buffer A) with 20 mM imidazole, for 20 column volumes (CVs), followed by a second wash step with 25 mM imidazole (10 CVs). Membrane protein was then eluted with buffer A supplemented with 250 mM imidazole in 4 CVs. The concentration of the eluted GFP fusion protein was estimated as described in ref. 78. An equimolar amount of Tobacco Etch Virus (TEV) protease was added to the eluant and dialysed overnight at 4 °C against 4.0 L of buffer B (20 mM Tris·HCl, pH 7.5, 150 mM NaCl, and 0.003% LMNG) using a 3.5 kDa MWCO regenerated cellulose membrane (Spectra-Por tubing). The next morning, the dialysate was syringe-filtered through a 0.22 µm filter supplemented with 10 mM imidazole and passed through a 5 ml HisTrap column (Cytiva; pre-equilibrated in gel-filtration buffer) to remove the His-tagged TEV protease, liberating GFP-His8 with Ni-affinity. The flow-through was collected and spin-concentrated in a 100,000 MWCO concentrator (Vivaspin 20, Sartorius) to 0.4 mL and injected onto a Superdex 200 Increase 10/300 GL SEC column (Cytiva; pre-equilibrated with buffer B) and run at a flow rate of 0.4 mL/min. Fractions containing protein were analysed with SDS-PAGE for their purity, pooled accordingly and concentrated in a 50,000 MWCO concentrator (Vivaspin 2, Sartorius or Amicon® Ultra) to a concentration of 32.4 mg·ml⁻¹. Protein concentration was determined with a NanoDrop ND-1000 spectrophotometer using extinction coefficients calculated with the Expasy ProtParam server for the purified protein (disulphides considered as reduced). C-terminally Avi-tagged (N-GLNDIFEAQKIEWHE-C) protein was purified as above to select binders against in the sybody pipeline. AtCAT4 biotin modification was carried out using the BirA ligase according to the manufacturer's instructions (Avidity). All purification and protein handling steps were performed at 4 °C.

### Cloning, expression, and purification of Human SLC7A4

The gene encoding HsSLC7A4-FLAG-stop (UniProt ID O43246) was synthesised (GeneArt) and cloned into pLexM with a C-terminal FLAG tag (AddGene ID: 167988). Variant forms of human SLC7A4 were created by site-directed mutagenesis of the gene in the pFAW vector using overlap PCR and then cloned into pLexM with a C-terminal FLAG tag. The plasmid DNA was syringe-filtered (0.22 µm) and stored at −20 °C, prior to transfection. Human embryonic kidney−293 F (HEK293F) cells

in FreeStyle™ 293 Expression Medium (Thermo Fisher Scientific) were used for transient transfection and expression. The cells were cultured in suspension phase at 37 °C and 8% $CO_2$. Cells were split to $0.7 \times 10^6$ viable cells/mL 24 hours before transfection. 24 h later, the cells reached a density of $1.3 - 1.5 \times 10^6$ cells/ml, at which point they were transiently transfected. 1.0 L of cells was transfected with 1.1 mg plasmid DNA (diluted in serum-free DMEM; Thermo Fisher Scientific) complexed with linear polyethyleneimine (PEI) MAX (MW 40,000; Polysciences Inc., USA) at a 1:2 (w/w) ratio. Immediately after transfection, syringe-filtered (0.22 μm) sodium butyrate was added to a final concentration of 8.0 mM. Cells were returned to the incubator and harvested 40 hours after transfection. They were then washed in PBS (pH 7.4), pelleted, and stored at −80 °C until use. Routinely, the cell biomass yield was 8 g/l culture.

The cell pellet was thawed and resuspended in cold PBS (3.125 ml·g$^{-1}$ total volume) containing deoxyribonuclease and two tablets of protease inhibitors per l worth of cells (SIGMA*FAST*™; each containing: 2 mM AEBSF, 0.3 μM Aprotinin, 130 μM Bestatin, 1.0 mM EDTA, 14 μM E-64, 1.0 μM Leupeptin). Cells were subsequently lysed on ice using sonication (Qsonica, USA), with an amplitude of 30% and a total sonication time of 2 min/l, consisting of 20 pulse cycles (2 s ON, followed by 4 s OFF). Unbroken cells and cell debris were pelleted at $12,000 \times g$ for 10 min at 4 °C, and the supernatant was ultracentrifuged at $235,872 \times g$ for 1 hour, washed with 15 mM HEPES·NaOH, pH 7.5 and 20 mM KCl and Dounce homogenised. The resuspended membranes were ultracentrifuged at $235,872 \times g$ for 1 hour, and the washed membranes were resuspended in PBS (10 ml·g$^{-1}$ final volume) and snap-frozen for storage at −80 °C until purification. On the purification day, the membranes were solubilised in ice-cold PBS pH 7.4 supplemented with 150 mM NaCl, 10% (v/v) glycerol, and 1% (w/v) DDM – 0.2% (w/v) CHS for 1.5 h. The insoluble fraction was pelleted by ultracentrifugation at $235,872 \times g$ for one hour. The supernatant was incubated in batch with 0.1 mL/L Pierce Anti-DYKDDDDK affinity resin (Thermo Fisher Scientific) for 2 hours and subjected to gravity flow. The flow-through was collected and incubated with 0.07 ml/l culture Anti-DYKDDDDK resin for another 1.0–1.5 h to collect all protein. The detergent was exchanged to GDN during gravity chromatography. Specifically, the two resin pools were separately washed with 30 CV of buffer A (PBS, pH 7.4, 150 mM NaCl, 10% glycerol, 0.1% (w/v) GDN) supplemented with 10 mM ATP/20 mM $MgCl_2$, followed by a 10 CV wash step in buffer A without ATP/$MgCl_2$. The protein was eluted from each resin pool after 0.5–1.0 h incubation with 2.3 and 1.3 CV 0.5 mg/ml 3′FLAG peptide (MDYKDHDGDYKDHDIDYKDDDDK) in buffer A. The two elution steps from each resin pool were combined, concentrated with a 50,000 MWCO concentrator (Vivaspin 20, Sartorius) to 0.4–0.5 ml and injected on a Superdex 200 Increase 10/300 GL SEC column (Cytiva; pre-equilibrated with 20 mM Tris pH 7.5, 150 mM NaCl, 0.01% (w/v) GDN). SLC7A4 fractions were collected without further concentration to avoid concentrating the detergent. Pooled gel filtration fractions were routinely 0.5–1.0 mg/ml (varied across preparations) and suitable for downstream applications.

### Thermal stability measurements

Differential scanning fluorimetry (DSF) measurements were performed with a NanoTemper Prometheus NT.48. Wildtype human SLC7A4 (purified in Glyco-diosgenin (GDN)), AtCAT4 (purified in LMNG) and mutants were investigated on their thermal stability in the presence and absence of 1 mM α-amino acids (from 10 mM stocks in water, filtered and stored at 4 °C). Protein was added at final concentrations of 0.5 mg·ml$^{-1}$ (human construct) and 1.0 mg·ml$^{-1}$ (*Arabidopsis thaliana* construct) and the pH was buffered (a) pH 7.5 with 20 mM Tris·HCl, or (b) pH 5.5 with 20 mM citrate. Other buffer components were 150 mM NaCl and 3× CMC of the selected detergent. AtCAT4, samples were supplemented with lipid (soybean polar extract: cholesterol, 9:1 w/w; Avanti Polar Lipids) prior to measurement

to observe ligand-induced stabilisation. The lipid stock (20 mg·ml$^{-1}$ in 20 mM Tris·HCl) was first disrupted with 15 mM sodium cholate until translucent to emulsify lipid vesicles and then incubated with the protein sample at a 1:1.2 (w/w) protein: lipid ratio for 15-30 min at room temperature before DSF measurements. 12 μl of each sample was loaded onto standard sensitivity capillaries (NanoTemper) and transferred to Prometheus NT.48 (NanoTemper). NanoDSF experiments were run at 30% (*Arabidopsis thaliana* construct) and 50% (human construct) intensity using 1 °C min$^{-1}$ increments in a range from 20 to 95 °C. The resulting melting curves were generated by plotting the first derivative of the fluorescence ratio at 330 nm/350 nm against temperature. The melting temperature ($T_m$) of the protein in the presence of the ligand was subtracted from that of the protein in the absence of the ligand to calculate the $\Delta T_m$ of the protein induced by ligand binding.

### Synthetic nanobody (Sybody) selection and purification

Sybody selection was performed against C-terminally Avi-tagged and biotinylated AtCAT4. The protocols for sybody selection have previously been described in detail[79]. Five specific anti-AtCAT4 binders were identified from the loop sybody library. The binders were assessed by means of Ni-affinity pulldowns using His$_6$-tagged binders (expressed from the pSb_init vector), followed by SDS-PAGE densitometry and ranked using biolayer interferometry. Biolayer interferometry was performed on an Octet Red 384 (Sartorius) to calculate $K_D$. A high-affinity sybody (57.3 ± 10.5 nM), SybB5, was identified as the higher-affinity sybody, which formed a stable complex with the transporter and co-eluted from a Sepax SRT-C SEC-300 column (Chromex). The identified SybB5 was expressed as C-terminal Myc-His$_6$-tagged binders using periplasmic extraction. Specifically, binder-transformed *E. coli* (MC1061; Lucigen) were grown in Terrific Broth with 100 μg·ml$^{-1}$ chloramphenicol until an optical density at 600 nm (OD$_{600}$) of 0.5 was reached, at which point the temperature was reduced to 22 °C and cultivation continued for an additional 120 min. Binder expression was induced with 0.02% $_L$-(+)-arabinose and left overnight at 22 °C and 180 rpm. Cells were harvested and immediately incubated in 50 ml/l culture of periplasmic extraction buffer (20% w/v sucrose, 50 mM Tris·HCl, pH 8.0, 0.5 mM EDTA and 0.5 μg/ml lysozyme, prepared fresh and used immediately) at 4 °C for 30 min with magnetic stirring. The suspension was diluted with 200 ml/L of 20 mM Tris·HCl (pH 7.5) containing 1 mM $MgCl_2$, incubated on ice for an additional 30 min, and then clarified by centrifugation at $4000 \times g$ for 30 min at 4 °C to obtain the periplasmic extract from the cell pellet. The periplasmic extract was supplemented with 15 mM imidazole and an additional 150 mM NaCl and incubated with HisPur Ni-NTA resin (Thermo Fisher Scientific) for one hour. The resin was washed with 20 CVs of 30 mM imidazole, 20 mM Tris·HCl, pH 7.5, 300 mM NaCl, before eluting the Myc-His$_6$-tagged binder with 300 mM imidazole. The eluant was dialysed overnight against 4 L of 20 mM Tris·HCl, pH 7.5, 150 mM NaCl in a 3.5 kDa MWCO regenerated cellulose membrane (Spectra-Por tubing). Next morning, the dialysate was syringe-filtered (0.22 μm) to remove any visible precipitated Sybody and the filtrate was spin-concentrated to 1.0 ml with 5000 MWCO concentrators (VivaSpin 20).

### Cryo-EM sample preparation and data acquisition

AtCAT4-(63.638 kDa)−SybB5 (15723 kDa) complexes were formed on the day of grid preparation by incubation of 0.3 mg·ml$^{-1}$ purified AtCAT4 purified in LMNG (from a 32.4 mg·ml$^{-1}$ stock) with 1:2 molar excess of the Myc-His$_6$-tagged SybB5 for 60 min on ice in a 400 μl reaction volume buffered with 20 mM Tris·HCl pH 7.5, 150 mM NaCl and supplemented with 0.003% LMNG. The formed 1:1 complex was separated from excess sybody by gel filtration, injecting the reaction into a Sepax SRT-3C SEC300 (Cytiva) column at 0.4 ml·min$^{-1}$. The pooled fractions were analysed by SDS-PAGE for complex formation

before concentrating the sample in a 50 MWCO centrifugal concentrator (Amicon® Ultra) at $4000 \times g$ and 4 °C until the desired complex concentration was achieved, as assessed spectrophotometrically using a NanoDrop ND-1000 spectrophotometer. The extinction coefficient was estimated using the AtCAT4 sequence (cleaved at the TEV site) and the corresponding Myc-His6-tagged sybody with the ExPASy ProtParam server. The final complex concentration was determined to be 3.4 mg·ml⁻¹, and 3.5 µl of the sample was pipetted onto the grid for a blot time of 3.0 s. For the apo AtCAT4 without SybB5, the LMNG-purified protein was plunge-frozen at 4.7 mg·ml⁻¹. For the SybB5+L-Orn-bound dataset, the preparation was similar, with minor modifications. Specifically, the AtCAT4—SybB5 complex was relipidated 1:1 w/w AtCAT4:lipid with soy PC: cholesterol mix (9:1 w/w, from 20 mg·ml⁻¹ stock in SEC buffer), followed by supplementation with 10 mM L-Orn (from 100 mM stock in water) before grid freezing 2.5 µl sample with blot time 1.5 s. The final complex concentration was determined at 5.0 mg·ml⁻¹.

The grid preparation was performed at 100% humidity and room temperature. A Vitrobot Mark IV (Thermo Fisher Scientific) was used to prepare the grid samples. 2.5–3.5 µl (see above for specific sample conditions) of sample was dispensed onto glow-discharged holey carbon-coated grids (Quantifoil 300 mesh, Cu R1.2/1.3, Agar Scientific). Grids were blotted for 1.5–3.0 s (Vitrobot parameters: blot force of -5, 100% humidity, 4 °C) and then plunge-frozen into liquid ethane. Grid samples were screened, without dataset collection, on a Talos Arctica 200 kV Cryo-TEM (Thermo Fisher Scientific) equipped with a Falcon 3 Direct Electron Detector (Thermo Fisher Scientific). For data collection, micrographs were collected in counted super-resolution mode on a Titan Krios G3 (FEI) operating at 300 kV with a BioQuantum imaging filter (Gatan) and K3 direct detection camera (Gatan) at 105,000× magnification, physical pixel size of 0.832 Å spanning a defocus range −2.0 to −1.0 µm, and beam-image shift acquisition. For the apo+SybB5 dataset, the total dose rate was 19.3 e⁻/Å², exposure time 3.00 s, corresponding to a total dose of 58 e⁻/Å² spread over 40 frames. The collection yielded a total of 17,271 movies, including 11,166,793 particles extracted at 256 × 256 pixel boxes. For the L-Orn+SybB5 dataset, the total dose rate was 25.0 e⁻/Å², exposure time 2.00 s, corresponding to a total dose of 50 e⁻/Å² spread over 50 frames, yielding a total of 13,942 movies, including 17,133,106 particles extracted at 288 × 288 pixel boxes. For the apo-SybB5 dataset, the total dose rate was 15.3 e⁻/Å², exposure time 2.60 s, corresponding to a total dose 39.9 e⁻/Å² spread over 50 frames, yielding a total of 30,511 movies including 8,226,459 particles extracted at 300 × 300 pixel boxes.

## Cryo-EM data processing

*Apo+SybB5 dataset*: The dataset of 17,271 movies was processed on-the-fly in SIMPLE 3.0[80] for patched (15×10) motion correction, dose weighting, a patched contrast transfer function (CTF) estimate, and particle picking. 11,166,793 particles were picked in real time with SIMPLE from collected micrographs. All subsequent processing was performed in either cryoSPARC[81] or RELION 3.1[82] interconverting between the two formats using the csparc2star.py script within UCSF pyem v0.5 (https://doi.org/10.5281/zenodo.3576629). Two rounds of reference-free 2D classification were performed. In the 1ˢᵗ round (350 classes), all proteinaceous densities were selected (7,053,972 particles, 63.2% remaining). These were passed to a 2ⁿᵈ round with stricter selection criteria, i.e., AtCAT4-SybB5 imitating densities displaying characteristic Syb protrusions and helical densities (2,459,086 particles, 22.0% of total particles remaining). Most classes showed a top/bottom view of the complex (~70%), whereas the remaining were side views. The 2D class averages (2D-CAVG) calculated in cryoSPARC revealed discernible structural elements, the most pronounced being the protruding density of the putative Syb binder and α-helical-like signals in some 2D-CAVGs. Multi-class ($N = 6$) ab-initio models were generated with particles from the 2ⁿᵈ 2D-CAVG round, one of which

gave an apparent Syb-bound transporter-like architecture with discontinuous helical densities (509,029 particles, 4.6% remaining). The class particles were submitted to a further "high-resolution" ab-initio round to regenerate the model. Further in silico purification of the dataset disposed of compositional heterogeneity using decoy "junk" maps. Specifically, the classes corresponding to "junk" particles in the dataset ($N = 5$, from the 1ˢᵗ ab-initio) were used as a reference along with the protein-target maps (from the 2ⁿᵈ ab-initio) to guide separation in distinct classes by cryoSPARC heterogeneous refinement ($K = 5$), low-pass-filtered to 10 Å. This provided a clear AtCAT4—SybB5 class, without broken densities and with 14 intact TM helices. Subsequently, 2D-CAVG (927,049 particles, 8.3% remaining) were CTF-refined, followed by non-uniform refinement and, finally, masked-local refinement, yielding a reconstruction at a global FSC resolution of 3.8 Å. The particles were Bayesian-polished and classified in 2D to generate a subset of 642,385 cleaned and polished particles, followed by homogeneous refinement, non-uniform refinement, and, finally, masked-local refinement, yielding a final reconstruction at 3.27 Å (GS-FSC 0.143), with the local resolution varying between 2.8 and 4.4 Å. Both Bayesian polishing (per-particle, reference-based beam-induced motion correction) and CTF refinement (per-particle defocus, magnification, and higher-order aberration estimation) improved the resolution of the AtCAT4—SybB5 map. The final reconstruction showed a "tight" LMNG micelle encapsulating the AtCAT4—SybB5 complex, likely rigidifying it into the presented conformation. Resolution estimates were derived from gold-standard Fourier shell correlations (FSCs) using the 0.143 criterion as calculated within cryoSPARC. Local resolution estimations were calculated within cryoSPARC.

*L-Orn+SybB5 dataset*: 13,942 movies were processed entirely in cryoSPARC v4.4 using patch motion correction and patch CTF estimation, followed by template-based picking with circular templates (d = 140 Å; reference = apo+SybB5 AtCAT4 map), yielding 5,444,589 particle images. To enrich for AtCAT4—SybB5 views, we performed three rounds of heterogeneous refinement (C1) using a transporter reference map imported from the apo+SybB5 dataset together with "junk" decoy maps to steer separation; this retained a clean class of 693,677 particles (12.7%), which non-uniformly refined to 3.19 Å. However, L-ornithine density was not clearly observed, requiring further classification. On this particle stack, a 2D cleanup was performed (N = 100), followed by ab initio reconstruction (K = 2) and non-uniform refinement of 376,110 particles to 2.86 Å. To isolate the ligand-bound subset, we then performed binding–site–focused 3D classification without alignment (K = 8) using a binding-site mask (defined to include residues within 10 Å of the canonical APC superfamily binding site) as the solvent mask. The 3D classification parameters were: hard classification ON; class similarity = 0.01; online-EM batch size = 10,000; online-EM learning rate = 1.0. This processing pipeline yielded a class with a clear density difference between TM1 and TM6 at the canonical APC binding site (44,789 particles). This particle stack was further refined with non-uniform refinement, reference-based CTF refinement, non-uniform refinement and reference-based motion correction (RBMC) to produce a ligand-positive reconstruction at 3.34 Å (GS-FSC 0.143). Continuing from the initial ligand-positive class (47,054 particles), we expanded the ligand-bound stack by (i) performing non-uniform refinement, then (ii) two successive rounds of binding-site–focused seeded 3D classification without alignment (K = 2) with duplicate removal (yielding 134,784 particles), followed by (iii) non-uniform refinement and an additional 2D cleanup (N = 50; force max over shift/poses OFF; selected 107,868 particles). We then carried out a further non-uniform refinement and a final TM-domain–focused 3D classification without alignment (K = 2) to remove classes with degraded TM density. Ligand-positive classes were pooled, duplicates removed, and non-uniformly refined, producing the final L-Orn–bound reconstruction (76,191 particles; 3.32 Å). FSC and local-resolution estimates were computed in cryoSPARC.

*Focused 3D classification of the apo + SybB5 dataset as a negative control for ligand-density selection:* To assess whether our binding-site–focused 3D classification procedure could generate spurious ligand-like density, we applied the same strategy used for the L-Orn +SybB5 dataset to the apo+SybB5 dataset (no added ligand), using the same aligned focus mask and identical classification settings (K = 8; filter resolution = 2 Å; online-EM batch size per class = 10,000; learning rate = 1; initialisation = PCA; class similarity = 0.1; force hard classification = true). Classes were compared after identical normalisation and low-pass filtering to 4.0 Å (class07 at 4.5 Å) and were contoured at a common level. Normalisation enables comparison of maps at identical contour values, and low-pass filtering standardises map sharpness across classes with different nominal resolutions for direct visual comparison. No class, including the most feature-rich and interpretable class, showed ligand-shaped density in the canonical substrate pocket (Supplementary Fig. 12) comparable to the density observed for L-Orn in Fig. 1d and Supplementary Fig. 2b-c.

For comparisons between maps from different datasets, the maps were normalised to zero mean and unit variance and contoured at identical σ values. This can be achieved in ChimeraX with *measure mapStats #1* followed by *volume scale #1 shift −'mean' factor '1/stdev'*. Maps focused on the binding-site (reported in Supplementary Fig. 12) were multiplied by the focus mask using the command *volume multiply #1 #2*.

*Apo (−SybB5) dataset*: 30,514 movies were processed in cryoSPARC v4.4 using patch motion correction and patch CTF estimation. For particle picking, we first applied a blob picker to 1,000 representative micrographs, classified those particles in 2D to generate templates, and then ran the template picker, yielding 8,226,459 particles. Four rounds of reference-free 2D classification (N = 100) reduced the stack to 1,235,585 particles, which underwent two additional 2D rounds (N = 50; circular mask diameter = 150 Å) to 724,282 particles. A multi-class ab-initio (C1, K = 3) identified a transporter-like class (239,837 particles; 33%). Two further ab-initio rounds (K = 2) produced a high-quality seed subset (65,088 particles; 27%), which was refined by non-uniform refinement to 6.26 Å and was then used for seeded multi-class ab-initio searches against the raw pool. The raw particles were partitioned into non-overlapping batches and de-duplicated (20 × 62,667; 5 × 246,850; 4 × 291,438); each batch was processed with seeded multi-class ab-initio, followed by three rounds of ab-initio (K = 2) and finally non-uniform refinement, yielding reconstructions at 4.79 Å (84,178 particles), 4.82 Å (152,676), and 4.89 Å (239,513). After removing duplicates, the merged stack (323,978 particles) underwent non-uniform refinement followed by local refinement to 4.53 Å. A final 3D classification without alignment (K = 3) isolated a single well-resolved class (120,868 particles; 37%), which, after local refinement, produced the final apo reconstruction at 4.23 Å (GS-FSC = 0.143). All steps, including FSC and local-resolution estimation, were performed in cryoSPARC.

## Model building and refinement

The AlphaFold2 model of AtCAT4 (CAT4; Alphafold2 ID AF-Q8W4K3-v3) was manually rigid-body fitted to the map using ChimeraX. A model of SybB5 was generated using ColabFold[83] with default parameters and AMBER force field relaxation settings. The SybB5 model was then fitted to the map and merged with the AtCAT4 model. The apo and L-Orn-bound models of the AtCAT4–SybB5 complex (or AtCAT4 apo model without SybB5) were manually adjusted using Coot v0.9.8[84], and real-space refined using PHENIX[85]. Cholesterol (CLR) stereochemical parameters were generated with AceDRG[86], and L-Ornithine (ORN) and POPC (LBN) parameters with ELBOW/PHENIX. The atomistic structural models were manually readjusted in Coot to set rotamers in a second round, followed by global real-space refinement in PHENIX. The +SybB5 and +SybB5+L-Orn models were further MDFF-refined in ISOLDE/ChimeraX[87] using the AMBER ff14SB force field for protein and GAFF2/AM1-BCC for L-Orn and lipids, to relax the ligands' geometries into their respective densities, resolve steric clashes, remove strain, and model TM kinks on both density and electrostatics. The ISOLDE-polished structures were real-space-refined to obtain final validation statistics. The figures depicting the molecular structures were prepared using ChimeraX[88].

## Proteoliposome reconstitution

SLC7A4 was reconstituted into liposomes consisting of 90:5:15 (w/w) *E. coli* polar extract:cholesterol:brain polar extract using biobeads. It was essential not to overconcentrate SLC7A4/GDN after the final gel-filtration step to avoid excess detergent. The lipids (Avanti) were solubilized in chloroform, evaporated on a rotary evaporator, and washed with *n*-pentane twice before being resuspended in 50 mM $KH_2PO_4/K_2HPO_4$ buffer pH 7.0 to final concentration of 20 mg/ml. These lipid vesicles were frozen and thawed five times in liquid nitrogen and stored at −80 °C until required. For protein reconstitution, purified SLC7A4 was diluted to 0.25-0.5 mg/ml in reconstitution buffer (20 mM $KH_2PO_4/K_2HPO_4$ pH 7.0, 150 mM NaCl). A lipid vesicle amount corresponding to a 1:80 protein-lipid ratio was thawed, diluted in reconstitution buffer to 10 mg/ml, and subsequently extruded through 0.8 μm and 0.4 μm filters 11 times. The extruded vesicles were then titrated with DDM:CHS (5:1 w/w) until 66% of $A_{540nm}$ was achieved; routinely, 0.025% (w/v) final DDM concentration was required. The disturbed lipid vesicles were batch-wise added to the protein over 1.5 h at room temperature, followed by 1 h incubation on ice. 250 μl biobeads were added (for a 300 μg reconstitution) and incubated for 1 h at 4 °C under 10 rpm rotation, followed by an additional 333 μl for 3 h, and a final exchange to 333 μl fresh biobeads overnight. Next morning, biobeads were discarded and replaced with fresh to 333 μl biobeads for 1 h before harvesting the proteoliposomes at 200,000 × g, 4 °C, for 35 min, before resuspension in reconstitution buffer at a final protein concentration of 0.25 μg μl⁻¹. The proteoliposomes were freeze-thawed three times in liquid nitrogen before storage at −80 °C. The amount of SLC7A4 successfully reconstituted into the liposomes was quantified by SDS–PAGE densitometry after solubilizing the liposomes with 1.5% DDM:CHS (5:1). Successfully reconstituted protein was considered as the fraction of the total DDM-extracted protein that did not precipitate after subjected to 200,000 × g, 4 °C, for 35 min. Depending on the preparation, we found that 25–30% of SLC7A4 could be successfully reconstituted in liposomes and used this ratio to assign protein concentration to the proteoliposomes.

## Liposome-based Radioactive transport assay

To observe concentrative transport of L-leucine in SLC7A4 proteoliposomes, an acetate-induced pH gradient was generated[89]. Proteoliposomes were thawed and the required amount for the assay (5.0 μg per time point) was harvested through ultracentrifugation (200,000 × g, 4 °C, 30 min) before resuspending in inside buffer (20 mM $KH_2PO_4/K_2HPO_4$ pH 6.5, 100 mM KAc, 2 mM $MgSO_4$). The resuspended proteoliposomes were freeze-thawed five times to achieve buffer exchange and extruded via 0.4 μm filters 11 times, before being harvested again and resuspended in minimal amount of inside buffer, i.e., 25-fold less volume than transport reaction. To initiate transport, the proteoliposomes were 25-fold diluted in outside buffer (20 mM $NaH_2PO_4/Na_2HPO_4$ pH 6.5, 100 mM NaCl, 2 mM $MgSO_4$) supplemented with 100 μM L-Leu, and trace amounts of ¹⁴C-L-Leu [(stock: 295.86 μM 338 mCi/mmol), 2.96 μM final assay concentration, 1:100 hot:cold dilution]. The transport reaction was incubated on a temperature-controlled heat block at 25 °C with periodical gentle mixing. Transport reaction time-points were taken by immediately halting transport by rapidly filtering onto 0.22 μm filters, which were then washed with 2 × 2 ml ice-cold 0.1 M LiCl. The amount of L-Leu transported into the liposomes was calculated by scintillation

counting in Ultima Gold (PerkinElmer) and compared to a standard curve for the uptake mix.

## Adherent cell transfection for assays

HCT116-SLC7A5-KO cells (Resolute Cell line ID CE051H-1[62]) were maintained in RPMI 1640 GlutaMAX medium (Thermo Fisher Scientific; Catalogue number 61870036) supplemented with 10% foetal bovine serum under 5% CO2 at 37 °C. One day prior to transfection, $2.5 \times 10^5$ cells/ml were seeded into 12-well plates. On transfection day, the cells were monitored to be 65–70% confluent and had their media exchanged to 2% FBS media to allow transfection. 1 µg plasmid DNA (wild type SLC7A4 or mutants in pLexM, with a C-terminal FLAG tag) was complexed with 3 µg PEI-MAX in Opti-MEM Reduced Serum Medium and incubated for 5–10 min at room temperature. 100 µl of transfection mix per well was added dropwise to the cells and gently mixed. 16 h post-transfection, the media were changed to fresh media containing 10% FBS. Expression was allowed for 38-40 hours, after which the cells were used for transport assays, fluorescence microscopy, or expression analysis. Cells above 18 passages were discarded.

## Cell-based radioactive transport assay

Cells were washed twice with 1 ml of pre-warmed assay buffer. We used sodium-free buffers for the cell-based assay to reduce the background intrinsic transport activity of Na-dependent transport systems. The assay buffer composition was 10 mM HEPES × KOH, pH 7.5 or 10 mM MES × KOH, pH 5.5-6.5, 1 mM $MgSO_4$, 2 mM $CaCl_2$, 5 mM KCl, 137 mM choline chloride (as a sodium chloride substitute of equal ionic strength) and 10 mM D-glucose. The cells were gently washed with 0.3 ml/well assay buffer twice, before application of 0.3 ml of the assay buffer supplemented with 1 µM final $^{14}$C L-Leu concentration [(stock: 295.86 µM 338 mCi/mmol), 2.96 µM final assay concentration, 1:300 hot:cold dilution] and 10 µM cold L-Leu substrate (either L-Leu or any of the other amino acids, as indicated in the range from 0 µM to 1000 µM) onto the cells and incubated for 1, 3, 8, and 15 min at 37 °C. After the desired time, the radioactive assay buffer was removed, and the cells were quickly washed twice with 0.3 ml ice-cold assay buffer without substrate. The cells were then lysed with 0.05% Triton X-100 (w/v) in PBS, pH 7.4, for 5 min and placed in scintillation vials with 0.3 ml scintillation fluid (Ultima Gold). The amount of transported $^{14}$C L-Leu inside the cells was calculated by scintillation counting (Perkin Elmer) and compared to a standard curve prepared from known amounts of uptake mix collected on the day of the experiment. To assay the pH dependence of the SLC7A4 mutants, time points at 15 min were measured. Total protein content of the cells was quantified using the BCA protein assay (Pierce™ BCA Protein Assay Kits Catalogue Number A55864), and 2 µg of samples were used to check expression of wild-type SLC7A4 and mutants using a Western blot with an anti-FLAG antibody (Merck F1804) at 10,000x dilution and anti-β-actin antibody (Merck A2228).

## Statistical comparisons

Analyses were performed in GraphPad Prism v10 (GraphPad Software). Bars show mean ± s.d. with overlaid points for each independent measurement; $n$ denotes the number of independent measurements (technical replicates across biological replicates were averaged). All tests were two-sided with α = 0.05.

For Fig. 3d, we used a one-way ANOVA to compare uptake in the presence of each amino acid against the hot-only $^{14}$C-L-Leu control, followed by Dunnett's multiple-comparisons test (familywise α = 0.05). For Fig. 5a, data were analysed with an ordinary two-way ANOVA with factors "mutant" and "pH" (5.5, 7.5) using Type III sums of squares, including the interaction term. Simple-effects comparisons of pH 5.5 vs pH 7.5 within each mutant were performed with Šídák adjustment for multiple testing (familywise α = 0.05).

## Fluorescence microscopy localisation of SLC7A4 and mutants

Immunofluorescence was used to check for plasma membrane localisation of SLC7A4. Cells were seeded as a monolayer on glass coverslips, transfected as described above with FLAG-tagged SLC7A4, media were changed 12 hr post-transfection, and cells were processed for immunofluorescence 36 hr post-transfection. Cells were washed in PBS pH 7.4 and fixed in 4% paraformaldehyde for 7 min. Following quenching in 50 mM ammonium chloride for 3 mins and further PBS washing, the cells were permeabilised with 0.5% saponin for 8 mins, and coverslips were washed with PBS-GSA (1x PBS with 10 mM Glycine and 0.2% sodium azide). Coverslips were then blocked in 1% bovine serum albumin (BSA) in PBS for 1 hr. Cells were stained with 50 µL each of mouse anti-FLAG (1:200 dilution) and rabbit anti- Na$^+$/K$^+$ ATPase (1:50 dilution) (plasma membrane marker) primary antibodies for 1 hr, washed, and further stained with 50 µL each of goat anti-mouse IgG AlexaFluor-488 (1:200) and anti-rabbit IgG AlexaFluor-647 (1:200) secondary antibody-fluorophore conjugates for 45 mins in the dark. Cells were then stained with DAPI for 10 min in the dark and washed 4-5 times with PBS. Coverslips were mounted on glass slides using a drop of Immu-Mount (Epredia™ 9990402). Experiments were performed on an LSM-980 confocal microscope (Zeiss) and images were processed in ZenBlue (v3.9, Zeiss) and Fiji.

## Western Blot expression analysis of SLC7A4 and mutants

HCT116-SLC7A5-KO cells were harvested from 12-well plates 36 hr post-transfection by first detaching cells using TrypLE™ Express Enzyme (1X) (Gibco™ 12604013) for 3 min at room temperature and then collecting the detached cells using 1X PBS in eppendorfs. Harvested cells were spun at 3000 rpm for 5 min at 4 °C, and the pellet was then lysed using lysis solution (1x PBS, 1% DDM:0.2%CHS, DNase, protease inhibitor) on ice for 30 min. The lysate was separated by centrifugation at 30,000 g for 20 min at 4 °C. Total cellular protein was estimated using the BCA assay (Pierce™ BCA Protein Assay Kits). Two micrograms of the sample were loaded onto a 12% SDS-PAGE gel, then transferred to a PVDF membrane (Immobilon). The membrane was blocked using 3% milk in PBST for 1 hour, cut across 55 kDa, and each section was then treated overnight with primary antibodies anti-FLAG (Merck F1804) at 1:10,000 for the target band and anti-beta actin (Merck A2228) at 1:10,000 for the loading control. After primary antibody incubation, the cut membranes were treated with goat anti-mouse antibody (Agilent P044701-2), HRP (1:10,000) for 45 min. The membranes were then treated with ECL solution (Cytiva) and developed on X-ray films.

## Molecular dynamics simulations

*Preparation of outward-open AtCAT4 for MD*: The coordinates of SybB5 and cholesterols/PC lipids were deleted from the L-Orn-bound model. IL5 (residues 408-458 not resolved in cryoEM) were grafted from the AF2 predicted model (AlphafoldDB: AF-Q8W4K3-F1-v4) to fill in missing loops. A restrained geometry minimisation of residues within 6 Å of the binding site was performed using the conjugate gradient method in the gas phase until a convergence criterion of 0.01 kcal mol$^{-1}$ Å$^{-1}$ was reached, yielding the AtCAT4 MD model.

*Preparation of outward-open human SLC7A4 for MD*: We used the AtCAT4 MD model to build a homology model for human SLC7A4 in the outward-facing state. Specifically, the human SLC7A4 and AtCAT4 sequences were aligned with Clustal Omega, and the AtCAT4 coordinates were supplied as the template structure. MODELLER[90] was used (with 10 models generated), and the model with the highest DOPE score (89.705) was selected. The model was superimposed onto the inward-open human LAT2 leucine-bound structure (PDB: 7CMI), and the leucine coordinates were used. Since SLC7A4 and LAT2 are in the outward and inward-facing states, respectively, the model superpositions were based on the scaffold domain residues (scaffold residues in human SLC7A4 numbering: 96-233, 291-414, 474-591), excluding the mobile bundle domain that moves between

conformations. Tyr246 torsions were set to $\chi_1 = +178.3°$ (trans) and $\chi_2 = 74.1°$, to cap the docked ligand via pi-cation interactions as seen in other occluded ligand-bound SLC7 structures. Coordinates for the TM1-6-7 binding site water molecule were copied from the GkApcT crystal structure (PDB:5OQT). A restrained geometry minimisation of residues within 6 Å of the binding site was performed using the conjugate gradient method in the gas phase until a convergence criterion of 0.01 kcal mol$^{-1}$ Å$^{-1}$ was reached, yielding the *human SLC7A4 MD model*.

*Parameterisation and System Setup for MD*: MD systems were set up as listed in Supplementary Table 2. For outward-facing simulations, we prepared (i) the human SLC7A4 homology model bound to L-leucine and (ii) AtCAT4 bound to L-ornithine (cryo-EM derived). Additional conditions (apo and/or alternative starting conformations) were prepared and simulated as described below and in Table 2. The protein-ligand complexes were assembled in fully solvated lipid bilayers with the CHARMM-GUI membrane builder[91]. The surrounding membrane was composed of a 4:1 POPC: cholesterol mixture of target size 12 × 12 nm². We removed CHARMM-GUI waters and ions and added ACE/NME capping residues using PyMOL. Protein topologies were generated using the Amber99SB-ILDN forcefield[92] and lipids with Slipid forcefields[93]. The bound zwitterionic amino acid ligands were modelled with parm99SB atom types, as described by A.H.C. Horn[94]. The Horn parameters for L-leucine, available as.off file inputs (http://amber.manchester.ac.uk/), were loaded into AMBER18 tleap and subsequently converted to GROMACS-compatible.itp files via the acpype package[95]. The protein-lipid-ligand system was then solvated with 34,000 TIP3P[96] water molecules (the precise number varies between different systems), neutralised to an ionic concentration of 0.15 M with NaCl using GROMACS[97]. Each system was subjected to two rounds of refinement with heavy atom restraints, using both steepest descent and conjugate gradient methods, until convergence was achieved with $F_{max} < 500$ kJ/mol.

L-ornithine parameters were not available in the Horn force field set for proteinogenic zwitterionic amino acids (parm99SB-ILDN). They were therefore generated following the standard AMBER force field protocol for small molecules. The initial geometry of zwitterionic and protonated L-ornithine was optimised at the Hartree–Fock level with the 6–31 G* basis set using *Gaussian09*. The electrostatic potential was computed at the same level of theory and used to derive atomic partial charges via the restrained electrostatic potential (RESP) fitting scheme. Atom types and bonded parameters were assigned using the General Amber Force Field 2 (GAFF2) as implemented in *AmberTools18*. The resulting ligand library and parameter/topology files were then converted to GROMACS-compatible.itp files using the acpype package. This approach is consistent with the parameterisation strategy applied by Horn for zwitterionic amino acids.

Molecular dynamics simulations in explicit solvent used GROMACS versions 2020.3 and 2021.4 as the MD engine (the slight version discrepancy is because of different installations on the two compute clusters we used). The electrostatic interactions were calculated using particle mesh Ewald[98] with a short-range cutoff of 1.2 nm. The Lennard–Jones interaction cutoff was set to 1.2 nm, including the energy-pressure dispersion correction scheme implemented in GROMACS. Bonds involving hydrogen atoms were constrained by the LINCS algorithm, and a time step of 2 fs was used for the integration of the equations of motion. We assigned initial velocities and equilibrated each system according to the following protocol. The equilibration protocol used the weak-coupling Berendsen thermostat and barostat[99] to achieve exponential convergence of pressure and temperature. The protocol was over seven successive steps, two in the NVT ensemble (2× 125 ps) and five in NPT (125, 500, 1000, 2× 10,000 ps), using a leap-frog integrator. We gradually reduced position restraints over the seven steps, starting from 1000 kJ/mol for the ligand, 4,000 kJ/mol for the protein backbone, 2000 kJ/mol for the sidechains, and 1000 kJ/mol for

POPC P atoms and cholesterol O atoms (see step6.x_equilibration.mdp files). After equilibration, a 10 ns unrestrained NPT run was performed under production parameters, with semi-isotropic Parinello-Rahman pressure coupling[100] set to 1 bar and the modified v-rescale thermostat[101] (with a stochastic term) set to 310 K, to fully relax the system.

*Unbiased MD:* Unbiased production runs were carried out as summarised in Table 2 (AtCAT4 apo: 3×1 μs; AtCAT4 + L-Orn: 3×1 μs; Hs SLC7A4 HM + L-Leu: 1×1 μs) using the same simulation parameters as the final 10 ns equilibration step and retaining velocities. Snapshots were recorded every 50 ps. Trajectories were aligned on the TM-core backbone to remove global translation/rotation prior to analysis. In the cases where we needed to obtain representative bound poses for downstream short-repeat sampling (5×200 ns for AtCAT4 + L-Orn and 3×200 ns for the Hs SLC7A4 HM + L-Leu systems), we clustered protein–ligand configurations using Gromacs gmx cluster on defining the binding site as all heavy atoms of the residues within 10 Å of the ligand. We clustered with an RMSD cutoff of 1 Å. Clusters were ranked by population, and the representative structure was chosen as the cluster median (central structure, i.e., frame with minimal RMSD to other cluster members) of the most populated cluster. From this representative structure, we initiated independent repeats (distinct random initial velocities) using production settings. The protein-ligand interactions were calculated with MDAnalysis v. 2.9.0[102] and ProLIF v. 2.1.0[103].

### Reporting summary

Further information on research design is available in the Nature Portfolio Reporting Summary linked to this article.

### Data availability

Atomic coordinates for SLC7A4 have been deposited in the Protein Data Bank under accession codes: 9HJK (Apo with Syb), 9SP8 (L-Orn with Syb) and 9SQH (Apo without Syb). The cryo-EM maps have been deposited in the Electron Microscopy Data Bank (EMDB) under accession codes: EMD-52217, EMD-55065 and EMD-55110. Source data are provided with this paper.

### Code availability

The processed molecular dynamics simulations trajectories have been uploaded to Zenodo: (https://doi.org/10.5281/zenodo.17184930).

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

## Acknowledgements

This research was supported by Wellcome awards (215519/Z/19/Z & 219531/Z/19/Z) to SN and UKRI BBSRC award BB/Z517215/1 to JLP and SN. Computing was supported via the Advanced Research Computing facility, Oxford, the EPSRC ARCHER2 UK National Supercomputing Service and JADE (*EP/X035603/1*), granted via the High-End Computing Consortium for Biomolecular Simulation (HECBioSim-https://www.hecbiosim.ac.uk), supported by EPSRC (EP/X035603/1) to PCB. DK was supported by a BBSRC studentship (BB/ M011224/1) and an Onassis Foundation PhD scholarship award (F ZO 035-1/2018-2019). AB was supported by a BBSRC studentship (BB/T008784/1). The authors gratefully acknowledge the Micron Bioimaging Facility for their support & assistance in this work.

## Author contributions

D.K., J.L.P. and S.N. conceived the project. D.K. and A.B. performed all cloning, protein preparation and transport assays. D.K., T.K. and Y.C.Z performed all cryo-EM sample processing, data collection and image analysis. D.K. and S.N. constructed the atomic models with assistance from Y.C.Z. D.K., S.L. and P.C.B. performed all molecular dynamics simulations and analysis. D.K. and S.N. wrote the manuscript and prepared figures with contributions and discussions from J.L.P.

## Competing interests

The authors declare no competing interests.
