## [Transparent Peer Review file · Nature Communications]

Structural basis for pH-responsive amino acid transport via SLC7A4.

Corresponding Author: Professor Simon Newstead

Version 0:

Reviewer comments:

Reviewer #1

(Remarks to the Author)

The manuscript by Kolokouris et al. deals with a very important topic in the field of membrane transporters, that is, the resolution of a 3D structure of a 14TM membrane transporter. This finding is, per se, noteworthy. Moreover, the identification of substrate specificity of SLC7A4 from At and Hs triggers an advancement in the field with positive outcomes for human and plant physiology. The mechanism of induced fit changes in TM1 is also an interesting feature for the SLC7 family biology. Finally, the identification of a transport mechanism sensing protons is again a novelty in the field. For all the mentioned reasons the manuscript deserves to be published in Nat Comm. Only a few concerns arose which are listed below:

- 1) The introduction lacks of background in plant. Considering that the 3D structure is of an At transporter, the relevance in plant physiology, besides that in humans, should be discussed with particular reference to H⁺-dependence. Moreover, is not clear if in At the protein as an intracellular localization (e.g. vacuolar) or is in plasma membrane as authors show with IF for Hs isoform. This information should be provided.
- 2) In the introduction the reference to the result summary is not relevant in my opinion; I would rather substitute it with a rationale of the entire work.
- 3) In the result section, I would add a 2D topology of the transporter to show how the 14TMs are distributed and which helices are involved in the harm, hash or bundle domains if this classification is maintained.
- 4) The lack of activity for AtSLC7A4 in none of the used system may open question on the quality of the protein produced for CryoEM structure determination. Is that possible that the protein used for CryoEM is not functional and, hence, maybe in a folding state that is not stable enough?
- 5) The IC₅₀ for Leu and Arg in text are reported within a mM range, whereas in the figure are in uM range. Please correct.
- 6) Taking together the reported data, how do authors explain the different specificity towards Ornithine or Leu of the two transporters from At and from Hs? This should be explained also considering plant physiology.
- 7) Arg transport using the human transporter should be also tested to verify the hypothesis of a residual CAA transport activity. This is relevant also in terms of pH dependency to show if the pH dependency only occurs when neutral amino acids are transported or if it is conserved also in CAA.
- 8) Which is the physiological explanation for a pH dependent Leu transport occurring at the plasma membrane? They mention placenta and sperm maturation but without a specific speculation on the role for this transporter-pH changes-leucine in these body districts
- 9) In methods: information on the concentration of radioactive Leu used is missing, i.e. the Ci of the stock solution of ¹⁴C leucine.

Reviewer #2

(Remarks to the Author)

While general fold in this family is well explored, the SLC7A4 structure has a number of unique features that make this study novel and interesting. Helix 1 is straight but can unwind, the TM3/4 loop is elongated, and several lipids are found binding to the transporter. Moreover, the transporter is specific for leucine. The structure is that of Arabidopsis SLC7A4, while the human transporter was a homology model. The transporter mechanism appears plausible. Particularly interesting is the allosteric activation by acidic pH.

I have only a couple of minor comments.

Fig. 1B, please indicate that Fig. 1b shows the apo-structure and explain LMNG in the legend.

The space vacated by the flipping away of

Page 6: It is tempting to speculate that LMNG is replaced by a lipid in physiological circumstances. Have you seen any evidence of this?

Page 7: The flexibility of EL4 could be addressed by MD simulations.

Page 12: The units are wrong for the IC₅₀. This should be micromolar not millimolar.

Page 16: Is there active Arginine transport in the tri-M mutation or just inhibition?

Discussion:

Physiologically, the transporter is difficult to understand. It is entirely specific for leucine but inhibited by other amino acids. It could be that this transporter is active in acidic compartments but for what purpose? Perhaps it is more a sensor than a transporter, but this would go too far for this study.

Reviewer #3

(Remarks to the Author)

Kolokouris et al. report on structural and functional aspects underlying pH-responsive amino acid transport mediated by SLC7A4, the cationic amino acid transporter CAT4. The authors determined three structures of a plant homolog from *Arabidopsis thaliana*, i.e., in the apo form at 4.2 Å resolution, the apo form bound to a sybody at 3.3 Å resolution and the L-ornithine-sybody complex at 3.3 Å resolution. In addition, the authors performed functional assays and molecular dynamics (MD) simulations using a human SLC7A4 homology model, constructed based on the newly obtained structure of the SLC7A4 homolog from plant (*A. thaliana*). They identify and describe a pH sensor conserved in the bacterial homolog GkApcT whose structure they previously solved and extrapolate this finding to human SLC7A4. They propose a model for pH sensing, although additional data are required to fully elucidate the underlying mechanism.

In general, the submitted manuscript is unsatisfactory: it is suboptimally written, poorly structured and contains several major flaws (as detailed below). Furthermore, the manuscript would benefit from thorough proofreading to eliminate typographical errors and inconsistencies.

Major points

1. The manuscript is difficult to read and follow, as it lacks a clear and coherent structure. A major revision and thorough reorganization are required to improve its readability and logical flow. It is surprising, especially for a submission to a renowned journal such as *Nature Communications*, that the manuscript does not contain a distinct structure and clear red line. The section following the Introduction appears to combine elements of "Results and Discussion", yet a separate Discussion section is included later, creating confusion for the reader. Furthermore, some results (e.g., the pKa determination) are presented within the Discussion, which is unconventional and misleading. Overall, the current structure significantly detracts from the manuscript's clarity, forcing the reviewer to focus on organizational issues rather than on evaluating the scientific content.

2. As mentioned in my introductory paragraph above, this manuscript reports three structures of an SLC7A4 homolog from *Arabidopsis thaliana*. However, the authors do not explore in depth the structure or function of this protein. Instead, they use the obtained structure primarily as a template to build a homology model of the human SLC7A4, for which they did not determine the structure experimentally.

At this point, the reviewer is left wondering whether including the plant homolog's structure was indeed necessary for this manuscript. The same modeling, molecular dynamics simulations and functional experiments could potentially have been carried out using existing structures of mouse CAT1 and GkApcT as templates or simply using an alphafold predicted model.

Based on these considerations, the authors should generate homology models of human SLC7A4 using the published mouse CAT1 and GkApcT structures as well as the human SLC7A4 model from alphafold, and compare these to the model of human SLC7A4 derived from the *Arabidopsis* homolog structure. If these alternative models predict similar structural features, the inclusion of the new cryo-EM structure becomes questionable. If they differ, the authors should justify their choice of the plant homolog as a structural basis by including and discussing a comparative analysis of all human SLC7A4 models (obtained from the mouse CAT1, the *Arabidopsis* SLC7A4 homologue and GkApcT as well as the alphafold model) in the Supplementary Information.

3. Cryo-EM density of the L-ornithine

- Page 8, text: "The cryo-EM map reveals a well-resolved density for L-Orn in the canonical amino acid binding site..."

The statement "well-resolved" is a bold statement for a density, which basically fails to show atomistic features of L-ornithine nor nicely surrounds the molecule as shown in Fig. 1d. Please moderate such statements.

- The density of the substrate in Fig. 1d and Extended Data Fig. 6a look to me "carved". Furthermore, only one viewing angle is shown, i.e., the same view in Fig. 1d and Extended Data Fig. 6a.

If the density was indeed carved, please include the uncarved density in Fig. 1a and Extended Data Fig. 6. In the latter, three different views of the uncarved density (substrate and surrounding interacting amino acids) should be provided to allow a comprehensive assessment of the quality of the obtained substrate density (and surrounding). Finally, please indicate in these two Figures at which threshold the density is displayed.

- Major point: Based on Extended Data Fig. 2, only about 12% of particles of the "At SLC7A4-sybody-and-L-ornithine bound" structure apparently bound L-ornithine. This reviewer is not convinced by the final "3D-classification without alignment" run used to select the L-ornithine bound state.

To exclude the possibility of classification artifacts or spurious density being refined as ligand, the authors must verify their procedure with the apo structure. Briefly, perform the same analysis (as in Extended Data Fig. 2, e.g., using same number of

3D classes and same parameters) with the At SLC7A4-sybody apo structure (no L-ornithine) and prove that no comparable density is found in the substrate binding pocket arising from artefacts. This analysis should be included in Supplementary Information as a new Extended Data Figure.

Alternatively, the authors could repeat the cryo-EM experiment using a higher concentration of L-ornithine to achieve improved occupancy of the substrate binding site, thereby providing a clearer and more convincing density for L-ornithine and enabling the reliable construction of a publishable structural model.

4. After reading the manuscript, the claim that SLC7A4 proteins are, or could be, pH-responsive amino acid transporters remains somewhat hypothetical, indicating a lack of demonstrated novelty. Additional experimental evidence is needed to substantiate this assertion, which is strongly emphasized in the title but not sufficiently supported or demonstrated in the presented data.

Significant points to be addressed:

5. The SLC nomenclature should be carefully verified and applied consistently. In my view, it is incorrect to refer to the *Arabidopsis thaliana* homolog of (human) SLC7A4 as "At SLC7A4". For instance, solute carriers from mouse are written as Slc7a4 (in contrast to SLC7A4 for the human one). Therefore, in the context of this manuscript, the current "At SLC7A4" designation should be replaced with AtCAT4 or described as "the homolog from *Arabidopsis thaliana* (or plant) of (human) SLC7A4."

Importantly, this adjustment would eliminate the existing confusion in the manuscript, where it is not always clear whether the authors are referring to the plant homolog (AtCAT4) or to human SLC7A4.

6. Abstract: The Abstract is written in very general terms, making it unclear what was actually accomplished in the present study. While this broad formulation may make the work sound more appealing, it omits key information such as the fact that the structure determined is that of a plant homolog of human SLC7A4, and that the subsequent analyses are based primarily on a homology model of human SLC7A4 derived from this plant structure.

As stated above, this raises serious questions about the necessity of including the AtCAT4 structure, since an equally strong homology model could likely have been built using the already published mouse CAT1 or GkApcT structures, or an SLC7A4 alphafold model. As it stands, the manuscript appears somewhat patchwork-like, combining datasets that do not fit seamlessly together.

Therefore, the authors are encouraged to revise the Abstract so that it more realistically reflects the actual scope and content of the presented work, avoiding any impression of overselling.

Other points, mainly minor:

- General comment: Providing the manuscript with line numbering would have been helpful and would have facilitated the review process.

- Page 2 "Structures of many eukaryotic HAT transporters...": Please define the abbreviation HAT upon first use.

- Page 3: "Nevertheless, we were intrigued by the close homology of SLC7A4 to GkApcT...".

As the authors provided the amino acid sequence identity between SLC7A4 and SLC7A1/2, this reviewer suggests providing also the amino acid sequence identity between SLC7A4 and GkApcT.

- Page 4, last paragraph: This section is written in an unclear manner. The authors report the structure determination of the apo and L-ornithine-bound AtCAT4 in complex with a sybody. However, Figure 1 also includes an apo structure without the sybody. The authors are therefore asked to align the text more precisely with the figure panels in Figure 1.

In addition, providing the individual resolutions of the obtained structures would strengthen the statement "...to aid particle alignment during data processing..." and help readers without a structural biology background better understand the rationale for using a sybody.

Furthermore, I guess Extended Data Fig. 2 shows the processing workflow for the sybody-bound structures. Please clarify this point, for example by adding the corresponding information to the legend of Extended Data Fig. 2.

- Page 5, paragraph below Fig. 1 caption ("Although the apo structure..."):

First, the rationale for comparing At SLC7A4 with SLC7A5 and SLC7A7 is unclear - why were these transporters chosen instead of, for example, mouse CAT1 or GkApcT, which appear to be more closely related? Please clarify the reasoning behind the selected structural comparisons by providing pertinent information.

Second, the statement that the structure "...closely resembles..." these transporters is somewhat misleading given the reported RMSD values of 4.5-5.6 Å, which indicate only moderate structural similarity.

- Page 5, Figure 1d: The reviewer requests visualizing the molecules and hydrogen bonds without including hydrogen atoms, as their positions cannot be assigned with confidence at the achieved resolution. To enable a clearer assessment of the substrate pose within the binding pocket, the reviewer further requests displaying distances (numbers, Å) measured between heavy atoms directly in the figure.

- Page 6, "The function of these additional helices is currently unclear. The most likely explanation is that they compensate for the absence of the heavy chain subunit and aid stability or folding of CAT family members."

Since the heavy-chain subunit interacts with the opposite side of the transporter compared to TMs 11 and 12 of At SLC7A4 (as described in the manuscript), it is questionable whether such a compensatory effect can be rationalized.

- Page 6, "...occupied by a Lauryl-maltose neopentyl glycol (LMNG) detergent molecule (Extended Data Fig. 3e). The acyl chains of the detergent..." and onward to the end of this paragraph.

The authors are asked to provide cryo-EM density proof for those statements in an Extended Data Figure.

- Pages 6-7, "We thus conclude that whilst we cannot rule out the impact of the sybody on the position of EL4 in the higher-resolution structure, our data suggest that EL4 is dynamic and may flip away from the transporter in the outward-facing state."

The authors have addressed the potential impact of sybody binding; however, was the possibility also considered that the observed "flipped-out" conformation of EL4 might result solely from its interaction with the LMNG molecule? The reviewer believes that this possibility should be examined before concluding with confidence that EL4 is inherently dynamic.

- Page 7, "...well-defined densities for lipid molecules (Extended Data Fig. 5a-b)."

Based on the information presented in Extended Data Fig. 5a, this reviewer remains unconvinced by the authors' claim of "well-defined density." Please clarify this statement and provide additional views of the corresponding density in an Extended Data Figure to substantiate the claim.

- Page 7, "Our data indicate this pocket is a conserved cholesterol-binding site within the CAT sub-family and is worth further study to understand any regulatory role cholesterol might have on CAT transporter function."

Referring to the site as "conserved" appears to be an overstatement given that this conclusion is based on a single structure. Please revise the wording accordingly to avoid overinterpretation.

- Page 8, "In contrast, the ϵ -amino group on the L-Orn side chain interacts with Asp116 on TM3 via a salt bridge and to Asp243 on TM6b via a water-mediated hydrogen bond (W1) (Fig. 1d and Extended Data Fig. 6a)."

This reviewer is concerned that, at the achieved resolution, reliable placement of water molecules may be challenging. Are the positions of the modeled water molecules supported by the MD simulations presented later in the manuscript? If not, please remove this statement since not support by well defined, high-resolution density.

- Page 8, "The structure of SLC7A4 bound to L-Ornithine (L-Orn) exhibits..."

I assume this structure corresponds to the sybody-bound form. Please clarify this point and revise the text accordingly, as it is not clearly stated in the current version

- Page 8, "...forming a short π -helix bulge..."

This is an interesting structural feature; however, Fig. 1c appears to show an elongated helical element rather than a bulge. Please provide a more appropriate image to illustrate this feature accurately.

- Page 8, "...water-mediated hydrogen bond..."

The obtained resolution (>3 Å) is generally insufficient to support such detailed structural conclusions. Please revise the statement accordingly.

- Page 9, "The binding pose we observe for L-Orn in SLC7A4 is also consistent with the pose we reported previously for the more distantly related prokaryotic homologue, GkApcT (PDB 5OQT), which also involves an interaction of the side chain guanidino group with an acidic side chain on TM337 (Extended Data Fig. 6c)."

Please rephrase this sentence to clearly indicate that the structure also includes GkApcT bound to L-arginine. In addition, note that the PDB code differs from that provided in Extended Data Fig. 6c and actually corresponds to the alanine-bound GkApcT (likely a typographical error). Considering the major concern raised regarding the ligand cryo-EM density, the authors are kindly requested to reassess whether the available density sufficiently supports the structural comparisons between the binding sites of SLC7A1 and GkApcT.

- Page 11, "Compared to the cryo-EM model, Ala44 is shifted by an additional ~ 1.7 Å over the span of the MD repeats, further drawing the ligand backbone carboxyl into the fully accessible GAG backbone pocket."

The MD simulations are presented as supporting evidence for the proposed induced-fit mechanism. However, this reviewer remains uncertain about the interpretation of the events observed during the simulations. Specifically, the further unwinding of TM1, the entry and coordination of a water molecule and the "better accommodation" of the substrate. Why are these features not observed in the cryo-EM data and what is their significance in the context of the proposed mechanism? Please adapt text and provide additional information and argumentation.

- Page 11, Figure 2, panels b-d: The extent of the unwinding motion around the GAG-motif is a bit difficult to assess from the provided views of TM1 and TM6. Maybe superimposed structures of a closer view around the binding pocket would be helpful to improve assessment.

- Pages 19-20: The description of the proposed transport model suffers from issues of sequencing and redundancy, which negatively affect readability and comprehension. The induced-fit binding of L-leucine is described after the transition from the outward-open, substrate-free state to the outward-facing occluded, L-leucine-bound state (i.e., the transition from state i to ii), which disrupts the logical flow. Additionally, protonation of Glu125 is mentioned twice (passages 1 and 2) and could be streamlined. The hypothesized rotation of TM3 into the position observed in the AtCAT4 structure is also described somewhat differently than on page 19. This reviewer therefore requests that this paragraph be restructured and improved for greater clarity and consistency.

- Page 26, "Thermal stability measurements".

Please report on the final protein concentration and buffer constituents used for the assay/measurement.

-Extended Data: The Extended Data section does not include representative micrographs, 2D class averages or viewing direction (azimuthal angle) distribution plots. Please add these essential elements to provide a more complete overview of the cryo-EM data quality and processing.

Reviewer #4

(Remarks to the Author)

The manuscript by Kolokouris et al. focusses on the transport mechanism of SLC7A4, an amino acid transporter of the solute carrier protein family. Unlike homologous proteins some structural and functional features are unique, which are shown by means of cryoEM structures, MD simulations, and pharmacological experiments. In particular two aspects are interesting and deserve publication. First the unique profile of transported amino acids (selective for leucine). And second, the pH sensitivity of the transporter. I want to state that I am not an expert in the technical aspects of cryoEM, so I cannot fully judge on this particular part.

Given the interesting topic, the sound methodology used, and the way of presenting the results, I would like to recommend publication of this manuscript in Nature Communications. However, some aspects have to be improved by the authors before publication.

1) While the MD simulations have been carried out in a sound manner, I was confused by their description. E.g., I assume that the Table 2 is right, but in the methods section the authors state they have simulated in triplicates rather than 5 independent 200 ns simulations. It is also not clear from the text how many μ s MDs have been carried out and from which of those replicas the clusters were selected. Please make the method section consistent and describe in more details how the clustering and selection of the representative frame was done.

2) As a minor point, the authors mix one-letter and three-letter code for amino acids. Since it makes sense for the figures to use one-letter code, please also apply this throughout the manuscript.

3) I miss a detailed structural comparison with other SLCs/amino acid transporters. Since new structures are reported in this manuscript, it would be nice to compare them with previous structures and highlight the unique features in more detail.

4) The section on structural basis for leucine selectivity in Hs SLC7A4 is quite interesting, but misses a detailed discussion on the molecular level. The authors show the lipophilic residues that are interacting with leucine's side chain and then directly head to arginine. It would be beneficial to discuss more closely related amino acids, e.g. valine, isoleucine. Why is the binding site so prone to leucine?

Version 1:

Reviewer comments:

Reviewer #1

(Remarks to the Author)

Authors addressed all my concerns

Then, manuscript is suitable for publication

Reviewer #2

(Remarks to the Author)

The authors have addressed all concerns raised by this assessor.

Reviewer #3

(Remarks to the Author)

The corrected manuscript version of Kolokouris et al. reads substantially more clearly, and the authors have addressed the points raised by the reviewer. The abstract now more accurately reflects the scope of the work. The implementation of the suggested protein nomenclature enhances readability, and section transitions were refined through targeted edits while the overall structure was retained. The discussion has been clarified, with previously noted issues in logical flow largely resolved, particularly with respect to the proposed transport model.

General concerns regarding the rationale of the work remain. Nevertheless, the authors now provide a clearer and more consistent argument for the relevance of the AtCAT4 structure in the context of residue-level analyses of SLC7A4, which is sufficient to justify the approach taken.

To assess whether the limited quality of the ligand density might represent an artefact, the authors performed the suggested negative control experiment for ligand-density selection. This analysis supports the modelling of L-ornithine. Furthermore, the additional functional data on SLC7A4 support pH-dependent regulation of transport activity; however, the proposed proton-sensing model remains speculative and will require further validation beyond the present study.

Overall, while the manuscript does not fully overcome its intrinsic limitations, the authors have adequately addressed the

reviewer's concerns, and the study is suitable for publication in its current form. Minor improvements to the presentation, such as providing better-resolved images in Extended Data Fig. 2c and Extended Data Fig. 3 and clarifying the description of the particle stacks used for the second refinement round in the Methods and Extended Data Fig. 2b, would further improve clarity.

Reviewer #4

(Remarks to the Author)

The authors have revised the manuscript and addressed all points raised during review.

While I still believe that a more detailed structural comparison of the new cryoEM structure with previous ones or closely related transporters would strengthen the manuscript, it doesn't represent a point that should hinder publication of this manuscript. I also understand the rationale of using three-letter code in the text and one-letter code in the figures.

Response to reviewers' comments:

We appreciate and value the additional workload involved in reviewing our paper and thank the reviewers for their time and feedback. We have extensively reworked the text to address queries regarding clarifications, rationale, and interpretations of the data. Below, we outline the major changes to the manuscript. Amendments to the main text are highlighted in green and included below to help identify key updates

Please note that, in light of the reviewer's comments, we have revised our nomenclature in the paper and refer to the plant SLC7A4 protein as AtCAT4 and the recent structures of the *Mus Musculus* SLC7A1 and MmCAT1 throughout.

Reviewer 1.

The manuscript by Kolokouris et al. deals with a very important topic in the field of membrane transporters, that is, the resolution of a 3D structure of a 14TM membrane transporter. This finding is, per se, noteworthy. Moreover, the identification of substrate specificity of SLC7A4 from At and Hs triggers an advancement in the field with positive outcomes for human and plant physiology. The mechanism of induced fit changes in TM1 is also an interesting feature for the SLC7 family biology. Finally, the identification of a transport mechanism sensing protons is again a novelty in the field. For all the mentioned reasons the manuscript deserves to be published in Nat Comm. Only a few concerns arose which are listed below:

We thank the reviewer for their support and enthusiasm for our study.

1) The introduction lacks of background in plant. Considering that the 3D structure is of an At transporter, the relevance in plant physiology, besides that in humans, should be discussed with particular reference to H⁺-dependence. Moreover, is not clear if in At the protein as an intracellular localization (e.g. vacuolar) or is in plasma membrane as authors show with IF for Hs isoform. This information should be provided.

We thank the reviewer for pointing out this omission. We have added a paragraph introducing the *Arabidopsis thaliana* CAT homologues to the introduction. From the literature, it is unclear whether AtCAT4 is pH dependent, as, to the best of our knowledge, no one has functionally characterised this protein prior to our study. Nevertheless, in *Arabidopsis*, CAT1, CAT5 and CAT6 are reported to localise to the plasma membrane and to mediate pH-dependent amino-acid uptake, as we now highlight in the main text and in the references, and these sequences are included in an updated Extended Data Fig. 1a-b.

2) In the introduction the reference to the result summary is not relevant in my opinion; I would rather substitute it with a rationale of the entire work.

We have removed the standalone results-summary paragraph and replaced it with a motivation-focused paragraph that outlines the rationale for the study:

Here, we identify SLC7A4 as a pH-responsive plasma membrane leucine transporter in humans and demonstrate that its plant homologue can bind both cationic amino acids and leucine. Combining structural, biochemical and functional data, we further demonstrate the close evolutionary conservation between human and plant transporters and identify key side chains required for cationic amino acid recognition within the SLC7 family. (lines 103-108).

3) In the result section, I would add a 2D topology of the transporter to show how the 14TMs are distributed and which helices are involved in the harm, hash or bundle domains if this classification is maintained.

We have now included a 2D topology map of AtCAT4, the binding epitope of SybB5 in the vicinity of the density modelled as an LMNG molecule, and AtCAT4 + SybB5 dissociation constant determination and size-exclusion co-purification (Extended Data Fig. 3).

Revised Extended Data Fig. 3. Cryo-EM structure of AtCAT4 bound to sybody B5 (SybB5). a. Cryo-EM structure of apo AtCAT4 bound to SybB5 coloured from the N-terminus (blue) to the C-terminus (red). The topology diagram for the structure is shown to the right. The inverted topology repeats (TMs 1-5 & 6-10) are shown as inverted triangles. **b.** Electrostatic surface representation of AtCAT4 showing the structural epitope recognised by SybB5. **c.** LMNG binding interface on apo AtCAT4 bound to SybB5. The cryo-EM density for the bound LMNG molecule is shown (blue and threshold 0.3). **d.** Binding affinity (K_d) measurement for SybB5, n=2 independent measurements. **e.** Size Exclusion Chromatography (SEC) profile for the AtCAT4-SybB5 complex using a Superdex S200 10/300 GL Increase column. Inset, SDS-PAGE analysis of the fractions highlighted in the chromatogram. **f.** Circular Dichroism (CD) spectra for the LMNG purified sample used in the cryo-EM experiments. **g.** Thermal stabilisation of the LMNG purified AtCAT4 in the presence of increasing

concentrations of L-Ornithine. Increasing concentrations of L-Alanine were used as a negative control.

4) The lack of activity for AtSLC7A4 in none of the used system may open question on the quality of the protein produced for CryoEM structure determination. Is that possible that the protein used for CryoEM is not functional and, hence, maybe in a folding state that is not stable enough?

We thank the reviewer for raising the important question of whether the AtCAT4 protein used for cryo-EM could be non-functional due to misfolding or poor stability. Several independent observations argue against the misfolding/instability of the purified proteins used in this study.

First, the circular dichroism (CD) spectra of AtCAT4 in the detergent used for cryo-EM (LMNG, LMNG + lipids) show predominantly α -helical secondary structure with characteristic minima at 208 and 222 nm. We have added this as new panel f in the revised Extended Data Fig. 3.

Additionally, the thermal stability nanoDSF measurements demonstrate ligand-selective stabilisation. Indeed, AtCAT4 shows a titratable increase in T_m upon L-Orn addition (0-40 mM) but not upon L-alanine addition (Extended Data Fig. 3g). The nanoDSF assay has been validated as a reliable indicator for functional transporters¹.

Consistent with adopting the correctly folded state, WT AtCAT4 shows a strong thermal response to L-Orn, as shown in Fig. 1e (~4.5 °C), whereas binding-site mutants show markedly reduced responses (≤ 1 °C), including F237V. Notably, mutation of the homologous aromatic position has been reported to abolish transport in related SLC7 transporters (e.g., Y244A in System Xc²). We are confident these data support our conclusion that AtCAT4 is stably folded in LMNG and able to bind cationic amino acids.

Secondly, although we attempted to reconstitute the AtCAT4 transporter into proteoliposomes and to perform liposome-based transport assays. However, as shown in Figure 1 below, the reconstitution efficiency was $<20\%$, and the transporter did not show any activity under counterflow conditions or under an acetate-induced pH gradient.

Reconstitution of AtCAT4 in proteoliposomes. Left - 12 % SDS-PAGE gel showing the reconstitution of AtCAT4 from DDM-CHS.SN-supernatant. Right - ³[H] Amino Acid uptake assay using counterflow driven uptake.

In parallel, we reconstituted the bacterial transporter GkApcT as a positive control under identical conditions³. We suspect the lower reconstitution efficiency is due to the poor stability in the DDM-CHS detergent mix. Unfortunately, we have found that LMNG is a poor detergent for liposome reconstitution assays because it cannot be fully removed, resulting in leaky liposomes in our hands. Our conclusion from these experiments is that AtCAT4 is not suitable for liposome reconstitution assays without considerable additional optimisation.

However, we successfully reconstituted the human SLC7A4 protein and observed acetate-induced uptake across a pH gradient. We have included this new data in Extended Data Fig. 10c-d.

Together, our new biophysical and biochemical data support the interpretation that the AtCAT4 cryo-EM sample was folded and ligand-responsive, and that human SLC7A4 can transport L-Leucine into proteoliposomes.

5) The IC₅₀ for Leu and Arg in text are reported within a mM range, whereas in the figure are in μ M range. Please correct.

Done.

6) Taking together the reported data, how do authors explain the different specificity towards Ornithine or Leu of the two transporters from At and from Hs? This should be explained also considering plant physiology.

We have added a dedicated paragraph in our results section “The structural basis for leucine selectivity in SLC7A4” to address this.

Taken together, our data explain why the Arabidopsis CAT4 and human SLC7A4 homologues have diverged towards preferential recognition of L-ornithine versus L-leucine, and why this divergence is physiologically coherent. In AtCAT4, the binding pocket remains highly polar and is organised around the conserved TM3 acidic residue (Asp116) and surrounding hydrogen-bonding network. This network stabilises protonated, flexible side chains such as L-ornithine (and related cationic amino acids), while still tolerating L-Leu binding as indicated by our nanoDSF screen. Prior work further reports that AtCAT4 localises predominantly to the tonoplast and is broadly expressed. In this context, an Orn/CAA-biased pocket is most consistent with a role in vacuolar amino-acid buffering and nitrogen remobilisation. It would help control cationic amino-acid pools that feed plant nitrogen economy and polyamine/arginine-ornithine metabolism during growth and reproductive transitions. In contrast, HsSLC7A4 has remodelled the side-chain environment around the canonical TM1/TM6 motif into a predominantly hydrophobic pocket, centred on Val332, Leu336 and Leu394. This pocket tightly accommodates the branched side chain of L-Leu but disfavours the more extended and charged side chains of cationic amino acids. This is reflected functionally by a ~30-fold lower IC₅₀ for L-Leu compared with L-Arg and the absence of detectable L-Arg transport, despite measurable L-Arg binding. The Tri-M mutant, in which we introduce three CAT1-like residues into this pocket, re-establishes high-affinity L-Arg recognition and transport while modestly weakening L-Leu affinity. This demonstrates that a small set of side chains can switch the CAT4 scaffold between an Orn/CAA-biased state suited to plant vacuolar homeostasis and a Leu-biased state suited to mammalian leucine handling at specialised interfaces.

7) Arg transport using the human transporter should be also tested to verify the hypothesis of a residual CAA transport activity. This is relevant also in terms of pH dependency to show if the pH dependency only occurs when neutral amino acids are transported or if it is conserved also in CAA.

We agree that assessing arginine transport would be informative, especially for establishing the transport characteristics of the Tri-M.

We attempted to establish an arginine-uptake assay for human SLC7A4 in our HCT116 LAT1-knockout background, but basal [³H]-L-Arginine uptake remained high and prevented robust separation of SLC7A4-dependent signal from endogenous transport. Indeed, we also transfected the currently available cell line with human CAT1 and did not record robust transport above background under assay conditions, despite CAT1 being a well-known arginine transporter. We turned to HeLa cells for the same assay but also observed high levels of arginine background noise across time points. This observation is consistent with quantitative transporter data in the literature, showing that arginine uptake in cells is largely mediated by y⁺LAT2 (system y⁺L) and CAT transporters, with y⁺LAT2 accounting for the majority of the flux⁴.

We therefore considered using a combination of pharmacological tools to reduce the background signal. However we could not find a clean way to suppress the endogenous background uptake without simultaneously confounding the activity for SLC7A4: N-ethylmaleimide (NEM) inhibits CAT family transporters (system y⁺; potentially including SLC7A4) but not y⁺LAT isoforms⁵ and thus cannot eliminate the dominant y⁺LAT2 component; conversely, competitive suppression of y⁺LAT2 with neutral amino acids such as leucine is not diagnostic because leucine is also the primary substrate/inhibitor for SLC7A4 (IC₅₀ for L-Leu 24 ± 5 μM, Fig. 3f). Other neutral amino acids also inhibit SLC7A4 transport as shown in Fig. 3d.

Therefore, a definitive cellular arginine-transport assay for SLC7A4 would require additional genetic removal of the major endogenous arginine transporters (e.g., CAT1 and y⁺LAT2) in addition to LAT1 KO in HCT116 cells, which is challenging and beyond the scope of the current study.

As noted above, we successfully reconstituted human SLC7A4 into liposomes. However, the level of reconstitution was ~ 25 % and the activity, whilst significantly above background (**Extended Data Fig. 9c-d**), is too low for the detailed biochemical studies asked for by the referee. It is our intention to follow up on this line of questioning once we have optimised the liposome reconstitution assay to enable robust activity measurements. In the meantime, we have revised the manuscript to state that our results support the role of V332S/L336S/L394M in amino acid recognition, and to leave the question of whether the triple mutant can transport arginine for a follow-up study.

8) Which is the physiological explanation for a pH dependent Leu transport occurring at the plasma membrane? They mention placenta and sperm maturation but without a specific speculation on the role for this transporter-pH changes-leucine in these body districts

We have revised our final discussion paragraph to reflect our current hypothesis for the role of SLC7A4-mediated pH-dependent leucine transport at the plasma membrane. Specifically:

More broadly, our study uncovers an additional plasma membrane leucine transporter within the SLC7 family, but with the distinctive feature that it has evolved to respond to changes in extracellular pH. Gene expression data report SLC7A4 enrichment in the choroid plexus (blood–brain barrier) and testis, with secondary expression reported in placental trophoblasts. These tissues share a common physiological property: they function as highly regulated nutrient exchange interfaces, where acid–base handling plays an integral role^{6,7}. In this physiological setting, a pH-gated leucine transport system could be advantageous as it would

harness local acidity changes as an activation mechanism that promotes cellular leucine uptake.

9) In methods: information on the concentration of radioactive Leu used is missing, i.e. the Ci of the stock solution of ¹⁴C leucine.

We thank the reviewer for noting this and have now included the information in the Methods in lined 1122-1123 and 1154-1155:

... ¹⁴C-L-leucine [(stock: 295.86 µM 338 mCi/mmol), 2.96 µM final assay concentration, 1:100 hot:cold dilution].

Reviewer 2.

We thank the reviewer for their comments, which have benefited our manuscript, and have fully implemented their recommendations in this version.

While general fold in this family is well explored, the SLC7A4 structure has a number of unique features that make this study novel and interesting. Helix 1 is straight but can unwind, the TM3/4 loop is elongated, and several lipids are found binding to the transporter. Moreover, the transporter is specific for leucine. The structure is that of Arabidopsis SLC7A4, while the human transporter was a homology model. The transporter mechanism appears plausible. Particularly interesting is the allosteric activation by acidic pH.

I have only a couple of minor comments.

Fig. 1B, please indicate that Fig. 1b shows the apo-structure and explain LMNG in the legend.

We have added an updated Fig. 1b legend with: “**b.** Cryo-EM structure of apo AtCAT4 with TMs 1 and 6 highlighted and an LMNG and cholesterol molecules bound to the extracellular side of the outward-open transporter shown as blue sticks.”

Page 6: It is tempting to speculate that LMNG is replaced by a lipid in physiological circumstances. Have you seen any evidence of this?

We agree this is an attractive idea, but we do not have direct evidence that LMNG is replaced by a specific endogenous lipid under physiological conditions.

What we do observe is a well-defined non-protein density in the TM1/TM7/EL4 region where the “flipped-in” EL4 should occupy during the transport cycle. Our density is best explained by an LMNG molecule (Extended Data Fig. 4f): (i) its size/shape is consistent with an LMNG-like detergent headgroup plus hydrophobic tails, (ii) it is present across all LMNG-containing datasets we collected (±L-Orn, +SybB5), and (iii) in all these conditions the transporter remains outward-open, consistent with LMNG occupying the space normally taken by the “flipped-in” EL4 conformation during the transport cycle. In our samples, LMNG was present at ~30 µM (0.003% w/v), which is sufficient to populate such a hydrophobic cavity and stabilise a single conformational state. Notably, TM7 in SLC7A4 is elongated relative to other APC-fold members, resulting in a shorter EL4, which may facilitate the flipped-out configuration captured in our cryo-EM reconstructions.

While we have no direct evidence that an endogenous lipid occupies the cavity formed by the flipping out of EL4 in SLC7A4, structurally analogous vestibule pockets in SLC6 transporters can accommodate lipid-like allosteric ligands (e.g., oleoyl-D-lysine in GlyT2^{8,9}) supporting the plausibility that detergent can mimic occupation of a hydrophobic site.

Page 7: The flexibility of EL4 could be addressed by MD simulations.

We have now addressed EL4 flexibility directly using the MD simulations we already performed. Specifically, we analysed six independent all-atom trajectories of the outward-open cryo-EM starting model (apo, $n = 3$; +L-Orn, $n = 3$) and, for comparison, trajectories initiated from an inward-facing AlphaFold2-derived model in which EL4 is “flipped-in” ($n = 3$; +L-Orn). We summarise these results in a new Extended Data Fig. 5.

Across the combined MD ensemble, starting from our cryo-EM structures, we observed EL4 stably in the flipped-out conformation. Principal component analysis on backbone atoms of EL4 (including the inward-facing flipped-in MD AF2-derived ensemble) shows that the dominant mode (PC1; >80% of the variance) corresponds to a reaction coordinate describing EL4 “flip-out to flip-in” motion. This coordinate clearly separates the outward-open (apo and +L-Orn; cryo-EM-derived) simulations from the inward-facing (AF2-derived) simulations, consistent with EL4 adopting distinct conformational states throughout the transport cycle. Importantly, within the outward-open state, EL4 dynamics are not detectably altered by ligand, i.e. per-residue RMSF profiles show no statistically significant differences between apo and +L-Orn trajectories. As expected, EL4 RMSF is higher in the outward-open simulations (where the loop is solvent-exposed) than in the inward-facing simulations (where EL4 is more constrained by contacts with TM1b). Overall, these MD results are consistent with our proposed mechanism in which EL4 is flipped out in the outward-open conformation and subsequently stabilises downstream TM1 rearrangements required for occluded/inward-facing states, as observed for the mouse CAT1 transporter captured in inward-facing conformations^{10,11}.

The MD data on EL4 prompted us to re-examine the low-resolution Syb dataset (also collected in LMNG) with a specific focus on EL4. As noted in the main text, this reconstruction shows clear density for all 14 transmembrane helices and for the structured, symmetry-related loop region IL1, but no interpretable density for EL4, consistent with substantial mobility. The MD simulations further indicate that both “flipped-out” and “flipped-in” EL4 configurations can be stable endpoints of the transition, motivating us to test whether multiple EL4 states could be resolved from our cryo-EM data. Consistent with this hypothesis, focused classification of the final particle stack of the -Syb dataset yields a minor class with an alternative, more “flipped-in” EL4 placement packed against TM1b (Extended Data Fig. 12), supporting that EL4 can sample multiple conformations even under identical detergent conditions. It is thus our conclusion that LMNG in the pocket in the +Syb datasets results from EL4 physiologically adopting its flipped-out conformation. The void left is then likely occupied by an LMNG molecule.

Extended Data Fig. 5. Molecular Dynamics and cryo-EM analyses support the conformational heterogeneity observed in the EL4 loop. a, Essential-dynamics/PCA of EL4 backbone coordinates using a shared basis across simulations of the outward-open cryo-EM starting model (apo, $n = 3$; +L-Orn, $n = 3$) and an inward-facing AlphaFold2-derived model with EL4 “flipped-in” (+L-Orn, $n = 3$). 2D projections (PC1/PC2) show that the dominant mode (PC1; $>80\%$ variance) corresponds to an EL4 flip-out \rightarrow flip-in coordinate, separating the outward-open ensembles from the inward-facing ensemble. The PC1 mean is similar for apo and +L-Orn outward-open simulations. Per-residue EL4 C α RMSF (mean \pm SD across repeats) is not detectably changed by ligand in the outward-open state, and is higher than in the inward-facing ensemble, consistent with solvent exposure when EL4 is flipped out and constraint against TM1b when flipped in. The PCA variance spectrum is shown for the pooled trajectories. **b**, Structural context for the EL4 states. Representative outward-open cryo-EM model shows EL4 flipped out and a detergent-sized cavity adjacent to TM1/TM7. The low-resolution -Syb dataset (LMNG) lacks interpretable EL4 density, consistent with mobility. Preliminary EL4-focused 3D classification of the -Syb particle stack yields a minor class (34,329/120,868 particles, 28%) with an alternative, more flipped-in EL4 placement packed against TM1b. Density threshold shown at 0.1.

Page 12: The units are wrong for the IC50. This should micromolar not millimolar.

Done.

Page 16: Is there active Arginine transport in the tri-M mutation or just inhibition?

Due to high arginine background noise in the SLC7A5-KO HCT116 cells we used for our assays, we were not able to robustly quantify arginine transport in the Tri-M. A future direction for achieving arginine tracking would be to optimise our HCT-116 cell line by serially testing CAT1-3 knockouts until a minimal background could be achieved with an empty plasmid control. Indeed, we also transfected the currently available cell line with CAT1 and did not record robust transport above background under assay conditions, despite CAT1 being a well-known arginine transporter. We turned to HeLa cells for the same assay but also observed high levels of arginine background noise across time points. We reserve this optimisation for future work.

Discussion:

Physiologically, the transporter is difficult to understand. It is entirely specific for leucine but inhibited by other amino acids. It could be that this transporter is active in acidic compartments but for what purpose? Perhaps it is more a sensor than a transporter, but this would go too far for this study.

We thank the reviewer for highlighting the need to clarify the physiological relevance of our findings. While defining a specific *in vivo* role will require dedicated *in vivo* studies beyond the scope of this work, the mechanistic insights we report, together with recently published Resolute expression data, point to SLC7A4 functioning as a leucine transporter in epithelia characterised by tightly regulated acid-base microenvironments and a need to maintain nutrient-rich intracellular amino acid pools. Accordingly, we have revised the Discussion to first present the proposed transport mechanism (as before) and then to add the following physiological clarifications to contextualise these molecular findings (lines 755-763):

More broadly, our study uncovers a new plasma membrane leucine transporter within the SLC7 family, with the distinctive ability to respond to changes in extracellular pH. Gene expression data indicate SLC7A4 enrichment in the choroid plexus (blood–brain barrier)¹² and testis¹³, with secondary expression reported in placental trophoblasts¹⁴. These tissues share a common physiological property: they function as highly regulated nutrient-exchange interfaces, where acid–base handling plays an integral role^{6,7}. In this physiological setting, a pH-gated leucine transport system could be advantageous as it would harness local acidity changes as an activation mechanism that promotes cellular leucine uptake.

Reviewer 3.

Kolokouris et al. report on structural and functional aspects underlying pH-responsive amino acid transport mediated by SLC7A4, the cationic amino acid transporter CAT4. The authors determined three structures of a plant homolog from *Arabidopsis thaliana*, i.e., in the apo form at 4.2 Å resolution, the apo form bound to a sybody at 3.3 Å resolution and the L-ornithine-sybody complex at 3.3 Å resolution. In addition, the authors performed functional assays and molecular dynamics (MD) simulations using a human SLC7A4 homology model, constructed based on the newly obtained structure of the SLC7A4 homolog from plant (*A. thaliana*). They identify and describe a pH sensor conserved in the bacterial homolog GkApcT whose structure they previously solved and extrapolate this finding to human SLC7A4. They propose a model for pH sensing, although additional data are required to fully elucidate the underlying mechanism.

In general, the submitted manuscript is unsatisfactory: it is suboptimally written, poorly structured and contains several major flaws (as detailed below). Furthermore, the manuscript would benefit from thorough proofreading to eliminate typographical errors and inconsistencies.

We thank the reviewer for the detailed critique. We have revised the manuscript extensively to address the readability, organisation, and substantiation concerns raised and to ensure that all statements are appropriately supported by the data presented.

Major points

1. The manuscript is difficult to read and follow, as it lacks a clear and coherent structure. A major revision and thorough reorganization are required to improve its readability and logical flow. It is surprising, especially for a submission to a renowned journal such as *Nature Communications*, that the manuscript does not contain a distinct structure and clear red line. The section following the Introduction appears to combine elements of "Results and Discussion", yet a separate Discussion section is included later, creating confusion for the reader. Furthermore, some results (e.g., the pKa determination) are presented within the Discussion, which is unconventional and misleading. Overall, the current structure significantly detracts from the manuscript's clarity, forcing the reviewer to focus on organizational issues rather than on evaluating the scientific content.

We take on board the reviewer comments, although we also appreciate, as will the reviewer, that every researcher has a particular style when writing and presenting their research. While our manuscript follows the standard structure (Introduction, Results, Discussion, Methods) we have now strengthened the "red line" through the Results/Discussion.

We also moved results-style content out of the Discussion (including the pKa analysis and other quantitative/technical statements), reserving the Discussion for interpretation in the broader amino-acid transporter literature.

However, we chose not to undertake a wholesale reordering of the Results because we consider the existing sequence to provide a coherent progression from the AtCAT4 structures to functional measurements and mechanistic interpretation of human SLC7A4. Nevertheless, we made targeted edits throughout to improve clarity, reduce redundancy, and sharpen transitions. This overall presentation strategy is consistent with prior *Nature Communications* transporter studies that use a plant homologue structure to inform mechanism and human functional determinants (e.g., the *Arabidopsis* cystinosin structures alongside human cystinosin functional/mutational analysis)¹⁵.

2. As mentioned in my introductory paragraph above, this manuscript reports three structures of an SLC7A4 homolog from *Arabidopsis thaliana*. However, the authors do not explore in

depth the structure or function of this protein. Instead, they use the obtained structure primarily as a template to build a homology model of the human SLC7A4, for which they did not determine the structure experimentally. At this point, the reviewer is left wondering whether including the plant homolog's structure was indeed necessary for this manuscript. The same modeling, molecular dynamics simulations and functional experiments could potentially have been carried out using existing structures of mouse CAT1 and GkApcT as templates or simply using an alphafold predicted model. Based on these considerations, the authors should generate homology models of human SLC7A4 using the published mouse CAT1 and GkApcT structures as well as the human SLC7A4 model from alphafold, and compare these to the model of human SLC7A4 derived from the Arabidopsis homolog structure. If these alternative models predict similar structural features, the inclusion of the new cryo-EM structure becomes questionable. If they differ, the authors should justify their choice of the plant homolog as a structural basis by including and discussing a comparative analysis of all human SLC7A4 models (obtained from the mouse CAT1, the Arabidopsis SLC7A4 homologue and GkApcT as well as the alphafold model) in the Supplementary Information.

We carefully noted and considered the reviewers points in these comments. The Arabidopsis thaliana homologue (AtCAT4; Q8W4K3) was selected from a panel of SLC7A4 homologues (Extended Data Fig. 1c-d; Rn, Mm, Dr, Hs) because it is a eukaryotic CAT-family member closely related to human SLC7A4, showed superior biochemical stability for cryo-EM, and retains CAT-subfamily features absent from the prokaryotic template GkApcT (notably TM11–TM12 and the conserved IL5 region) that are relevant to the eukaryotic architecture and transport cycle.

The importance of protein stability presents important practical benefits, such as the ability to withstand the generation and selection of sybodies¹⁶. Indeed, we could not improve the resolution of the AtCAT4 dataset without sybody SybB5, and despite collecting a large cryo-EM dataset for human SLC7A4 in leucine (~30,000 movies) we did not obtain a structure, consistent with the broader difficulty of solving human CAT-family proteins relative to stable orthologues. This is consistent with the broader pattern that CAT-family structures have more commonly been obtained from non-human orthologues (e.g., MmCAT1 rather than HsCAT1¹¹ and with the use of viral proteins to stabilise the mouse CAT1 transporter for structural studies¹⁰.

Available CAT-family templates also represent different conformational endpoints (AtCAT4 outward-open vs. MmCAT1 inward-open and GkApcT inward-facing occluded) and are therefore complementary rather than redundant for understanding the conformational landscape of the SLC7 cationic amino acid transporters. Notably, no existing structure provided an experimentally determined outward-open CAT framework for residue-level hypotheses central to our study (e.g., E125-unwound TM10 and Tri-M effects), so we use the AtCAT4 cryo-EM structures as the primary basis for mechanistic inference and for designing/interpreting human SLC7A4 mutagenesis and transport assays, while referencing MmCAT1- and GkApcT-based homology models where they inform other states (summarised in Fig. 6).

Finally, while additional homology models built from more distant templates in different conformations could be informative for other stages of the transport cycle, they would necessarily differ by construction and would not alter the conclusions anchored in the experimentally determined outward-open AtCAT4 structures.

3. Cryo-EM density of the L-ornithine

- Page 8, text: “The cryo-EM map reveals a well-resolved density for L-Orn in the canonical amino acid binding site...” The statement “well-resolved” is a bold statement for a density, which basically fails to show atomistic features of L-ornithine nor nicely surrounds the molecule as shown in Fig. 1d. Please moderate such statements.

While we are confident in our modelling, we have rephrased the beginning of that paragraph (lines 283-307):

The structure of AtCAT4 bound to L-Ornithine (L-Orn) adopts an outward-open conformation that closely matches the apo structure (Extended Data Fig. 2b), with an r.m.s.d. of 1.16 Å over 512 C_a atoms. The structure provides valuable insight into the mechanisms of ligand capture in the CAT family when adopting an outward-open conformation. The cryo-EM map shows continuous, ligand-shaped density in the canonical substrate pocket. Its location and overall pose are consistent with the conserved α -amino-acid binding geometry described across SLC7 transporters in outward-facing substrate-bound states¹⁷⁻²⁰.

The L-ornithine ligand is located between TMs 1 and 6 from the helix bundle domain and TMs 3, 10 and 8 from the scaffold domain (Fig. 1d and Extended Data Fig. 2c). While the local resolution in the pocket (3.06 Å; measured within 5 Å of L-Orn) does not support atom-by-atom features of L-ornithine, the assignment is supported by the surrounding binding-site geometry and complementary biochemical evidence (Fig. 1d-e, Extended Data Fig. 3g).

- The density of the substrate in Fig. 1d and Extended Data Fig. 6a look to me "carved". Furthermore, only one viewing angle is shown, i.e., the same view in Fig. 1d and Extended Data Fig. 6a. If the density was indeed carved, please include the uncarved density in Fig. 1a and Extended Data Fig. 6. In the latter, three different views of the uncarved density (substrate and surrounding interacting amino acids) should be provided to allow a comprehensive assessment of the quality of the obtained substrate density (and surrounding). Finally, please indicate in these two Figures at which threshold the density is displayed.

The density shown in Fig. 1d and the now Extended Data Fig. 8a was not artificial; however, to enable a more direct assessment of the ligand density and its surroundings, we now provide two unmasked density views of the substrate pocket. We kept Fig. 1d and Extended Data Fig. 8 (we removed hydrogens from both), which focused on interaction geometry and cross-transporter comparisons (AtCAT4, GkApcT, MmCAT1), because overlaying all-atom residues with the full pocket density substantially reduces readability in those panels.

To address the reviewer's point without overloading these figures, we added further information to our Extended Data Fig. 2 and cite it in the L-ornithine structure section. Extended Data Fig. 2c shows the complete atomistic modelling of L-Orn-AtCAT4 with focused elements on each protein domain. We included a section focusing on the density around L-Orn, including the surrounding binding-site residues, in two views (the original "top" view used in Fig. 1d and an additional 90-degree orientation). We report the density contour threshold used. We additionally include a panel with hydrogens omitted and heavy-atom distances annotated for the key contacts to demonstrate that the H-bonds are within the usual H-bond distance criteria. (Map contoured at 6.5 σ).

- Major point: Based on Extended Data Fig. 2, only about 12% of particles of the "At SLC7A4-sybody-and-L-ornithine bound" structure apparently bound L-ornithine. This reviewer is not convinced by the final "3D-classification without alignment" run used to select the L-ornithine bound state.

To exclude the possibility of classification artifacts or spurious density being refined as ligand, the authors must verify their procedure with the apo structure. Briefly, perform the same analysis (as in Extended Data Fig. 2, e.g., using same number of 3D classes and same parameters) with the At SLC7A4-sybody apo structure (no L-ornithine) and prove that no comparable density is found in the substrate binding pocket arising from artefacts. This analysis should be included in Supplementary Information as a new Extended Data Figure.

Alternatively, the authors could repeat the cryo-EM experiment using a higher concentration of L-ornithine to achieve improved occupancy of the substrate binding site, thereby providing a clearer and more convincing density for L-ornithine and enabling the reliable construction of a publishable structural model.

We thank the reviewer for this request. We have implemented all the suggested changes and we thank the reviewer for enhancing the robustness of our `L-Orn` bound structural data through their recommendations. Below, we list the modifications we have made:

Validation of the focused 3D classification protocol by an apo negative control: We repeated the identical focused 3D classification protocol (same binding site-focused mask, K = 8, identical parameters) on the apo + SybB5 dataset and observed no ligand-shaped density in the canonical substrate pocket in any class, including the best-resolved class. We have constructed a new Extended Data Fig. 11 summarising the classification results. This supports that the ligand density observed in the L-Orn dataset is not a classification artefact. All volumes were low-pass filtered to 4 Å, normalised to zero mean and unit variance, and contoured at identical σ values. The chimera commands used for this processing are described in the Methods. Collectively, the methods section now includes the following description (starting at line 1066):

Focused 3D classification of the apo + SybB5 dataset as a negative control for ligand-density selection: To assess whether our binding-site-focused 3D classification procedure could generate spurious ligand-like density, we applied the same strategy used for the L-Orn+SybB5 dataset to the apo+SybB5 dataset (no added ligand), using the same aligned focus mask and identical classification settings (K = 8; filter resolution = 2 Å; online-EM batch size per class = 10,000; learning rate = 1; initialization = PCA; class similarity = 0.1; force hard classification = true). Classes were compared after identical normalization and low-pass filtering to 4.0 Å (class07 at 4.5 Å) and were contoured at a common level. Normalization enables comparison of maps at identical contour values, and low-pass filtering standardizes map sharpness across classes with different nominal resolutions for direct visual comparison. No class, including the most feature-rich and interpretable class, showed ligand-shaped density in the canonical substrate pocket (Extended Data Fig. 12) comparable to the density observed for L-Orn in Fig. 1d and Extended Data Fig. 2b-c.

For comparisons between maps from different datasets, the maps were normalised to zero mean and unit variance and contoured at identical σ values. This can be achieved in ChimeraX with `measure mapStats #1` followed by `volume scale #1 shift --mean '1/stdev'`. Maps focused on the binding-site (reported in ED Figure R3) were multiplied by the focus mask using the command `volume multiply #1 #2`. (lines 1066-1086)

Low Orn-bound particle numbers concern: We clarify that the originally cited ~12% refers to the focused-classification step (44,789 / 376,110 AtCAT4 particles that refine well), not the full dataset. To increase ligand-class particle count we have extended the processing and have managed to increase the final ligand-bound reconstruction from 47k to 76k particles (76,191; 3.32 Å), corresponding to ~20% at the focused-classification stage (now presented in the revised version of Extended Data Fig. 2b). Although not a massive improvement we observe and increase in L-Orn side chain density compared to the first submission. The apparently low occupancy is also consistent with biochemical data for L-Orn on purified protein. Specifically, with 10 mM L-Orn added to ~64 μ M AtSLC7A4 (5 mg/mL complex with SybB5) for cryo-EM, the 20% occupancy is consistent with the apparent binding behaviour from our L-Orn titration experiment in nanoDSF (presented in Extended Data Fig. 3g). We have updated the cryo-EM maps presenting the L-Orn binding site with the latest map (Fig. 1d, Extended Data Fig. 2b-c, and Extended Data Fig. 8a).

4. After reading the manuscript, the claim that SLC7A4 proteins are, or could be, pH-responsive amino acid transporters remains somewhat hypothetical, indicating a lack of demonstrated novelty. Additional experimental evidence is needed to substantiate this assertion, which is strongly emphasized in the title but not sufficiently supported or demonstrated in the presented data.

We have now included additional liposome assay data showing that SLC7A4 transports L-Leucine under an acetate-induced pH gradient (Extended Data Fig. 9c-d). However, we do acknowledge that more in-depth biochemical assays are needed to fully understand the relationship between pH and leucine transport via SLC7A4. However, we do consider our data sufficiently robust to support our statement that SLC7A4 is a leucine transporter, which is the first such report in the literature, and is activated under acidic external conditions, which is also the first report that pH regulates human SLC7A1-4/CAT transporter activity.

Significant points to be addressed:

5. The SLC nomenclature should be carefully verified and applied consistently. In my view, it is incorrect to refer to the Arabidopsis thaliana homolog of (human) SLC7A4 as “At SLC7A4”. For instance, solute carriers from mouse are written as Slc7a4 (in contrast to SLC7A4 for the human one). Therefore, in the context of this manuscript, the current “At SLC7A4” designation should be replaced with AtCAT4 or described as “the homolog from Arabidopsis thaliana (or plant) of (human) SLC7A4.”

Importantly, this adjustment would eliminate the existing confusion in the manuscript, where it is not always clear whether the authors are referring to the plant homolog (AtCAT4) or to human SLC7A4.

As the reviewer no doubt appreciates, nomenclature is tricky when different fields work on the same proteins and use different names. Our rationale was to stick with the SLC nomenclature, as our main aim was to study the human transporter. However, we appreciate that, within the field of plant biology, the CAT nomenclature is still widely used.

We have therefore revised the nomenclature throughout to eliminate ambiguity: the Arabidopsis thaliana transporter is now referred to as AtCAT4 (CAT4; UniProt Q8W4K3), which is consistent with the plant physiology field²¹ and the human transporter as SLC7A4, consistent with recent studies on human SLC transporters¹². Where relevant, we explicitly describe AtCAT4 as “the Arabidopsis (plant) homologue of human SLC7A4” at first mention and in figure legends, and we have corrected all instances where “At SLC7A4” could be misconstrued. Similarly, we have included ‘human SLC7A4’ in the Figure titles.

6. Abstract: The Abstract is written in very general terms, making it unclear what was actually accomplished in the present study. While this broad formulation may make the work sound more appealing, it omits key information such as the fact that the structure determined is that of a plant homolog of human SLC7A4, and that the subsequent analyses are based primarily on a homology model of human SLC7A4 derived from this plant structure.

As stated above, this raises serious questions about the necessity of including the AtCAT4 structure, since an equally strong homology model could likely have been built using the already published mouse CAT1 or GkApcT structures, or an SLC7A4 alphafold model. As it stands, the manuscript appears somewhat patchwork-like, combining datasets that do not fit seamlessly together. Therefore, the authors are encouraged to revise the Abstract so that it more realistically reflects the actual scope and content of the presented work, avoiding any impression of overselling.

We appreciate the reviewer’s point here. Our aim in the first submission was to write a general abstract that would appeal to the general science audience of Nature Communications.

Nevertheless, in light of these comments, we have revised the abstract to specifically detail what new data we have generated, the methods used and the conclusions drawn.

Regarding the necessity of our structures in the outward-open state (not captured before for a CAT family member), we do not agree that existing templates (MmCAT1, GkApcT, or even predicted AlphaFold structures) are interchangeable for the questions addressed within our study: these alternatives represent different conformational endpoints (inward-facing or occluded) and therefore cannot substitute for an experimentally determined outward-open structure, which is the state that anchors our residue-level mechanistic interpretation and the design/interpretation of the human mutagenesis and functional assays. We clarify this point in the revised text while keeping the Abstract appropriately concise.

Other points, mainly minor:

- **General comment: Providing the manuscript with line numbering would have been helpful and would have facilitated the review process.**

We have added numbering to the revised manuscript.

- **Page 2 “Structures of many eukaryotic HAT transporters...”: Please define the abbreviation HAT upon first use.**

We have defined HATs in line 57.

- **Page 3: “Nevertheless, we were intrigued by the close homology of SLC7A4 to GkApcT...”. As the authors provided the amino acid sequence identity between SLC7A4 and SLC7A1/2, this reviewer suggests providing also the amino acid sequence identity between SLC7A4 and GkApcT.**

Our revised Extended Data Fig. 1 now shows a multiple sequence alignment of CAT transporters (AtCAT1, 4, 5, 6), the bacterial GkApcT, Human CAT1, CAT2, CAT3, Human LAT1 (SLC7A5; another leucine transporter), all compared to human SLC7A4 as the reference as per the reviewer’s request. To demonstrate that AtCAT4 is a close eukaryotic CAT homologue to SLC7A4, we used our multiple sequence alignment to build the phylogenetic relationship between GkApcT, AtCAT4 and SLC7A4 (Extended Data Fig. 1b), which showed that AtCAT4 is an evolutionary closer homologue to human SLC7A4 than the distant bacterial GkApcT homologue. We have referenced this figure in the main results section “The **plant CAT4 homologue was of particular interest, as it provided a closer evolutionary link to the human transporter than the bacterial counterpart, GkApcT** (Extended Data Fig. 1b).” (lines 120-122).

- **Page 4, last paragraph: This section is written in an unclear manner. The authors report the structure determination of the apo and L-ornithine-bound AtCAT4 in complex with a sybody. However, Figure 1 also includes an apo structure without the sybody. The authors are therefore asked to align the text more precisely with the figure panels in Figure 1.**

In addition, providing the individual resolutions of the obtained structures would strengthen the statement “...to aid particle alignment during data processing...” and help readers without a structural biology background better understand the rationale for using a sybody.

Furthermore, I guess Extended Data Fig. 2 shows the processing workflow for the sybody-bound structures. Please clarify this point, for example by adding the corresponding information to the legend of Extended Data Fig. 2.

We thank the reviewer for this comment. We have revised the text in the Results section to closely align the described structures with the panels in Fig. 1, clearly distinguishing the sybody-bound apo and L-ornithine-bound reconstructions from the sybody-free apo reconstruction. We also now report the individual global resolutions for each reconstruction at

first mention to clarify the rationale for using SybB5 to aid particle alignment during data processing. Finally, we have updated the legend of Extended Data Fig. 2 to state explicitly that it shows the processing workflow for the sybody-bound datasets, and we direct the reader to the corresponding figure for the sybody-free apo processing/workflow. However, due to the complexity of the narrative in the study, we felt that an explicit strict adherence to the panel order in Fig. 1 would disrupt the flow of the paper.

- Page 5, paragraph below Fig. 1 caption (“Although the apo structure...”):

First, the rationale for comparing At SLC7A4 with SLC7A5 and SLC7A7 is unclear - why were these transporters chosen instead of, for example, mouse CAT1 or GkApcT, which appear to be more closely related? Please clarify the reasoning behind the selected structural comparisons by providing pertinent information.

Second, the statement that the structure “...closely resembles...” these transporters is somewhat misleading given the reported RMSD values of 4.5-5.6 Å, which indicate only moderate structural similarity.

We agree that our original wording and the rationale for the comparison required clarification. We compared AtCAT4 to SLC7A5/LAT1 and SLC7A7/y+LAT1 because these HAT light chains are among the few eukaryotic SLC7 members with substrate-bound outward-open/outward-facing cryo-EM structures, thus providing the most appropriate structural reference set for an outward-open binding-pocket comparison.

By contrast, the two available CAT subfamily structures capture different conformational endpoints (i.e., mouse CAT1 is inward-facing¹⁰, while GkApcT is occluded³), making them complementary for understanding the transport cycle but less suitable for direct comparison with an outward-open structure, as reported here for AtCAT4.

We have therefore revised the text to (i) explicitly state that the LAT1/y+LAT1 comparisons are made to benchmark the outward-open/outward-facing state and conserved SLC7 binding-pocket geometry, and (ii) moderate the similarity language: we now state that AtCAT4 shares the conserved APC/LeuT-like fold with these transporters rather than “closely resembles” them, consistent with the reported RMSD values (4.5-5.6 Å) indicating moderate global similarity.

Although the apo AtCAT4 structure adopts the same outward-facing states observed in LAT1/SLC7A5 (PDB: 8KDP)²² and y+LAT1/SLC7A7 (PDB: 9KJU)²⁰, the root mean square deviation (r.m.s.d.) values indicate only moderate global similarity (4.5 Å over 280 Cα and 5.6 Å over 240 Cα, respectively). However, we chose to use these LAT structures for our comparison, as they are the most evolutionarily close reference structures available. In contrast, recent CAT-family structures of SLC7A1 from *Mus musculus*, hereafter referred to as MmCAT1, and GkApcT capture different conformational states (inward-facing apo and occluded)^{3,10,11}. (lines 164-171).

- Page 5, Figure 1d: The reviewer requests visualizing the molecules and hydrogen bonds without including hydrogen atoms, as their positions cannot be assigned with confidence at the achieved resolution. To enable a clearer assessment of the substrate pose within the binding pocket, the reviewer further requests displaying distances (numbers, Å) measured between heavy atoms directly in the figure.

We thank the reviewer for this suggestion. We have removed hydrogen atoms from Fig. 1d and Extended Data Fig. 8. In addition, we now provide an Extended Data panel (Extended Data Fig. 2c) showing the L-Orn binding pocket with hydrogen atoms omitted and with the relevant heavy-atom distances (Å) annotated for the key ligand-protein contacts. We did not

add these distance labels to Fig. 1d because the panel is already information-dense and became visually cluttered when annotated.

- Page 6, "The function of these additional helices is currently unclear. The most likely explanation is that they compensate for the absence of the heavy chain subunit and aid stability or folding of CAT family members."

Since the heavy-chain subunit interacts with the opposite side of the transporter compared to TMs 11 and 12 of At SLC7A4 (as described in the manuscript), it is questionable whether such a compensatory effect can be rationalized.

We thank the reviewer for this point and agree that "compensation for the heavy chain" was imprecise, given that the SLC3 heavy-chain transmembrane helix contacts the opposite face of the transporter.

We have removed this statement and now describe the additional helices (TMs 11-12 in eukaryotic CATs) more conservatively as CAT-subfamily features that may contribute to stability and/or higher-order assembly. Importantly, in many LeuT/APC-fold transporters that lack these extra helices (e.g., AdiC) and in SLC12 cotransporters (e.g., NKCC1 and KCC2), the homodimer interface is formed by the terminal helical pair of the core fold (TM11-TM12 in those proteins), which corresponds to TM13-TM14 in eukaryotic CATs. In this architecture, the CAT-specific TMs 11-12 sit adjacent to, and could flank or support, the TM13-TM14 face rather than directly substituting for the SLC3 heavy-chain interaction. We conclude by suggesting a potential hypothetical role in homodimerization of CAT transporters, currently beyond the scope of this work.

Although the function of these additional helices is currently unclear, the placement of the CAT-specific TM helices (TM11-12) creates an extended hydrophobic surface compatible with protein-protein contacts and raises the possibility that CAT-family members may form higher-order assemblies. In related LeuT/APC-fold transporters that lack these extra helices (e.g., AdiC) and in SLC12 cotransporters (e.g., NKCC1 and KCC2), homodimers are formed via the terminal helical pair of the core fold (TM11-TM12), which corresponds to TM13-TM14 in eukaryotic CATs (Fig. 1b); in this architecture, the CAT-specific TMs 11-12 would flank and potentially support the TM13-TM14 dimerisation interface^{23,24}. Given that SLC7 HATs already function as obligate heterodimers with an SLC3 heavy chain, it will be interesting to determine whether the CAT subfamily also employs oligomerisation as a regulatory mechanism. (lines 180-191).

- Page 6, "...occupied by a Lauryl-maltose neopentyl glycol (LMNG) detergent molecule (Extended Data Fig. 3e). The acyl chains of the detergent..." and onward to the end of this paragraph. The authors are asked to provide cryo-EM density proof for those statements in an Extended Data Figure.

We have now included the requested cryo-EM density of the LMNG molecule we find bound on AtCAT4+Syb Orn-bound complex, which we have included in Extended Data Fig. 4f. The equivalent site for the apo AtCAT4+Syb dataset is shown in Extended Data Fig. 3c. In both case the map threshold is stated in the figure legend.

- Pages 6-7, "We thus conclude that whilst we cannot rule out the impact of the sybody on the position of EL4 in the higher-resolution structure, our data suggest that EL4 is dynamic and may flip away from the transporter in the outward-facing state."

The authors have addressed the potential impact of sybody binding; however, was the possibility also considered that the observed "flipped-out" conformation of EL4 might result solely from its interaction with the LMNG molecule? The reviewer believes that this possibility should be examined before concluding with confidence that EL4 is inherently dynamic.

We thank the reviewer for raising this point, and as they note, we are acutely aware of overinterpretation in cryo-EM experiments. As described above in our response to the flexibility of EL4 and summarised in the new Extended Data Fig. 5a, our new MD analysis supports our original hypothesis regarding the flexibility of EL4. Our interpretation of the data is that EL4 exists in dynamic equilibrium between flipped-in and flipped-out states, with the LMNG molecule stabilising the latter state.

We therefore interpret the LMNG density as occupying and stabilising a pre-existing hydrophobic cavity that becomes available when EL4 disengages in the outward-open state, rather than as the primary driver of EL4 displacement.

This interpretation is supported by the MD simulations shown in Extended Data Fig. 5a, which were performed without LMNG yet still show substantial EL4 mobility/outward-open EL4 behaviour. Consistent with this interpretation, focused classification of the low-resolution Syb dataset (also in LMNG) yields a minor class with an alternative, more "flipped-in" EL4 placement (Extended Data Fig. 5b), supporting the idea that EL4 can sample multiple configurations even under identical detergent conditions. We have revised the main text to explain and cite the data presented in Extended Data Fig. 5.

- Page 7, "...well-defined densities for lipid molecules (Extended Data Fig. 5a-b)."

Based on the information presented in Extended Data Fig. 5a, this reviewer remains unconvinced by the authors' claim of "well-defined density." Please clarify this statement and provide additional views of the corresponding density in an Extended Data Figure to substantiate the claim.

We have now modified the language to 'In the cryoEM map for the L-Orn-bound structure, we observed several densities, which are consistent with lipid molecules (Extended Data Fig. 7a-b)'. We have shown the density we discuss in Extended Data Fig. 7a.

- Page 7, "Our data indicate this pocket is a conserved cholesterol-binding site within the CAT sub-family and is worth further study to understand any regulatory role cholesterol might have on CAT transporter function."

Referring to the site as "conserved" appears to be an overstatement given that this conclusion is based on a single structure. Please revise the wording accordingly to avoid overinterpretation.

We thank the reviewer for this point. We have revised the text to: "Our data indicate that this pocket is broadly conserved within the CAT subfamily and warrants further study to understand any regulatory role cholesterol might play in CAT transporter function" and now state that we observe cholesterol in this cavity in AtCAT4, while broader sterol occupancy and functional relevance remain to be established.

We additionally note two lines of support for describing the pocket as conserved at the level of pocket chemistry: (i) the residues lining the cavity are broadly conserved across CAT-family homologues by BLOSUM62 similarity scoring relative to AtCAT4 (e.g., MmCAT1 83% similarity; human CAT1/SLC7A1 83%; human SLC7A4 78%; table shown), consistent with

preservation of a hydrophobic sterol-compatible surface; and (ii) an equivalent sterol ester (CHS) has been modelled in the corresponding pocket in the MmCAT1 structure (PDB: 9FQT), albeit in an inward-open conformation, indicating that this cavity can accommodate sterols in another CAT-family member and in a distinct state of the transport cycle.

Note: We report BLOSUM62 similarity (not strict identity) for residues aligned to the AtCAT4 pocket (F66-A72, P222-A233, I543, Y546, L547, N550), which is appropriate for lipid-binding cavities where conservative hydrophobic/aromatic substitutions are expected to preserve pocket geometry and chemistry.

	AtCAT4 reference
AtCAT4	-
SLC7A4	78
SLC7A1	83
MmCAT1	83
AtCAT1	78
HsCAT2B	74
HsCAT3	74
AtCAT2	91
AtCAT3	91
AtCAT5	70
AtCAT6	74
AtCAT7	83
AtCAT8	78
AtCAT9	57

Table 1 Similarity matrix for residues aligned to AtCAT4 residues F66-A72, P222-A233, I543, Y546, L547, N550 (SLC7A4 residues: F75-A81, P231-A242, L555, C558, L559, K562) that were identified to interact with cholesterol in PDB 9HJK using ChimeraX.

- Page 8, " In contrast, the ϵ -amino group on the L-Orn side chain interacts with Asp116 on TM3 via a salt bridge and to Asp243 on TM6b via a water-mediated hydrogen bond (W1) (Fig. 1d and Extended Data Fig. 6a)."

This reviewer is concerned that, at the achieved resolution, reliable placement of water molecules may be challenging. Are the positions of the modeled water molecules supported by the MD simulations presented later in the manuscript? If not, please remove this statement since not support by well defined, high-resolution density.

We thank the reviewer for raising this point. We agree that at the nominal and local resolution of the L-Orn map, explicit placement of individual water molecules is not sufficiently supported

by the density alone. In the revised model, we have therefore removed W1/W2 and deleted the statement about a “water-mediated hydrogen bond” in the main text.

At the same time, our MD simulations consistently support the presence of a bridging water at this position: in five independent 200-ns repeats, a single water molecule occupying the gap between the L-Orn ϵ -amino group and Asp116 is present for ~98% of the simulation time (Extended Data Fig. 9d-e), which is mechanistically reasonable given that the side chain is an Asp (rather than a longer Glu) and the ligand is ornithine (shorter than Lys/Arg). We now mention this as supportive mechanistic evidence only in the MD section, making clear that the water is inferred from simulation rather than directly resolved in the updated cryo-EM map.

Across the five seeded simulation repeats, we consistently observe a water molecule occupying the gap between the L-Orn ϵ -amino group and Asp116, present for ~98% of the gross simulation time (Extended Data Fig. 9d-e). The presence of water in this position within the binding site is mechanistically reasonable, especially given that the interacting AtCAT4 side chain is an aspartate, which is shorter than glutamate, and the ligand is ornithine, which is shorter than either lysine or arginine. Thus, a water molecule can stabilise the bound L-Orn molecule, as observed with the shorter aspartate side chain in AtCAT4. Collectively, the MD data suggest a coordinated water network is likely to exist within the AtCAT4 binding site. However, due to the resolution of the cryo-EM dataset we obtained, we could not reliably model the water molecules into the structure. (lines 409-419)

- Page 8, “The structure of SLC7A4 bound to L-Ornithine (L-Orn) exhibits...”
I assume this structure corresponds to the sybody-bound form. Please clarify this point and revise the text accordingly, as it is not clearly stated in the current version.

We have addressed this by changing the protein nomenclature.

- Page 8, “...forming a short π -helix bulge...”
This is an interesting structural feature; however, Fig. 1c appears to show an elongated helical element rather than a bulge. Please provide a more appropriate image to illustrate this feature accurately.

We have removed the word ‘bulge’ from the text.

- Page 8, “...water-mediated hydrogen bond...”
The obtained resolution (>3 Å) is generally insufficient to support such detailed structural conclusions. Please revise the statement accordingly.

We thank the reviewer for this point. Our original interpretation was guided by both extra density features near L-Orn and MD simulations, which consistently show long-lived water bridges between the L-Orn ϵ -amino group and nearby acidic residues, making a water-mediated interaction mechanistically plausible, especially for the shorter ligand L-Orn (compared to L-Lys and L-Arg). However, we agree that at the local resolution of the L-Orn site (3.0 Å) these waters cannot be assigned with sufficient confidence from the cryo-EM map alone. In line with the reviewer’s recommendation, we have therefore removed explicit “water-mediated hydrogen bond” claims and the modelled waters from the EM description, and now only discuss these bridging waters in the MD section as simulation-derived mechanistic insight, not as directly resolved structural features. We provided the revised passages above.

- Page 9, “The binding pose we observe for L-Orn in SLC7A4 is also consistent with the pose we reported previously for the more distantly related prokaryotic homologue, GkApcT (PDB 5OQT), which also involves an interaction of the side chain guanidino group with an acidic side chain on TM337 (Extended Data Fig. 6c).”

Please rephrase this sentence to clearly indicate that the structure also includes GkApcT bound to L-arginine. In addition, note that the PDB code differs from that provided in Extended Data Fig. 6c and actually corresponds to the alanine-bound GkApcT (likely a typographical error). Considering the major concern raised regarding the ligand cryo-EM density, the authors are kindly requested to reassess whether the available density sufficiently supports the structural comparisons between the binding sites of SLC7A1 and GkApcT.

We have changed the description to the following to make this clear. "The binding pose we observe for L-Orn in SLC7A4 is also consistent with the pose we reported previously for the more distantly related prokaryotic homologue, GkApcT (PDB 6F34), which was captured bound to L-Arginine. In the GkApcT L-Arg bound structure we also observed an interaction of the side chain guanidino group with Glu115 on TM3³ (Extended Data Fig. 8c)".

- Page 11, "Compared to the cryo-EM model, Ala44 is shifted by an additional ~ 1.7 Å over the span of the MD repeats, further drawing the ligand backbone carboxyl into the fully accessible GAG backbone pocket." The MD simulations are presented as supporting evidence for the proposed induced-fit mechanism. However, this reviewer remains uncertain about the interpretation of the events observed during the simulations. Specifically, the further unwinding of TM1, the entry and coordination of a water molecule and the "better accommodation" of the substrate. Why are these features not observed in the cryo-EM data and what is their significance in the context of the proposed mechanism? Please adapt text and provide additional information and argumentation.

We thank the reviewer for this point. Our MD simulations were performed after embedding the cryo-EM starting model in an explicit lipid bilayer and solvent and removing the sbody and detergent, so local relaxation and sampling of nearby minima is expected. The additional TM1 unwinding, water entry/coordination, and the ~ 1.7 Å shift around Ala44 therefore represent MD-observed relaxation within the outward-open ensemble under a membrane environment, not a separate structural state we claim is resolved by cryo-EM. At ~ 3.2 – 3.3 Å nominal resolution and with local heterogeneity, small-amplitude backbone adjustments and dynamic/partial-occupancy waters are expected to be averaged out and thus not appear as distinct density features. These simulations used the widely applied AMBER ff99SB-ILDN protein force field and the SLipids membrane force field, both of which have been extensively benchmarked and used for membrane-protein dynamics and local conformational transitions^{25,26}.

- Page 11, Figure 2, panels b-d: The extent of the unwinding motion around the GAG-motif is a bit difficult to assess from the provided views of TM1 and TM6. Maybe superimposed structures of a closer view around the binding pocket would be helpful to improve assessment.

We appreciate the reviewers' comments regarding Fig. 2. However, the style we chose aligns with previous representations of the SLC7 fold in the literature²⁷, so we hope to retain it in the final version.

- Pages 19-20: The description of the proposed transport model suffers from issues of sequencing and redundancy, which negatively affect readability and comprehension. The induced-fit binding of L-leucine is described after the transition from the outward-open, substrate-free state to the outward-facing occluded, L-leucine-bound state (i.e., the transition from state i to ii), which disrupts the logical flow. Additionally, protonation of Glu125 is mentioned twice (passages 1 and 2) and could be streamlined. The hypothesized rotation of TM3 into the position observed in the AtCAT4 structure is also described somewhat differently than on page 19. This reviewer therefore requests that this paragraph be restructured and improved for greater clarity and consistency.

We have addressed this in our revised paragraph of the mechanism discussion section. It is in the “transport model” block immediately before Fig. 6 (lines 699-737).

- Page 26, “Thermal stability measurements”. Please report on the final protein concentration and buffer constituents used for the assay/measurement.

The concentrations we reported in our original submission were actually the final protein concentrations, but we have rephrased it in the Methods section to say “final concentrations of…” to make the final concentrations used in the assay clear.

-Extended Data: The Extended Data section does not include representative micrographs, 2D class averages or viewing direction (azimuthal angle) distribution plots. Please add these essential elements to provide a more complete overview of the cryo-EM data quality and processing.

Done.

Reviewer 4.

The manuscript by Kolokouris et al. focusses on the transport mechanism of SLC7A4, an amino acid transporter of the solute carrier protein family. Unlike homologous proteins some structural and functional features are unique, which are shown by means of cryoEM structures, MD simulations, and pharmacological experiments. In particular two aspects are interesting and deserve publication. First the unique profile of transported amino acids (selective for leucine). And second, the pH sensitivity of the transporter. I want to state that I am not an expert in the technical aspects of cryoEM, so I cannot fully judge on this particular part.

Given the interesting topic, the sound methodology used, and the way of presenting the results, I would like to recommend publication of this manuscript in Nature Communications. However, some aspects have to be improved by the authors before publication.

We thank the reviewer for their positive assessment and recommendation.

- 1) While the MD simulations have been carried out in a sound manner, I was confused by their description. E.g., I assume that the Table 2 is right, but in the methods section the authors state they have simulated in triplicates rather than 5 independent 200 ns simulations. It is also not clear from the text how many μ s MDs have been carried out and from which of those replicas the clusters were selected. Please make the method section consistent and describe in more details how the clustering and selection of the representative frame was done.

We thank the reviewer for their comment. We have rephrased the Methods section on the MD simulations we ran to align with the information presented in Extended Data Table 2. We also rephrased the relevant main text section to clearly state the purpose of each simulation and how seeded runs were generated to allow multi-repeat fingerprint sampling.

- 2) As a minor point, the authors mix one-letter and three-letter code for amino acids. Since it makes sense for the figures to use one-letter code, please also apply this throughout the manuscript.

We appreciate the reviewer's perspective, but after discussion with colleagues, we believe that using the three-letter code in the main text enhances readability. If the reviewer feels strongly, we can revise the manuscript accordingly, but our preference is to retain the three-letter code in the main text and use the single-letter codes in the figures.

- 3) I miss a detailed structural comparison with other SLCs/amino acid transporters. Since new structures are reported in this manuscript, it would be nice to compare them with previous structures and highlight the unique features in more detail.

We appreciate the reviewer's comment and, in principle, agree. Indeed, we have included several sections in the paper that directly compare our AtCAT4 structure with closely related members of the SLC7 and APC family of transporters, as illustrated in Extended Data Figures 4, 7, and 8. However, we feel that including any further comparisons could confuse the paper's narrative. We are currently working on a review paper that places our work in the context of the wider SLC7/APC family, which will hopefully address this point.

- 4) The section on structural basis for leucine selectivity in Hs SLC7A4 is quite interesting but misses a detailed discussion on the molecular level. The authors show the lipophilic residues that are interacting with leucine's side chain and then directly head to arginine. It would be beneficial to discuss more closely related amino acids, e.g. valine, isoleucine. Why is the binding site so prone to leucine?

Several closely related α -amino acids can access the conserved SLC7 pocket and competitively inhibit leucine uptake, but our data indicate that transport is much more selective than binding. In particular, although non-polar amino acids can compete in the assay context, direct ³H-L-isoleucine uptake is indistinguishable from the transport-deficient mutant, and L-Leu shows markedly higher apparent affinity than L-Arg (IC_{50} $24 \pm 5 \mu M$ vs $680 \pm 19 \mu M$). This pattern is consistent with a balanced requirement for side-chain fit that couples binding to the conformational activation step: very small side chains (e.g., Ala) may under-occupy the hydrophobic cavity and fail to stabilise the induced-fit network, whereas bulkier branched side chains (e.g., Ile) can still block access yet impose a packing mismatch that prevents progression through the transport cycle. Structurally, the leucine side chain is accommodated by a hydrophobic pocket centred on Val249, Val332, Leu336, L394 and Tyr246 (the thin extracellular gate), providing a concrete basis for why leucine is uniquely positioned to support transport rather than binding.

We added this clarification in the main text:

The hydrophobic pocket, V332/L336/L394, is tuned for a branched side chain of leucine, providing a simple explanation for why closely related BCAAs can compete for access yet fail to show measurable uptake (e.g., isoleucine does not show detectable transport, (Fig. 3d-e), consistent with a balanced requirement for side-chain fit that couples binding to the conformational activation step. (lines 555-560)

References.

1. Majd, H. et al. Screening of candidate substrates and coupling ions of transporters by thermostability shift assays. *Elife* **7**(2018).
2. Parker, J.L. et al. Molecular basis for redox control by the human cystine/glutamate antiporter system xc(). *Nat Commun* **12**, 7147 (2021).
3. Jungnickel, K.E.J., Parker, J.L. & Newstead, S. Structural basis for amino acid transport by the CAT family of SLC7 transporters. *Nat Commun* **9**, 550 (2018).
4. Gauthier-Coles, G. et al. Quantitative modelling of amino acid transport and homeostasis in mammalian cells. *Nat Commun* **12**, 5282 (2021).
5. Beyer, S.R. et al. Identification of cysteine residues in human cationic amino acid transporter hCAT-2A that are targets for inhibition by N-ethylmaleimide. *J Biol Chem* **288**, 30411-30419 (2013).
6. Christensen, H.L. et al. The choroid plexus sodium-bicarbonate cotransporter NBCe2 regulates mouse cerebrospinal fluid pH. *J Physiol* **596**, 4709-4728 (2018).
7. Bernardino, R.L., Carrageta, D.F., Sousa, M., Alves, M.G. & Oliveira, P.F. pH and male fertility: making sense on pH homeodynamics throughout the male reproductive tract. *Cell Mol Life Sci* **76**, 3783-3800 (2019).
8. Wang, Y. et al. Mechanisms of transport and analgesic compounds recognition by glycine transporter 2. *Proc Natl Acad Sci U S A* **122**, e2506722122 (2025).
9. Mostyn, S.N. et al. Identification of an allosteric binding site on the human glycine transporter, GlyT2, for bioactive lipid analgesics. *Elife* **8**(2019).
10. Xia, L. et al. Structural insights into cationic amino acid transport and viral receptor engagement by CAT1. *Nat Commun* (2025).
11. Mingda, Y. et al. Amino acid and viral binding by the high-affinity Cationic Amino acid Transporter 1 (CAT1) from *Mus musculus*. *Research Square* (2024).
12. Wiedmer, T. et al. Accelerating SLC Transporter Research: Streamlining Knowledge and Validated Tools. *Clin Pharmacol Ther* **112**, 439-442 (2022).
13. Goldmann, U. et al. Data- and knowledge-derived functional landscape of human solute carriers. *Mol Syst Biol* **21**, 599-631 (2025).
14. Jansson, T. Amino acid transporters in the human placenta. *Pediatr Res* **49**, 141-7 (2001).
15. Lobel, M. et al. Structural basis for proton coupled cystine transport by cystinosin. *Nat Commun* **13**, 4845 (2022).
16. Zimmermann, I. et al. Generation of synthetic nanobodies against delicate proteins. *Nat Protoc* (2020).
17. Jeckelmann, J.M. et al. Structure, Function and Pharmacology of SLC7 Family Members and Homologues. *Chimia (Aarau)* **76**, 1011-1018 (2022).
18. Yan, R. et al. Mechanism of substrate transport and inhibition of the human LAT1-4F2hc amino acid transporter. *Cell Discov* **7**, 16 (2021).
19. Yang, H. et al. Structural insights into the substrate transport mechanism of the amino acid transporter complex. *J Biol Chem* **301**, 110569 (2025).
20. Dai, L. et al. Structural basis for the substrate recognition and transport mechanism of the human $\gamma(+)$ LAT1-4F2hc transporter complex. *Sci Adv* **11**, eadq0558 (2025).
21. Su, Y.H., Frommer, W.B. & Ludewig, U. Molecular and functional characterization of a family of amino acid transporters from *Arabidopsis*. *Plant Physiol* **136**, 3104-13 (2004).

22. Lee, Y. et al. Structural basis of anticancer drug recognition and amino acid transport by LAT1. *Nat Commun* **16**, 1635 (2025).
23. Xie, Y. et al. Structures and an activation mechanism of human potassium-chloride cotransporters. *Sci Adv* **6**(2020).
24. Yang, X., Wang, Q. & Cao, E. Structure of the human cation-chloride cotransporter NKCC1 determined by single-particle electron cryo-microscopy. *Nat Commun* **11**, 1016 (2020).
25. Lindorff-Larsen, K. et al. Improved side-chain torsion potentials for the Amber ff99SB protein force field. *Proteins* **78**, 1950-8 (2010).
26. Beauchamp, K.A., Lin, Y.S., Das, R. & Pande, V.S. Are Protein Force Fields Getting Better? A Systematic Benchmark on 524 Diverse NMR Measurements. *J Chem Theory Comput* **8**, 1409-1414 (2012).
27. Nicolas-Arago, A., Fort, J., Palacin, M. & Errasti-Murugarren, E. Rush Hour of LATs towards Their Transport Cycle. *Membranes (Basel)* **11**(2021).